



# NorESM-bcoi-v1: A bias-corrected reanalysis of ocean biogeochemistry at >40° N, 1980–2020, based on a global ocean model hindcast

Philip J. Wallhead[1], Jörg Schwinger[2], Jerry Tjiputra[2], Trond Kristiansen[3], Richard G. J. Bellerby[1]

[1]Norwegian Institute for Water Research, 5006 Bergen, Norway
[2]NORCE Climate & Environment, Bjerknes Centre for Climate Research, 5838 Bergen, Norway
[3]Farallon Institute, Petaluma, CA 94952, USA

*Correspondence to*: Philip J. Wallhead (philip.wallhead@niva.no)

**Abstract.** Accurate and gap-free historical datasets of oceanic nutrient concentrations, dissolved oxygen, and carbonate chemistry, are needed for the analysis of anthropogenic impacts and development of predictive models, including as boundary conditions for regional ocean models. We developed low-bias, quasi-optimal 4D interpolations of observed values at latitudes >40° N, excluding Mediterranean and Black Seas, for years 1980–2020 inclusive. Our approach used output from the NorESM-OCv1.2 ocean model hindcast as a basis for kernel-smoothing bias correction and optimal interpolation of anomalies, obtaining monthly reanalysis data at NorESM grid points (30–90 km spatial resolution) and all model depth levels. Based on cross-validation tests with spatially-separated training and test subset locations, the resulting products showed equal or better interpolative accuracy compared to standard climatological products from the World Ocean Atlas 2023 and GLODAPv2, and also compared to existing non-climatological products from machine learning and biogeochemical reanalysis/hindcast datasets from CMEMS/Copernicus. The new products showed skill in tracking smooth seasonal and interannual variations not included in the standard climatologies, and improved spatial coverage and detail resolution in northern/Arctic regions, compared to all tested alternatives. Significant uncertainties remained, however, largely due to sparse observational coverage in seasonal and permanent ice-covered regions, and limited resolution in coastal regions. The NorESM-bcoi-v1 reanalysis data, as well as the associated quality-controlled in situ observational compilations (NORBGCv1) and code used to develop the reanalysis, can be downloaded from https://doi.org/10.5281/zenodo.14525787 (Wallhead et al., 2024).

## 1 Introduction

The ability of the oceans to sustain life and absorb anthropogenic $CO_2$ depends strongly on their internal concentrations of dissolved nutrients, oxygen, and inorganic carbon (Sigman and Hain, 2012). It is therefore important to understand how and why these concentrations vary over space and time, such that we might predict the response of marine systems to future changes, and for this purpose we need accurate data on their historical variability. This need can in part be met with ocean observations, including ship-borne bottle samples and sensors, remote sensing data from satellites, and sensors installed on





floats/buoys and underwater vehicles such as gliders. However, the oceans are vast, and deep, and such observations generally only cover a small fraction of the total volume and period of interest. Methods are needed to gap-fill and interpolate the observations in some way, providing complete or 'gridded' datasets to facilitate analyses of historical change and the development of predictive models. This latter includes regional or 'downscaling' ocean models, which enable investigation of marine system responses at finer resolutions than typical global climate models, but which often require these concentrations

to be specified on their boundaries.

The need for gap-filled ocean datasets is perhaps especially acute at high northern/Arctic latitudes. Here, historical ocean datasets are more limited, largely due to the remote locations, harsh conditions, and presence of sea ice, which has acted as a barrier to ship-borne sampling, remote-sensing, and float/glider operation. Nevertheless, these regions are undergoing some of the strongest anthropogenic trends, including the decline of Arctic sea ice (IPCC, 2023), Arctic amplification of ocean

warming (Shu et al., 2022) and acidification (Qi et al., 2022), and rapid changes in surface nutrient concentrations (Zhuang et al., 2021) and primary productivity (Arrigo and van Dijken, 2015; Dalpadado et al., 2020).

One way to obtain gridded datasets is to apply a statistical interpolation method such as objective analysis directly to the historical observations. This approach has been widely applied to ocean temperature and heat content to obtain 4D monthly datasets (e.g. Cheng et al., 2024). For biogeochemical variables, however, the much sparser observational coverage may lead

to over-smoothing of important spatiotemporal variability and may necessitate compromises in the target level of temporal or spatial resolution in this approach. For example, the objective analysis of dissolved oxygen by (Ito, 2022) targets the 5-year running mean concentrations, while the World Ocean Atlas gridded products for nutrients and dissolved oxygen (Garcia et al., 2024a, b) target monthly climatological means over 58 years, and the GLODAPv2 products for carbonate chemistry (Lauvset et al., 2016) target annual climatological means over 42 years. Another potential difficulty is that biogeochemical observations

can include large errors due to measurement and reporting (e.g. confusion of units, decimal places, and variables) and these can strongly influence gridded estimates based only on observed values, if not corrected or filtered out by quality control procedures, especially where sampling is sparse relative to the target resolution.

Another approach is to build statistical or machine learning models to predict the target biogeochemical variables using existing gridded products for physical ocean variables, remote sensing, and climatological products, as well as spatio-temporal

coordinates, as predictor variables. This approach has been mostly applied to predict 3D surface carbonate chemistry variables (e.g. Chau et al., 2024; Lee et al., 2006) but has also more recently been applied to generate 4D dissolved oxygen products (Sharp et al., 2023; Xue et al., 2024). It has the advantage of exploiting a wider pool of information beyond the target observational data and may thereby achieve greater spatiotemporal resolution and robustness to outliers. However, it does not exploit output from mechanistic, process-based biogeochemical ocean models, which integrate various sources of information

on biogeochemical fluxes and transformations (rates of planktonic uptake, mineral dissolution, etc.) within a mass-conserving hydrodynamic framework of advection-diffusion-reaction equations. The approach therefore avoids influence of the many biases in mechanistic biogeochemical models, but at the same time may result in sub-optimal estimates, particularly where





extrapolation is required over sparsely sampled regions/depths, or when the focus is on underlying trends that may comprise a small fraction of the total observational variance.

A further approach is to produce a biogeochemical "reanalysis" – a quasi-optimal synthesis of historical information from observations and mechanistic ocean models, based on a consistent statistical method, usually involving sequential or variational data assimilation (see e.g. Bennett, 2002). These latter methods originated in the field of numerical weather prediction, and have subsequently been adapted for applications to physical ocean models, and more recently to ocean biogeochemical models (see Fennel et al., 2019 for a recent review). In theory they may offer an ideal solution, because

information from both observations and mechanistic models is exploited, and the observations are used to correct the model simulation online in such a way that dynamical consistency is mostly preserved, and other simulated variables and fluxes, beyond those for which observations are assimilated, can also be improved. This approach has proved particularly effective for physical variables such as temperature and salinity, and has resulted in widely-used global products such as SODA3 (Carton et al., 2018) and GLORYS12 (Lellouche et al., 2021). However, biogeochemical reanalyses seem to suffer from more

significant residual biases, well beyond levels expected due to measurement and representativity error, and the improvement in fit to observations relative to the free-running model hindcast is sometimes modest/partial. Fennel et al., (2019) highlight the more-limited observational datasets and methodological challenges (stronger non-linearity and non-Gaussianity) in biogeochemical applications, but other factors may include the complexity of model and observational error sources and uncertainty in how to represent them, as well as larger biases in the original model hindcast, driven by greater uncertainties in

biogeochemical model structure and parameter values (Schartau et al., 2017). These approaches also tend to have a high computational cost, often requiring tens or hundreds of model runs, which in turn may limit the scope to provide biogeochemical reanalyses over broad regions and multiple decades.

In this study we apply a simplified data assimilation approach with the aim of producing low-bias, quasi-optimal 4D interpolations of the historical observations for (nutrients, dissolved oxygen, dissolved inorganic carbon, and total alkalinity)

in oceanic regions at latitudes >40° N, excluding Mediterranean and Black Seas, and for years 1980–2020 inclusive. The work was originally motivated by the need to provide accurate biogeochemical boundary conditions for regional-downscaling ocean models, to explore the impacts of climate change in coastal regions. The approach involves post-hoc corrections of an existing ocean model hindcast, without any additional model runs, and consists of two steps: 1) bias correction by kernel-smoothing of model-observation residuals, and 2) adjustment for anomalies by objective analysis (aka. 'kriging' or 'optimal interpolation')

over time and horizontal space. It is applied to output from a multidecadal hindcast (i.e. driven by reanalysis atmospheric forcing) generated using the ocean-component of the Norwegian Earth System Model (NorESM-OCv1.2, Schwinger et al., 2016), resulting in reanalysis products NorESM-bcoi-v1, standing for "bias-corrected optimal interpolation". The new products are compared primarily with climatological products from WOA23 (Garcia et al., 2024a, b) and GLODAPv2 (Lauvset et al., 2016), since these are most commonly used to provide regional model boundary conditions, but we also compare with

machine learning based estimates for dissolved oxygen (Sharp et al., 2023) and surface carbonate chemistry (Gregor and





Gruber, 2021), as well as two widely-used CMEMS/Copernicus products for Arctic biogeochemical reanalysis (Wakamatsu et al., 2022a, b) and global biogeochemical hindcast data (Perruche, 2019). In addition to the NorESM-bcoi-v1 reanalysis data, we also provide, in the same repository, the collections of quality-controlled in situ observations NORBGCv1 that were used for product development, including the independent training/test subset identifiers such that other methods can be

compared with ours using the same cross-validation tests.

## 2. Methods

Figure 1 shows a schematic diagram of our methods, referring to the subsections below for detailed descriptions.

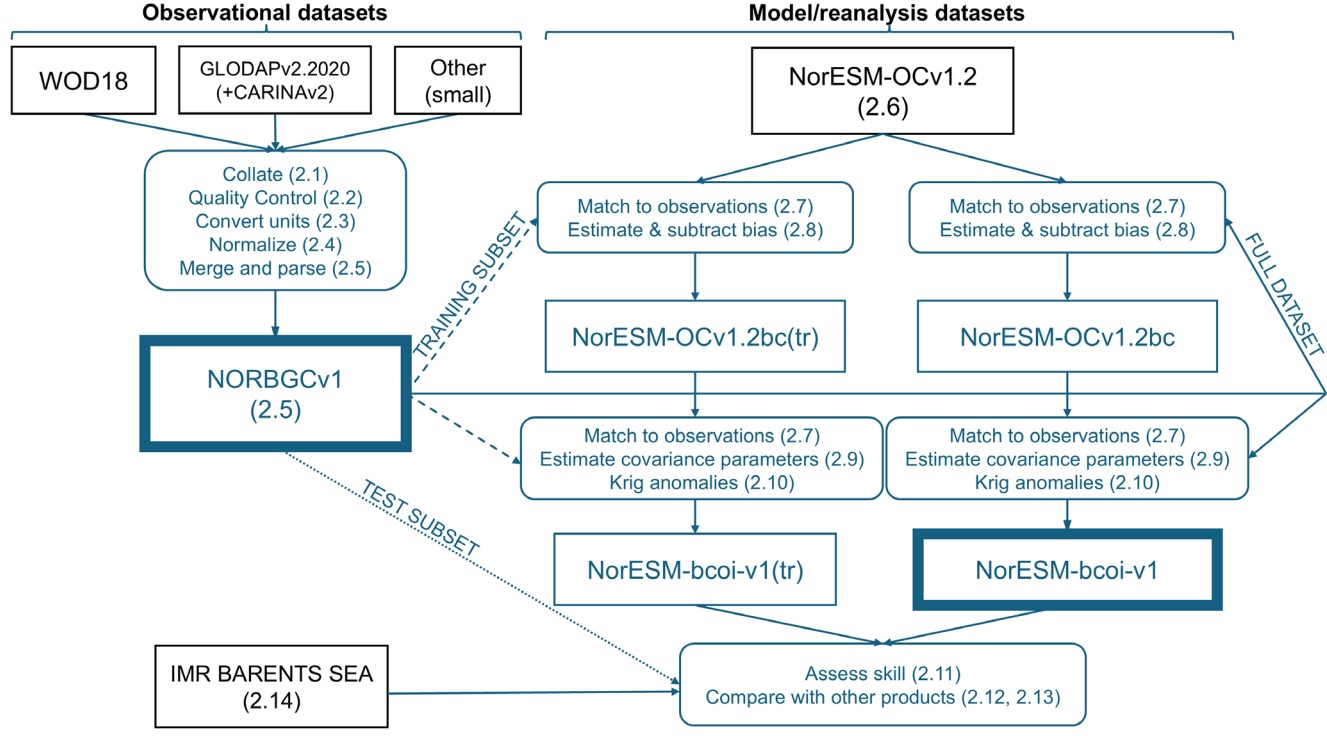

**Figure 1.  Schematic diagram of products and methods applied in this study, showing observational products on the left and model/reanalysis products on the right.  Black boxes show external source datasets, while blue boxes show our methods (rounded corners) and data products (square corners, with thicker borders for final products provided in the repository).  Lines with arrows are: solid for full dataset usage, dashed where only training subsets are used, stippled where only test subsets are used (see section 2.11).  Numbers in parentheses refer to subsections of Methods where the methods or datasets are explained in detail.**




## 2.1 Observational datasets

Our primary data source was the World Ocean Database (WOD18, Boyer et al., 2018), including measurements from: bottle samples (type OSD), Conductivity-Temperature-Depth probes (CTD), and profiling floats (PFL, which includes most of the Bio-Argo data). Data were also collated from the Global Ocean Data Analysis Project, version 2 (GLODAPv2.2020, Key et al., 2015; Olsen et al., 2016a, 2020), the CARINA Iceland and Irminger Sea Time Series version 2 (Olafsson et al., 2009, 2010), the Norwegian Environment Agency (NEA, https://vannmiljo.miljodirektoratet.no/), the Shirshov Institute on Oceanology (SIO, A. A. Polukhin pers. comm.), and cruise datasets from the MICROPOLAR (A. L. King, pers. comm.) and POMPA (E. Yakushev, pers. comm.) projects. We used (WOD, GLODAP, NEA, SIO, POMPA) datasets for nitrate ($NO_3$), nitrite ($NO_2$), phosphate ($PO_4$), silicate (Si), and dissolved oxygen ($O_2$), and (GLODAP, CARINA, NEA, MICROPOLAR) datasets for dissolved inorganic carbon (DIC) and total alkalinity (TA).

Most models and many observational datasets do not resolve nitrate vs. nitrite, so we targeted the sum ($NO_3+NO_2$) for the model product. WOD reports a mixture of $NO_3$ and $NO_3+NO_2$ as '$NO_3$'; we used these directly as estimates of $NO_3+NO_2$. For (GLODAP, SIO, NEA) we corrected $NO_3$ data to $NO_3+NO_2$ using the following multiple linear regression (MLR) relationship, fitted to log-transformed GLODAP data, to predict missing $NO_2$ data: $NO_2 = 0.1089 \times \exp(-0.0232 \times NO_3 - \frac{depth}{2485})$, with ($NO_2$, $NO_3$) in µmol kg$^{-1}$ and depth in metres (predicted $NO_2$ was <0.2 µM, so usually negligible).

To maximize (DIC, TA) data coverage, measured profiles of partial pressure of $CO_2$ ($pCO_2$) from the CARINA Irminger and Iceland Sea time series were merged into the GLODAP data. Reported $pCO_2$ values at in situ temperature (T) and laboratory pressure (P) were converted to values at in situ (T, P) by: i) converting back to values at laboratory (T, P) by undoing the 0.0423 °C$^{-1}$ temperature correction (Olafsson et al. 2009, 2010); ii) converting to in situ (T, P) using CO2SYS.m v2.0 (Lewis and Wallace, 1998; van Heuven et al., 2011) with dissociation constants from Lueker et al., (2000), Dickson, (1990), and Uppström, (1974). Missing Si and $PO_4$ were filled using our final reanalysis products where these could be interpolated in 4D, otherwise we took the climatology of our reanalysis output at the sampled month and nearest horizontal wet grid point and interpolated over depth, allowing constant extrapolation. Where laboratory T data were missing, we converted directly from the reported $pCO_2$ (at in situ T, atmospheric P) using CO2SYS and applied a linear bias correction to account for inaccuracy in the 0.0423 °C$^{-1}$ correction. The uncertainty added by this latter was estimated as ~1.5 µatm, i.e. less than the reported measurement uncertainty of 3 µatm. After merging into GLODAP, missing measured values of (DIC, TA) were filled with calculated values using the above method with CO2SYS and pairs of measured carbonate chemistry variables, following a similar priority order as that of GLODAP (Olsen et al., 2020), namely: (DIC/TA, pH), else (DIC/TA, $pCO_2$), else (DIC/TA, $fCO_2$) (where $fCO_2$ is the fugacity of $CO_2$).

## 2.2 Quality Control (QC) of observational data

We attempted to apply the primary QC flags, i.e. those from the source dataset managers, as strictly as possible. With WOD we used only data pre-interpolated over depth to standard depth levels, to leverage the more sensitive WOD QC flags on

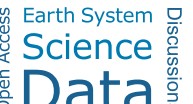

standard vs. observed levels (including region/depth-specific range checks, Garcia et al., 2018) and to avoid excessive vertical resolution in the CTD dataset for $O_2$. We used only WOD data with standard level QC flag = 0 ('accepted value'). For
GLODAP and CARINA data, we required primary QC flag = 0 ('approximated'), 2 ('good'), or 6 ('replicated'; Key et al., 2010; Olsen et al., 2016). The other datasets had no primary QC flags but were subjected to the secondary QC analyses described below.

After applying the primary QC flags, we applied our own secondary QC using a semi-manual procedure: i) the observational data were binned into coarse grid cells (1000×1000 km) on a polar stereographic projection, and depth bins based on WOD
standard levels; ii) outliers were 'proposed' for each cell and depth bin using the scaled median absolute deviation from the median (Leys et al., 2013); iii) for each cell, bad data were identified subjectively by examining distributions over depth, year, and sampled month, and considering the proposed outliers. For nutrients and (DIC, TA), we also considered deviations from model predictions based on cell-specific MLR functions of $(S, T, O_2)$, using a threshold of 10 residual standard deviations to propose outliers. We assumed that temporal variability in deep water data (>1000 m) should be consistently low, such that
outliers likely reflect analytical/sensor error rather than real intermittent variability. Having identified bad data, usually first in deep samples, we first checked the contributing profiles and flagged whole profiles if they showed evidence of high measurement imprecision (spikes, 'irregular' vertical variability) or bias (including confusion of reported units). In a small minority of cases, a correction factor for units confusion was identified with high certainty, and was therefore applied to the data (noting the correction with a different QC flag value). We checked all profiles from the contributing cruise(s) and flagged
whole cruises of data if such problems appeared to be widespread. We also flagged bottom water data where these showed evidence of possible benthic boundary layer effects, since these could not be represented at the resolution of the reanalysis data. Allowances were made for possible coastal effects (e.g. river inputs) by plotting data locations with respect to land masses (note: coastal data were thus not excluded).

Figure 2 illustrates the added value of our secondary QC, using WOD-OSD $NO_3$ as an example. All data shown have passed
the primary provider QC (WOD QC flag = 0) and are restricted here to locations >40 km from land, excluding the Baltic. Magenta dots show data flagged as suspect and excluded by our secondary QC, while cyan dots show data corrected for units confusion and retained by our secondary QC. The exclusions and corrections significantly improve the overall agreement between measured values and values predicted by regional MLR functions of $(S, T, O_2)$. The majority of exclusions/corrections are applied to data collected in the 1990s and especially the 1980s. Similar results are obtained for WOD-OSD $PO_4$ and WOD-
OSD Si (Figs. A1, A2).

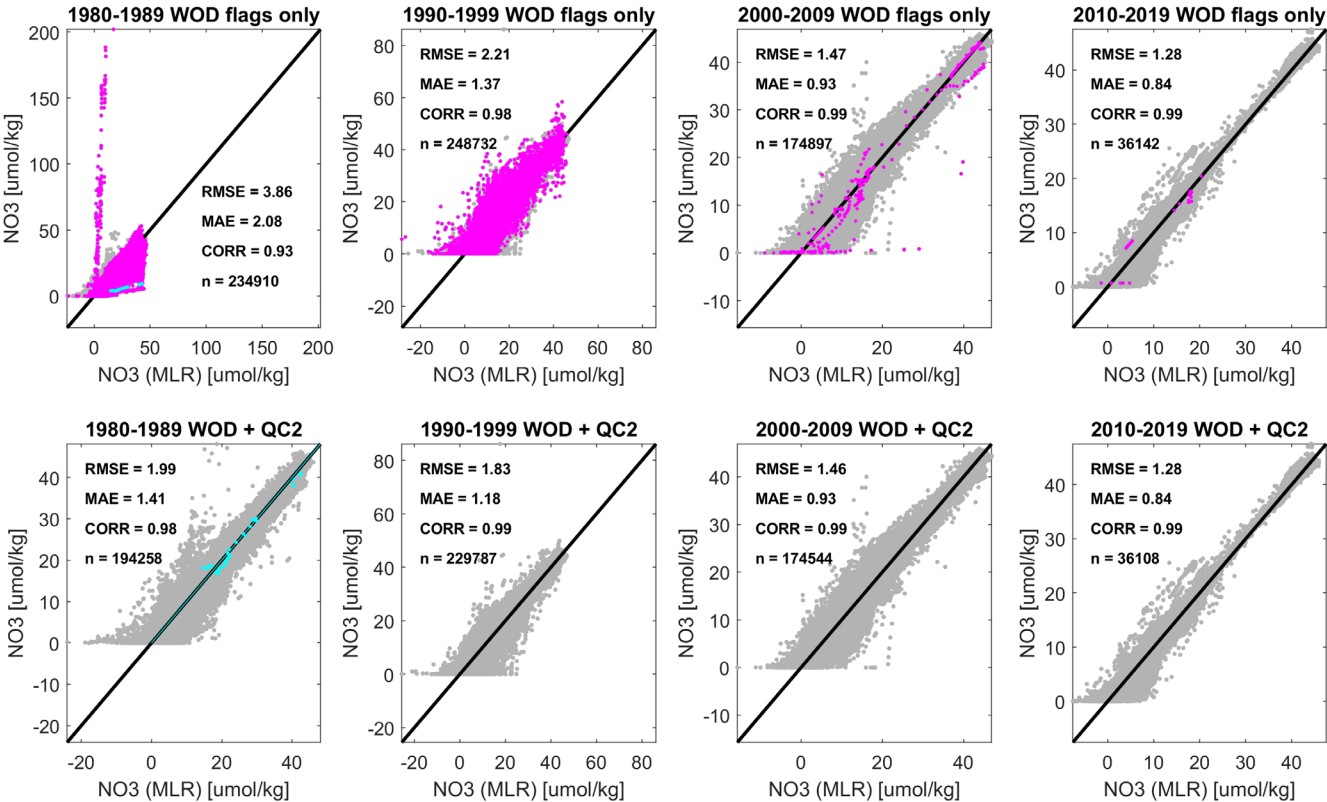

**Figure 2. Secondary quality control (QC) comparison of nitrate-plus-nitrite concentrations from the World Ocean Database (2018) bottle sample dataset (OSD), showing predicted concentrations from regionalized Multiple Linear Regression (MLR) models (functions of salinity, temperature, and dissolved oxygen, x-axes) with corresponding observed values (y axes), both in mass-specific**
**units (μmol kg[-1]), for different decades (left to right subplots), and before/after applying the secondary QC exclusions and corrections (upper/lower subplots). Magenta colour denotes suspect data based on secondary QC, and cyan denotes data that are corrected for units confusion by the secondary QC. All data shown have passed the primary QC from WOD18 (flag value = 0, 'accepted' values). Black lines are 1:1, and skill metrics measuring agreement level between observed and modelled values include: Root Mean Square Errors (RMSE), Mean Absolute Errors (MAE), Pearson correlations (CORR), and the number of model-observation pairs (n).**

**2.3 Units conversion to in situ volume-specific concentrations**

For each dataset, we converted units to volume-specific molar concentrations (μmol L[-1] = μM) at *in situ* temperature and pressure, these being the concentrations that are usually simulated in biogeochemical models. Mass-specific concentrations were converted using in situ densities calculated from matching (S, T) measurements (where available), applying an empirical relationship to estimate pressure from sample depth where necessary (Key et al., 2010). Where volume-specific concentrations

in measurement conditions were reported, we corrected using ratios of in situ : measurement density, calculated assuming a laboratory measurement temperature of 22 °C for nutrients, and the potential density for dissolved oxygen, following Olsen et al., (2020). If matching in situ (S, T) were not available we assumed standard values (35 psu, 5 °C) within the correction ratios; the median/maximum errors due to this approximation were estimated from the GLODAP dataset as 0.04/0.4%, which is likely within measurement uncertainty. For the WOD data, mass-specific concentrations were reported, but our extraction

did not record whether or not these were converted by WOD using the simplified factor 1.025 kg m$^{-3}$ (Boyer et al., 2018). For WOD-PFL and WOD-CTD datasets we assumed that the simplified factor was not applied. For WOD-OSD nutrient data we assumed that the original data were volume-specific in measurement conditions (22 °C, atmospheric pressure) and that the simplified factor had been applied; hence our conversions first 'undid' the use of the simplified factor. For WOD-OSD $O_2$ data, we hedged our bets by assuming mass-specific values that were averages of: 1) the values reported by WOD, and 2) those

obtained by undoing use of the simplified factor and assuming original units as volume specific at the potential density. This resulted in maximum (2.7, 1.4)% corrections to the mass-specific (nutrients, $O_2$) concentrations provided by WOD-OSD.

**2.4 Normalization of (DIC, TA) data**

To normalize the (DIC, TA) data, we first derived domain-wide MLR relationships using (S, T, $O_2$, P) as predictor variables (see Fig. 3 and Table 1). Here we used all the GLODAP data (>40° N), after filtering with primary and secondary QC flags

on both predictor and response variables, including both measured and CO2SYS-derived (DIC, TA) values. The regressions explained (94, 90)% of the variance in the (DIC, TA) data respectively (Fig. 3a,f). The strongest effects were from salinity (positive, Fig. 3b,g) then there were strong negative effects from $O_2$ (Fig. 3d,i) and weaker but significant effects from T and P (Fig. 3c,h,e,j). The fitted relationships were then used to normalize or 'correct' the (DIC, TA) data to standard conditions of (35 psu, 5 °C, 300 μM, 0 dbar). These normalizations reduced the interquartile ranges of (DIC, TA) by factors (0.40, 0.50)

respectively.

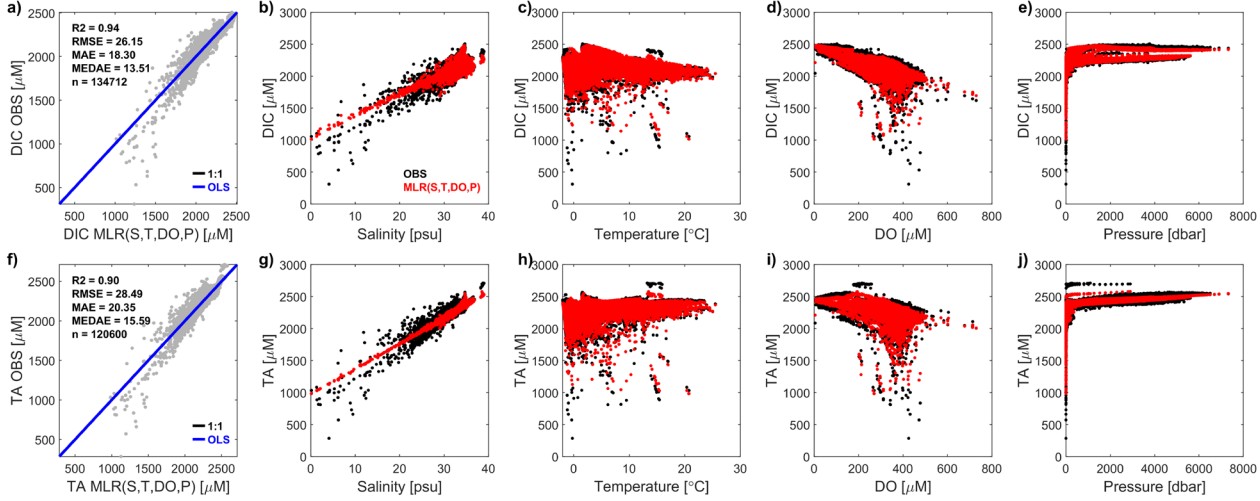

**Figure 3. Multiple Linear Regression (MLR) models used to normalize the data for dissolved inorganic carbon (DIC, upper) and total alkalinity (TA, lower) in in-situ volume-specific units (μmol L$^{-1}$). Leftmost subplots show scatterplots of observations (y axis) vs. fitted values (x axis), together with fitted Ordinary Least Squares regression lines (OLS, blue), 1:1 lines (black, covered by OLS**

**lines), and various skill metrics including: the R-squared values (fractions of variance explained, R2), the Root Mean Square Errors (RMSE), the Mean Absolute Errors (MAE), the median absolute errors (MEDAE), and the number of model-observation pairs (n). Remaining subplots show the fitted observational data (black dots, from the GLODAPv2 dataset) and the corresponding MLR modelled values (red dots), plotted against the 4 regressor variables: salinity, temperature, dissolved oxygen, and pressure.**





**Table 1.** **Fitted parameter values in multiple linear regression models used to normalize dissolved inorganic matter (DIC) and total alkalinity (TA). DIC and TA observations in both volume-specific and mass-specific units were regressed on salinity (S), temperature (T), dissolved oxygen ($O_2$), and pressure (P) using the model: $y = a_0 + a_1(S − S_0) + a_2(T − T_0) + a_3(O_2 − O_{20}) + a_4(P − P_0)$, where $(S_0, T_0, O_{20}, P_0) = (35\ psu, 5\ °C, 300\ \mu M, 0\ dbar)$ for volume-specific models and $(S_0, T_0, O_{20}, P_0) = (35\ psu, 5\ °C, 300\ \mu mol\ kg^{-1}, 0\ dbar)$ for mass-specific models. Therefore to convert a normalized variable ($y_n$) back to in situ concentrations (y) the user should apply: $y = y_n + a_1(S − S_0) + a_2(T − T_0) + a_3(O_2 − O_{20}) + a_4(P − P_0)$. Standard errors were very small (<2% of values shown) except for $a_4$ in the model for DIC [$\mu mol\ kg^{-1}$] (15%).**

| Predictand | Intercept ($a_0$) [$\mu M$ or $\mu mol\ kg^{-1}$] | S slope ($a_1$) [$\mu M\ psu^{-1}$ or $\mu mol\ kg^{-1}\ psu^{-1}$] | T slope ($a_2$) [$\mu M\ °C^{-1}$ or $\mu mol\ kg^{-1}\ °C^{-1}$] | $O_2$ slope ($a_3$) [$mol\ mol^{-1}$] | P slope ($a_4$) [$\mu M\ dbar^{-1}$ or $\mu mol\ kg^{-1}\ dbar^{-1}$] |
|---|---|---|---|---|---|
| DIC [$\mu M$] | 2193.5 | 29.311 | -11.637 | -0.89603 | 0.011107 |
| TA [$\mu M$] | 2364.5 | 39.123 | -1.655 | -0.30337 | 0.020614 |
| DIC [$\mu mol\ kg^{-1}$] | 2127.7 | 26.877 | -11.025 | -0.89406 | 0.000034 |
| TA [$\mu mol\ kg^{-1}$] | 2299.0 | 36.524 | -1.320 | -0.30052 | 0.009282 |

### 2.5 NORBGCv1: High-QC in situ observational data compilations for >40° N, 1980–2020

Observational datasets for biogeochemical variables were finally merged and parsed to produce the 'NORBGCv1' compilations, provided in the same repository used for the NORESM-bcoi-v1 data (see Sect. 4). These consist of 3 NetCDF files: NORBGC_NUTv1.nc, containing all ($NO_3+NO_2$, $PO_4$, Si) data with coincident (T, S, $O_2$) data where available; NORBGC_DOv1.nc, containing all $O_2$ data with coincident (T, S) data where available; NORBGC_CO2v1.nc, containing all (DIC, TA) data with coincident (T, S, $NO_3+NO_2$, $PO_4$, Si, $O_2$) data where available. The rationale here is to provide useful auxiliary variables for developing statistical models, e.g. for further multiparameter quality control, or for gap-filling missing data. NORBGC_CO2v1.nc also includes measured and CO2SYS-derived data for (pH, $pCO_2$) and saturation states ($\Omega_{ar}$, $\Omega_{ca}$), using the CO2SYS settings described in Sect. 2.1. Bad data identified by primary and secondary QC are retained in these datasets so that users may reassess QC judgments and avoid bad data re-entering when combining with other datasets. The parsing procedure identified duplicate profiles as those on the same day within a horizontal location tolerance of 1 km; these were restricted to one source, with higher preference for GLODAP/CARINA, assuming a higher overall level of QC for these data. Source variables are provided to record data provenance (e.g. WOD-OSD), and new QC flag values are provided to identify suspect data based on primary and secondary QC, or where corrected by our secondary QC, or where calculated with CO2SYS (see flag variable attributes). Concentrations are provided as both mass-specific and in situ volume specific values. The repository also includes pdf documents summarizing the cell-by-cell secondary QC analyses, allowing the user to review our QC judgments and focus on data of interest using the illustrated latitude/longitude/depth and year/month distributions.

Table 2 summarizes the composition and sample sizes of the NORBGCv1 datasets, as full datasets and under QC restrictions. It shows that our secondary QC flags exclude larger fractions of the (WOD-dominated) nutrient datasets than the primary QC flags (7-8% vs. 1-2%), reflecting a generally stricter level of quality control. By contrast for the carbonate chemistry data, mainly based on GLODAP/CARINA with already a high level of QC, our secondary QC results in very few additional flag values (<0.2%). It should be noted however that these datasets do not include 'underway' sensor data, which may provide the majority of available in situ data (e.g. for $pCO_2$) but tend to be restricted to near-surface measurements.





**Table 2. Composition and sample sizes for different target variables in the NORBGCv1 in situ observational data compilations for >40N, 1980-2020. Target variables include: nitrate-plus-nitrite ($NO_3+NO_2$), phosphate ($PO_4$), and silicate (Si) (see NORBGC_NUTv1.nc), dissolved oxygen ($O_2$, see NORBGC_NUTv1.nc), dissolved inorganic carbon and total alkalinity (DIC and TA, see NORBGC_CO2v1.nc). Source datasets for each variable are listed with % compositions based on the full sample size after parsing, with no exclusions due to quality control (QC) (see third column). Sample size post QC1 shows the number of target**
**variable data with acceptable primary QC flag values from the dataset providers (0 for WOD datasets; 0-2-6 from GLODAP/CARINA). Sample size post QC1+QC2 shows sample sizes after further requiring acceptable flag values from our secondary QC (NORBGC flag values 0-1-2). Sample size matched to NorESM shows the number of matchups to NorESM hindcast data that are used to develop the NORESM-bcoi-v1 reanalysis (i.e. the number passing QC1+QC2 and located where the model can interpolate without coastward extrapolation). The final column shows the initial test subset sample sizes, prior to restrictions on**
**distance from coastlines and matchup with other non-NorESM products.**

| Target Variable | NORBGCv1 source datasets | Full sample size (parsed) | Sample size post QC1 | Sample size post QC1+QC2 | Sample size matched to NorESM | Test subset sample size (initial) |
|---|---|---|---|---|---|---|
| $NO_3+NO_2$ | WOD-OSD(76%), WOD-PFL(11%), GLODAP(12%), NEA(1%), SIO+POMPA(<0.1%) | 2395444 | 2376891 | 2189338 | 1287577 | 52457 |
| $PO_4$ | WOD-OSD(89%), GLODAP(10%), NEA(1%), SIO+POMPA(<0.1%) | 2807351 | 2775755 | 2550071 | 1498936 | 60277 |
| Si | WOD-OSD(86%), GLODAP(13%), NEA(1%), SIO+POMPA(<0.1%) | 2368611 | 2346822 | 2191606 | 1328028 | 52885 |
| $O_2$ | WOD-OSD(47%), WOD-CTD(22%), WOD-PFL(25%), GLODAP(4%), NEA(2%), SIO+POMPA(<0.1%) | 8032715 | 7318122 | 6948005 | 5041252 | 204247 |
| DIC | GLODAP+CARINA(97%), NEA(3%), MICROPOLAR(<0.1%) | 152082 | 152082 | 151898 | 129047 | 7263 |
| TA | GLODAP+CARINA(97%), NEA(3%), MICROPOLAR(<0.1%) | 134363 | 134363 | 134176 | 113587 | 7208 |

## 2.6 Global ocean hindcast model: NorESM-OCv1.2

NorESM-OCv1.2 is the ocean carbon-cycle stand-alone configuration of the Norwegian Earth System model (NorESM). It is based on NorESM1, but with some notable improvements in ocean physics. The biogeochemical model is based on the HAMOCC model as described in (Ilyina et al., 2013) with some modifications to adapt the model to the isopycnic coordinate

of the ocean model, and an updated carbon chemistry following the recommendations of (Dickson et al., 2007). NorESM-OCv1.2, the model set-up, the reanalysis forcing, and the simulations used here are described in detail in (Schwinger et al., 2016). The model configuration has been evaluated by (Hauck et al., 2020) and used in annual updates of the Global Carbon Budget (e.g. Friedlingstein et al., 2023) as well as the RECCAP2 assessment (REgional Carbon Cycle Assessment and Processes phase 2; DeVries et al., 2023, Rodgers et al., 2023).




## 2.7 Calculation of model-observation residuals

We calculated residuals using the observations and model output, interpolated over space and time to the exact 4D positions of the observations. First, latitude-longitude pairs from observations and model grid points were converted to Cartesian (x,y) coordinates using a polar stereographic projection. Then, for each observation point, the model output was interpolated horizontally by linear scattered interpolation to the horizontal location of the observation, for the 4 modelled depth-time points enclosing the observation depth/time. Finally, these 4 points were interpolated by bilinear interpolation. No extrapolation was allowed over time or horizontally towards model coastlines (this was achieved by including land mask cells with missing values in the interpolation mesh). However, we did allow constant vertical extrapolation of the shallowest model values (at 1.25 m) towards the surface, and constant downwards extrapolation of the deepest non-masked values as far as the first masked model depth level, or without limit below the deepest layer (at 7312.5 m). The final set of residuals was formed for each variable by subtracting observational from interpolated model values.

## 2.8 Estimation of model bias

For each variable, model bias was estimated as a 4D climatological field (latitude, longitude, depth, day-of-year) on the original model grid but with day-of-year (days since January 1st) as a periodic time variable with target points at the middle of each month. First, the residuals were linearly interpolated over depth to the fixed model depth levels, here allowing constant vertical extrapolation. Next, for each month and depth level, the bias was estimated at the target (latitude, longitude, day-of-year) points by kernel smoothing the residuals (Hastie et al., 2009). Here we computed horizontal separations as spherical distances and day-of-year separations respecting periodicity. The horizontal and day-of-year separations were scaled using bandwidth parameters, then summed in quadrature before applying the exponential function to determine weights. A further weighting was applied where residuals were vertically extrapolated; this had value $\left(\frac{max\,(10,min(zd,zp))}{max\,(10,max(zd,zp))}\right)^{\gamma}$, where $zd$ is the depth of the closest (deepest or shallowest) raw residual, $zp$ is the depth of the extrapolated residual, $\gamma$ controls the severity of down-weighting, and a capping value of 10 m acts to switch off the weighting within a presumably well-mixed surface layer.

Based on a set of bandwidth selection experiments, using test observational profiles excluded from the bias estimation, we set spatial and seasonal bandwidths at 50 km and 15 days, and $\gamma = 5$ for all variables (we also tested a Cauchy kernel − this did not improve skill). To avoid implausible seasonal signals in the bias at deep depths, we added a depth dependence to the seasonal bandwidth such that it doubled with every 1000 m depth increase, starting from 15 days at zero depth. To avoid excessive extrapolations of bias over space and seasons in the sparsely sampled Arctic regions, we added zero-valued pseudo-data at each grid point, such that the hindcast output acted as 'prior' data to stabilize the bias estimates. These pseudo-data were subject to an additional weighting factor of 0.002, which was damped rapidly to zero south of 65° N. For Si and normalized carbonate chemistry variables (DICn, TAn) the maximum weighting was reduced to 0.0002 to account for relatively sparser sampling (for DICn, TAn) and for stronger biases in the hindcast model output (for Si). Residuals used for smoothing to each target grid point/day-of-year were limited to the 250 data with highest overall kernel weight.





After calculating the bias, we subtracted it from the monthly model output (for each month and depth level) to obtain bias-corrected hindcasts for 1980-2020.

**2.9 Estimation of covariance parameters for anomaly kriging**

First, we repeated the residual calculation (Sect. 2.7) using the bias-corrected output instead of the original model output, and vertically interpolated these residuals to the model depth levels (here allowing no extrapolation). Then, for each depth level, we estimated covariogram parameters as follows: i) outliers were filtered using the median absolute deviation (Leys et al., 2013); ii) initial covariance estimates were made by averaging the products of residual pairs within 2D bins of temporal and spatial lags; iii) covariance estimates were smoothed onto bin centres using a 2D, second-order kernel smoothing (local linear regression, exponential kernel) with bandwidths decreasing and data weightings increasing with increasing bin sample size; iv) a parametric covariance function (separable exponential) was fitted to the smoothed data by least squares, weighted by the product of the number of bin pairs and the covariance magnitude; v) the parameter estimates were finally smoothed over depth level using a 3-point median filter. In step (iv) the full (or 'sill') variance $V$ was fixed as the sample variance and the difference relative to the covariance at zero lag $C_0$ (i.e. the 'nugget' variance) was constrained to be at least as large as the estimated measurement precision for each variable. Final estimates showed a general increase in temporal and spatial decorrelation scales with depth, from around 200-500 days and 300-600 km in surface layers to around 1000-2000 days and 1000-2000 km in deep layers. We tested fixing the nugget variance to the measurement precision and refitting the scale parameters; this gave no improvement in skill.

**2.10 Calculation of 4D reanalysis products**

For each depth level and month during 1980-2020, we calculated a simple kriging estimate (Cressie, 1993) of the anomaly field, using the residuals and covariogram parameters for each depth level (Sect. 2.9) and considering each model grid point in turn. To limit computational expense, only the 20 residuals most correlated with the prediction point were used, and only residuals within a maximum temporal lag of 400 days were considered. These kriging anomalies were added to the bias-corrected output (Sect. 2.8) to form the final reanalysis.

For (DIC, TA), our methodology (Sect. 2.7–2.9) proceeded with the normalized variables (DICn, TAn) for both model and observations (see Sect. 2.4), and produced reanalysis products for (DICn, TAn). We also provide final (DIC, TA) products by converting back to in situ conditions using (S, T) from the SODA3.4.2 reanalysis product (Carton et al., 2018), $O_2$ from NORBGC-bcoi-v1, and pressure empirically from depth. However, if the user has access to more accurate local products for (S, T, $O_2$) e.g. from a regional coastal ocean model, we recommend that our normalized products (DICn, TAn) be converted by the user back to in situ conditions, using the relevant coefficients in Table 1. Note that the normalization model accounts mainly for basin-scale correlations driven by freshwater content and net community production, while the reanalysis of (DICn, TAn) accounts for local/regional variability and temporal variations not related to warming, freshening, or deoxygenation.



## 2.11 Assessment of model product skill

Skill of the NorESM model products was assessed using matchup pairs with product values interpolated in 4D to the positions and times of the in situ observations, as in Sect. 2.7, but here only allowing vertical extrapolation towards the sea surface. To assess interpolative skill, we divided the observational data into a 'training' subset, for developing a provisional model product, and a disjoint 'test' subset. Note, however, if we were to assign the test subset entirely at random, the excluded test data would usually be located at stations that are observed at other times within the training subset. As such, we would be mostly testing
the ability to interpolate observations over time, whereas we expect users to want to estimate values at some distance from observed locations. A better approach might exclude a random subsample of the distinct observed locations (stations); however, because of the tendency for stations to be clustered, this would still result in a weak test of spatial generalization relative to user needs.

To address this 'lax cross-validation' problem, we designed a station selection procedure to achieve training-test subsets that
statistically mimic the level of spatial interpolation implied by the final product data (see Appendix B1 for details). To illustrate, Fig. 4 shows the probability distributions for $NO_3+NO_2$ of minimum horizontal distances from grid points to any observed station (blue) in comparison to the minimum distances from test to training subset stations, under both a simple random assignment (green) and following our selection procedure (magenta). If the test stations are chosen at random from all distinct observed locations, then the test stations end up lying too close to the training stations (green), resulting in an
insufficiently challenging test and an over-optimistic skill assessment. Our selection procedure improves the cross validation design, though there is still some under-representation of the upper tail (grid points furthest from observed locations). Median distances from test to closest training stations were similar to the median distances from grid points to closest stations (32−38 km for nutrients, 25 km for $O_2$, 80 km for DIC, TA), although the 75th percentile values were 30-50% lower, reflecting a shortage of sufficiently-isolated stations (see Table C1). Hence our skill assessments may be still slightly optimistic, especially
in more poorly-observed regions, although this may be partly offset by the fact that all observations are used in the final reanalysis products (the whole process 2.7–2.10 is repeated with no test data excluded). The resulting spatial distributions of training and (initial) test subsets for $NO_3+NO_2$ are shown in Fig. 5.

Given the basin-scale focus of this study, and to avoid potential influence of near-coastal extreme values (e.g. of nutrient concentrations), we further restricted the test data subsets to only include data ≥40 km from any real coastline, using the
intermediate resolution data from GSHHGv2.3.7 (Wessel and Smith, 1996). Also, for comparing skill between different products, we further restrict to the common subset of test data that can be reached by all products without extrapolation in time, downwards, or 'coastwards' towards land masses, such that exactly the same set of observations is used in each product-observation comparison. For comparing with WOA23 and L16 products, these further restrictions lead to 'final' common test subsets that are ca. 15–20% smaller than the initial test subsets shown in Table 2. The NORBGCv1 files include indicator
variables to identify training subsets, initial test subsets, and final test subsets under comparison with WOA23/L16, so that users can exactly replicate our skill assessment.



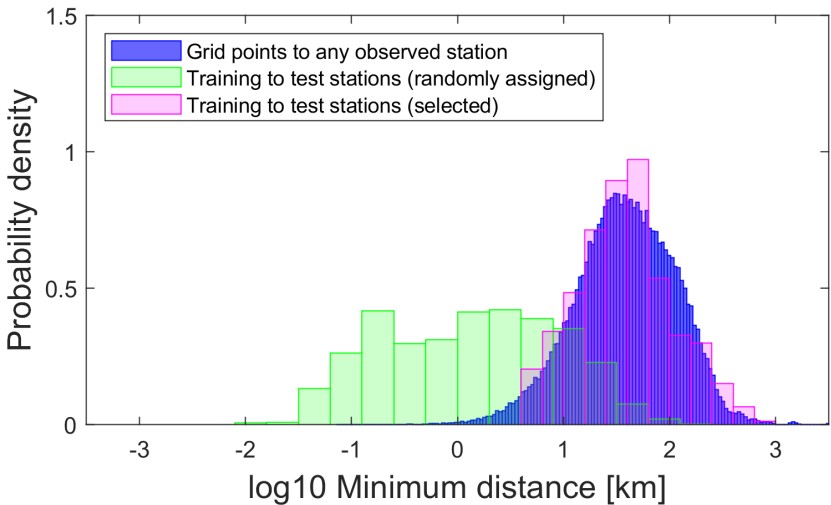

**Figure 4.** Probability distributions for nitrate-plus-nitrite of distances from model grid points to closest observational stations (blue), and of distances between test subset stations and closest training subset stations, first under simple random assignment (green) and then following our selection procedure (magenta). This latter was designed to provide more stringent cross validations that aim to measure interpolative accuracy in typical applications of the reanalysis products.

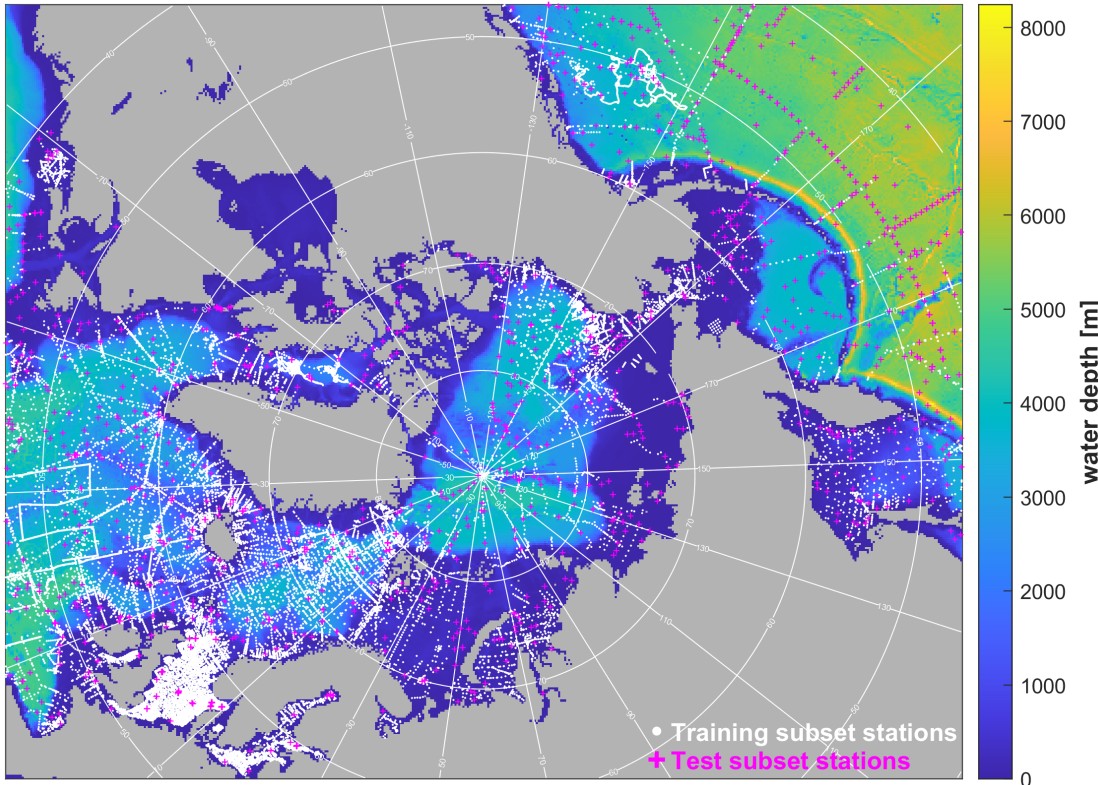

**Figure 5.** Training and test observational datasets for nitrate+nitrite (white dots and magenta crosses respectively). No test data locations (stations) overlap with training data locations, and test locations are selected to mimic the level of spatial interpolation implicit in application of the final model product. Background shows ETOPO1 bathymetry and white lines mark latitude/longitude on a polar stereographic projection.



## 2.12 Skill and climatology comparisons with standard climatological products

We compare the skill of our (nutrients, $O_2$) and (DIC, TA) products and with that of the World Ocean Atlas (WOA23) objective analysis climatologies (Garcia et al., 2024a, b) and the GLODAP mapped products of Lauvset et al. (2016) (L16) respectively.

These latter are considered the most standard climatologies for these variables, both provided at 1° horizontal resolution. For matching WOA data to observations, monthly climatologies were combined with repeated annual climatologies below 800 m (1500 m for $O_2$), and WOA data were interpolated bilinearly over (longitude, latitude, depth, day-of-year), with WOA values placed at the 15th of each month and augmented with (December, January) values at days-of-year (-17.25, 379.25) respectively. Native mass-specific concentrations were converted using in situ densities calculated from WOA23 salinity and temperature

monthly climatologies for the period 1980-2010, filled with interpolated seasonal climatologies at depths >1500 m. The annual L16 climatological data were matched to observations by bilinear interpolation over (longitude, latitude, depth), and converted to in-situ volume-specific concentrations using the annual salinity/temperature values provided in the L16 product (note: this reduced L16 product coverage slightly due to extra missing values in the L16 temperature data). For computing matchups, both products were augmented over longitude (using data from (179.5° E, -179.5° E) at (-180.5° E, 180.5° E) respectively)

and over latitude (repeating data at 89.5° N at 90.5° N) to avoid data loss or extrapolation. As for the NorESM products, extrapolation was allowed only in the vertical towards the sea surface.

For climatological comparisons, climatologies for NorESM products were computed for the 30-year period 1985–2014 inclusive. This should be comparable to the WOA23 climatologies for 1965–2022 (Garcia et al., 2024), and to L16 climatologies for 1972–2013 (but for DIC corrected to represent 2002 conditions, see Lauvset et al., 2016). To enable a fair

comparison of product coverage within the climatology plots, we used the same horizontal interpolation method, namely scattered interpolation after polar stereographic projection, and we retain the product land mask values in this process, thus generating missing values that are not gap-filled (e.g. by nearest neighbour values). For products on regular latitude-longitude grids the scattered interpolation was found to give a slight improvement in coverage over bilinear interpolation, but otherwise no significant differences in interpolated values. These figures use the colour map 'batlowW' (Crameri 2018; Crameri et al.,

395 2020).

## 2.13 Skill comparisons with alternative non-climatological gridded products

With insufficient resources to consider all interesting alternatives, we selected some candidate products of high relevance and topical interest.

As a statistical modelling product for $O_2$ we chose the GOBAIv2.2 product, which applies machine learning to Argo and

GLODAP bottle sample data to derive 4D global $O_2$ estimates for (2004–2022 inclusive) at (1°, monthly) resolution (Sharp et al., 2023). Native mass-specific concentrations were converted using the accompanying GOBAI salinity and temperature data to calculate in situ density. Matchups were calculated by bilinear interpolation of GOBAI output over (longitude, latitude, depth, time) to the observational coordinates, for each observation converting product pressure coordinates to depth using an

empirical function (Key et al., 2010) dependent on the observation latitude. Both product and observation longitudes were
converted to the range 20–380° E during matchup to achieve monotonic product coordinates. Due to the partial latitudinal
coverage (≤79.5° N), the product was not augmented over latitude.

As a statistical modelling product for surface (DIC, TA) we chose OceanSODA-ETHZ, which applies a cluster regression
method to SOCAT underway and remote sensing data to provide 3D global surface carbonate chemistry data for (1982–2022
inclusive) at (1°, monthly) resolution (Gregor and Gruber, 2021). Native mass-specific units were converted using the
accompanying surface temperature and salinity data, and bilinear interpolation was applied over (longitude, latitude, time) to
match with observations, with augmentation at longitude/latitude limits to avoid data loss.

As biogeochemical reanalysis products, using model-data assimilation in the classical sense, we chose the monthly Arctic
TOPAZ-ECOSMO products from CMEMS/Copernicus (Wakamatsu et al., 2022a, b), based on the HYCOM-EVP-ECOSMO
biogeochemical model (Yumruktepe et al., 2022) and joint model parameter-state estimation by sequential data assimilation
(a variant of the Ensemble Kalman Filter, Sakov et al., 2012; Simon et al., 2015). This is one of the few biogeochemical
reanalysis products that assimilates in situ (nutrients, $O_2$) observations as well as satellite chlorophyll, and data were extracted
for (>60° N, 2007–2021 inclusive) at (25 km, 1 month) resolution. Reanalysis data were matched to observations by first
converting observed (latitude, longitude) pairs to the regular (x,y) model coordinates using the native polar stereographic
projection, then interpolating model output bilinearly over (x, y, depth, time) to the observed coordinates. Tests using
horizontal nearest grid locations for all matchups made <0.4 % difference to matchup RMSE values.

We also considered data covering (>40° N, 1993–2019) at (0.25°, 1 month) resolution from the CMEMS Global Ocean
Biogeochemical Hindcast (GOBH) products, derived from a global NEMO-PISCES hindcast (Perruche, 2019). Though not
reanalysis products, these datasets are often considered a benchmark and have been used as reference datasets for bias-
correction and statistical downscaling of CMIP6 models (Kristiansen et al., 2024). Bilinear interpolation over (longitude,
latitude, depth, time) was applied to match with observations, with augmentation to avoid data loss.

**2.14 An additional validation dataset: The IMR Barents Sea observations**

After finalizing our products we became aware of another high-quality in situ observational dataset for (nutrients, $O_2$) in the
Barents Sea, based on bottle samples from cruises by the Institute of Marine Research during (1990–2019) (Gundersen et al.,
2022). These will be included in future versions of NORBGC and NorESM-bcoi; for the present study we use them as
additional validation data, having confirmed that none of these data were included in the datasets contributing to NORBGCv1
(Table 2). Since these data share no common sampled locations with NORBGCv1 and cover the Barents Sea approximately
uniformly (though with some seasonal bias), we consider that the skill of our products against this dataset approximately
measures their interpolative accuracy within the Barents Sea.



# 3. Results and discussion

## 3.1 Consistency with observational data and uncertainty relative to standard climatological estimates

Table 3 shows the Root-Mean-Square-Error (RMSE) for various products over test data subsets of quality-controlled in situ observations from NORBGCv1, showing also results further restricted to the GLODAP/CARINA source as a gold standard, and to different spatial subsets. Fig. A3 shows matchup scatterplots and other skill metrics for the Northern Seas subsets (first section of Table 3). We first compare the RMSE of the original model hindcast (NorESM-OCv1.2) with that of the bias-

corrected hindcast based only on the training subsets (NorESM-OCv1.2bc(tr)) and of the full reanalysis product based only on the training subsets (NorESM-bcoi-v1(tr)). Here the RMSE values measure interpolative accuracy, since the test subset locations include none of the training subset locations, and are selected to require a level of interpolation that is consistent with application of the final product (see Figs. 4, 5). Table 3 shows that the reanalysis achieves a large improvement in interpolative accuracy relative to the original hindcast, for all variables and all spatial domains, and most of the improvement is obtained by

the bias correction step (see NorESM-OCv1.2bc(tr)), with a small further improvement obtained by anomaly kriging (see NorESM-bcoi-v1(tr)). RMSE is generally reduced by a factor 2–6 relative to the original hindcast.

Comparing with the WOA23 climatological products for nutrients and dissolved oxygen, NorESM-bcoi-v1(tr) achieves similar RMSE, usually slightly lower (14 out of 20 comparisons in Table 3). Considering the full test subset (first section of Table 3), the RMSE of NorESM-bcoi-v1(tr) is 4–6% lower for nitrate and phosphate, ~1% higher for silicate, and 19% lower for

dissolved oxygen. Since the test subset data were excluded from the NorESM-bcoi-v1(tr) development, but were likely mostly included in the WOA23 product development, these comparisons favour WOA23 and should give conservative estimates of the improvement in interpolative accuracy achieved by NorESM-bcoi-v1. The final product based on training+test datasets (NorESM-bcoi-v1) achieves lower RMSE than WOA23 in all cases (19–42% lower), showing that it is more consistent with the observational data at the sampled times and locations.

Comparing with the GLODAP gridded climatological products (Lauvset et al., 2016, hereafter L16) for dissolved inorganic carbon (DIC) and total alkalinity (TA), the training subset product NorESM-bcoi-v1(tr) achieves similar RMSE (within 20%) in all test subsets. Over the full test subset, the RMSE of NorESM-bcoi-v1(tr) is 10% lower for DIC and 1% lower for TA. Again, since the test subset data were likely mostly included in the L16 product development, these are conservative estimates of the improvement in interpolative accuracy achieved by NorESM-bcoi-v1. The final product NorESM-bcoi-v1 is again

closer to the observations in all cases (1–35% lower RMSE).

**Table 3. Consistency of model and climatological products with observational data, as measured by Root Mean Square Error (RMSE) between interpolated product values and quality-controlled in situ observational data from test subsets of the NORBGCv1 compilations. Columns show results for nitrate-plus-nitrite ($NO_3+NO_2$), phosphate ($PO_4$), silicate (Si), dissolved oxygen ($O_2$), dissolved inorganic carbon (DIC), and total alkalinity (TA), all in in-situ volume specific units (μmol/L). Test subset RMSE values**

**are shown for the original model hindcast (NorESM-OCv1.2), the hindcast after bias correction using only training subset data (NorESM-OCv1.2bc(tr)), the hindcast after bias correction and anomaly kriging using only training subset data (NorESM-bcoi-v1(tr)), the World Ocean Atlas 2023 climatologies (WOA23) for (nutrients, $O_2$) and the GLODAP gridded product (Lauvset et al.,**



**2016) for (DIC, TA) (see italic font), and for the final reanalysis based on the full training+test observational datasets (NorESM-bcoi-v1). Different subsections show results for different subsets of the test data, based on spatial restrictions or source restriction (GLODAP/CARINA ONLY), and the first row of each subsection shows the number of product-observation matchups for each variable, applicable to all products (the same observation subsets are used for all products).**

| | NO₃+NO₂ [µM] | PO₄ [µM] | Si [µM] | O₂ [µM] | DIC [µM] | TA [µM] |
|---|---|---|---|---|---|---|
| **Northern Seas (test data within >40° N, years 1980–2020 inclusive, ≥40 km from coast)** | | | | | | |
| no. matchups | 43479 | 50963 | 42700 | 171812 | 6081 | 5861 |
| NorESM-OCv1.2 | 8.15 | 0.410 | 21.86 | 62.2 | 87.0 | 45.3 |
| NorESM-OCv1.2bc(tr) | 2.41 | 0.174 | 6.26 | 16.8 | 22.5 | 18.9 |
| NorESM-bcoi-v1(tr) | 2.22 | 0.158 | 5.49 | 14.3 | 22.2 | 18.7 |
| WOA23, *Lauvset2016* | 2.35 | 0.165 | 5.46 | 17.6 | *25.0* | *18.9* |
| NorESM-bcoi-v1 | 1.60 | 0.119 | 3.45 | 10.9 | 17.8 | 17.0 |
| **Northern Seas (test data within >40° N, years 1980–2020 inclusive, ≥40 km from coast), GLODAP/CARINA ONLY** | | | | | | |
| no. matchups | 26588 | 20835 | 20234 | 34080 | 6014 | 5795 |
| NorESM-OCv1.2 | 8.54 | 0.451 | 21.68 | 78.1 | 87.4 | 45.5 |
| NorESM-OCv1.2bc(tr) | 1.95 | 0.138 | 5.71 | 16.3 | 22.6 | 18.9 |
| NorESM-bcoi-v1(tr) | 1.86 | 0.127 | 5.29 | 14.8 | 22.3 | 18.7 |
| WOA23, *Lauvset2016* | 1.91 | 0.129 | 4.33 | 14.1 | *25.0* | *18.8* |
| NorESM-bcoi-v1 | 1.20 | 0.085 | 2.79 | 9.4 | 17.9 | 17.1 |
| **Arctic Seas (test data within >65° N, years 1980–2020 inclusive, ≥40 km from coast)** | | | | | | |
| no. matchups | 9677 | 11140 | 9378 | 31087 | 562 | 543 |
| NorESM-OCv1.2 | 4.69 | 0.271 | 15.58 | 35.0 | 73.3 | 64.7 |
| NorESM-OCv1.2bc(tr) | 1.62 | 0.153 | 3.44 | 13.4 | 49.4 | 53.9 |
| NorESM-bcoi-v1(tr) | 1.42 | 0.135 | 2.91 | 11.5 | 48.5 | 53.3 |
| WOA23, *Lauvset2016* | 1.68 | 0.154 | 3.28 | 13.5 | *59.5* | *55.4* |
| NorESM-bcoi-v1 | 1.12 | 0.101 | 2.10 | 9.0 | 44.4 | 50.5 |
| **North Atlantic (test data within 40–65° N, 80° W to 10° E, years 1980–2020 inclusive, ≥40 km from coast)** | | | | | | |
| no. matchups | 8271 | 17918 | 11295 | 64388 | 1337 | 1314 |
| NorESM-OCv1.2 | 3.82 | 0.259 | 9.19 | 30.5 | 36.9 | 26.8 |
| NorESM-OCv1.2bc(tr) | 1.62 | 0.135 | 1.71 | 15.2 | 14.4 | 12.9 |
| NorESM-bcoi-v1(tr) | 1.45 | 0.118 | 1.47 | 12.4 | 14.0 | 12.8 |
| WOA23, *Lauvset2016* | 1.59 | 0.131 | 1.44 | 15.9 | *15.4* | *11.2* |
| NorESM-bcoi-v1 | 1.11 | 0.097 | 1.16 | 10.6 | 12.0 | 11.1 |
| **North Pacific (test data within 40–65° N, 130° E to 120° W, years 1980–2020 inclusive, ≥40 km from coast)** | | | | | | |
| no. matchups | 25501 | 22718 | 22391 | 76427 | 4182 | 4004 |
| NorESM-OCv1.2 | 10.01 | 0.538 | 27.73 | 86.1 | 99.3 | 47.0 |
| NorESM-OCv1.2bc(tr) | 2.83 | 0.206 | 8.27 | 19.2 | 18.5 | 8.7 |
| NorESM-bcoi-v1(tr) | 2.63 | 0.191 | 7.27 | 16.5 | 18.4 | 8.5 |
| WOA23, *Lauvset2016* | 2.74 | 0.190 | 7.17 | 20.2 | *18.9* | *7.9* |





| NorESM-bcoi-v1 | 1.86 | 0.140 | 4.50 | 11.8 | 12.3 | 6.1 |

## 3.2 Time series comparisons with observations and standard climatological estimates

Here we show comparisons for some of the best-sampled observational time series stations in >40° N region. Figure 6 shows
estimates at 10 and 600 m depth at the Iceland Sea station in the North Atlantic (68° N, 12.67° W), comparing with all in situ
observations from NORBGCv1 within 35 km distance of the nominal location. The NorESM-bcoi-v1 reanalysis captures
much of the observed interannual variability, such as the gradual decrease in surface silicate concentrations during 1980–2000
(Fig. 6c), or the variations in deep oxygen concentrations (Fig. 6j), which is by definition absent in the climatological products.
It also captures well the seasonality and long term trends in the (DIC, TA) data, which are absent in the annual GLODAP
climatologies (Fig. 6e,f,k,l). The reanalysis seasonality is somewhat smoother than the WOA23 seasonality, which in some
cases seems to be affected by limited monthly coverage and extreme values beyond the 1980–2020 timeframe, or perhaps
filtered by our secondary QC (Fig. 6h,i). The net result is that the RMSE values, relative to the observations shown, are
generally lower for the NorESM-bcoi-v1 product – the one exception being for 10 m total alkalinity (Fig. 6f), where the salinity
data from SODA reanalysis (Fig. A4) drives some instability in the back-transform (see Sect. 2.10). Similar results, but without
the back-transform instability problem, are obtained for the Irminger Sea station on the south side of Iceland (Fig. A5).


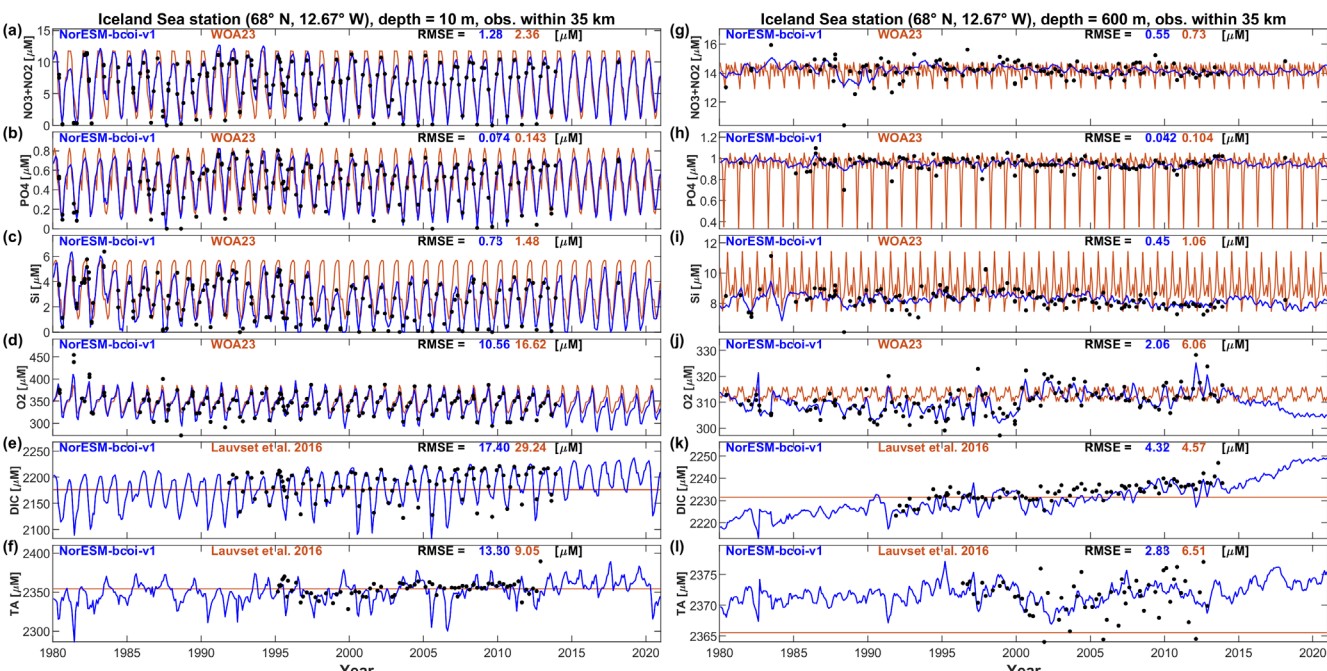

**Figure 6. Iceland Sea station time series comparison, showing monthly estimates from the NorESM-bcoi-v1 product (blue lines).
the World Ocean Atlas 2023 or GLODAPv2-gridded (Lauvset et al., 2016) climatologies (orange lines), and all in situ observations
within 35 km of the nominal station location (68° N, 12.67° W) (black dots). Rows from top to bottom show data for nitrate-plus-
nitrite (NO3+NO2), phosphate (PO4), silicate (Si), dissolved oxygen (O2), dissolved inorganic carbon (DIC), and total alkalinity**

Earth System
Science
Data

(TA), all in in-situ volume specific units (µmol L$^{-1}$). Columns show data at 10m depth (left) and 600m depth (right), applying linear vertical interpolation to product and observational profiles where necessary. Each subplot shows Root Mean Square Error (RMSE) values relative to observational values for the NorESM-bcoi-v1 product (blue) and the WOA23 or Lauvset et al. 2016 climatological product (orange).

Figure 7 shows results for a similarly well-sampled station in the Northwest Pacific at (50° N, 145° W). Again the reanalysis clearly captures more of the interannual variability, and seasonal variability for (DIC, TA), relative to the WOA/GLODAP climatologies, and again the WOA23 estimates show a lack of temporal stability/smoothness (Fig. 7c,g,h,i). In this case the RMSE values, relative to nearby observations, are 12–80% smaller for the reanalysis estimates. Here the SODA reanalysis is performing well at both depths (Fig. A6) and there is no instability issue with back-transformed TA (Fig. 7f,l).


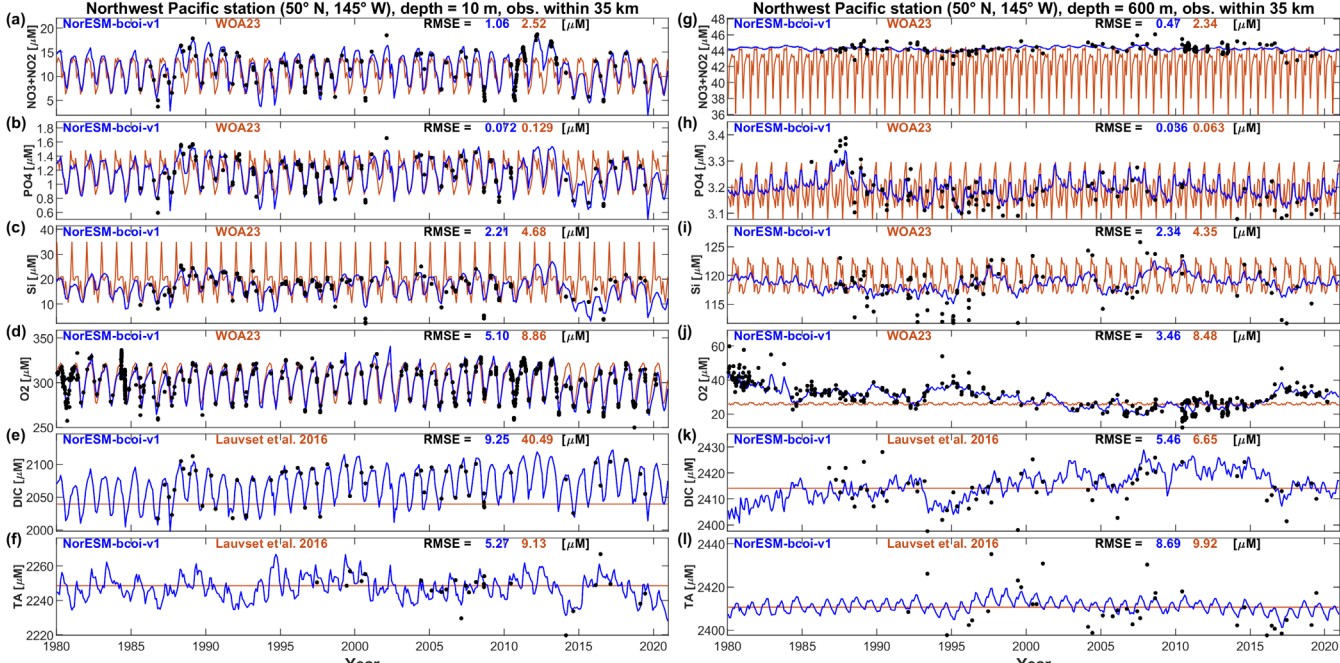

**Figure 7. Northwest Pacific station time series comparison, showing monthly estimates from the NorESM-bcoi-v1 product (blue lines). the World Ocean Atlas 2023 or GLODAP gridded (Lauvset et al., 2016) climatologies (orange lines), and all in situ observations within 35 km of the nominal station location (50° N, 145° W) (black dots). Rows from top to bottom show data for**
**nitrate-plus-nitrite (NO3+NO2), phosphate (PO4), silicate (Si), dissolved oxygen (O2), dissolved inorganic carbon (DIC), and total alkalinity (TA), all in in-situ volume specific units (µmol L$^{-1}$). Columns show data at 10m depth (left) and 600m depth (right), applying linear vertical interpolation to product and observational profiles where necessary. Each subplot shows Root Mean Square Error (RMSE) values relative to observational values for the NorESM-bcoi-v1 product (blue) and the WOA23 or Lauvset et al. 2016 climatological product (orange).**

The time series plots recall that summary accuracy metrics such as RMSE are not the only consideration when choosing a suitable product e.g. to provide boundary conditions for a regional model hindcast. Realistic seasonal cycles are potentially important, and the lack of seasonality in the L16 products could lead to biases, especially for DIC, which could penetrate significantly beyond regional model boundaries (Fig. 6e, 7e). Smoothness of the seasonal signal is also desirable, and here the WOA products, which lack temporal smoothing in their development (Garcia et al., 2024), appear to have some issues (Fig.

6h,i; Fig. 7c,g,h,i), although these could be addressed by the user with simple treatments. Another concern for boundary condition data is the interannual variations and long-term trends, which are lacking in the WOA23/L16 products, but well reproduced by the NorESM-bcoi-v1 products, at least in well-sampled regions (Fig. 6, 7, A5). Note that interannual variability usually contributes only a small component of the total signal variance over space and time, and is therefore not well assessed by summary accuracy metrics, yet it often forms an important part of the results of regional analyses and model simulations.

**3.3 Climatology comparisons with standard products**

Figure 8 compares nitrate-plus-nitrite climatologies computed from the uncorrected hindcast (NorESM-OCv1.2), the final reanalysis product (NorESM-bcoi-v1), and the WOA23 objective analysis product, for February and August at 10 m depth. NorESM climatologies were computed for the 30 years 1985–2014 inclusive, which should be comparable with the range 1965–2022 inclusive used by WOA23 (Garcia et al., 2024). Recall that the plots for each product were made using the same
interpolation method with no gap filling, and therefore illustrate the product coverage (non-grey areas) as well as the data values.

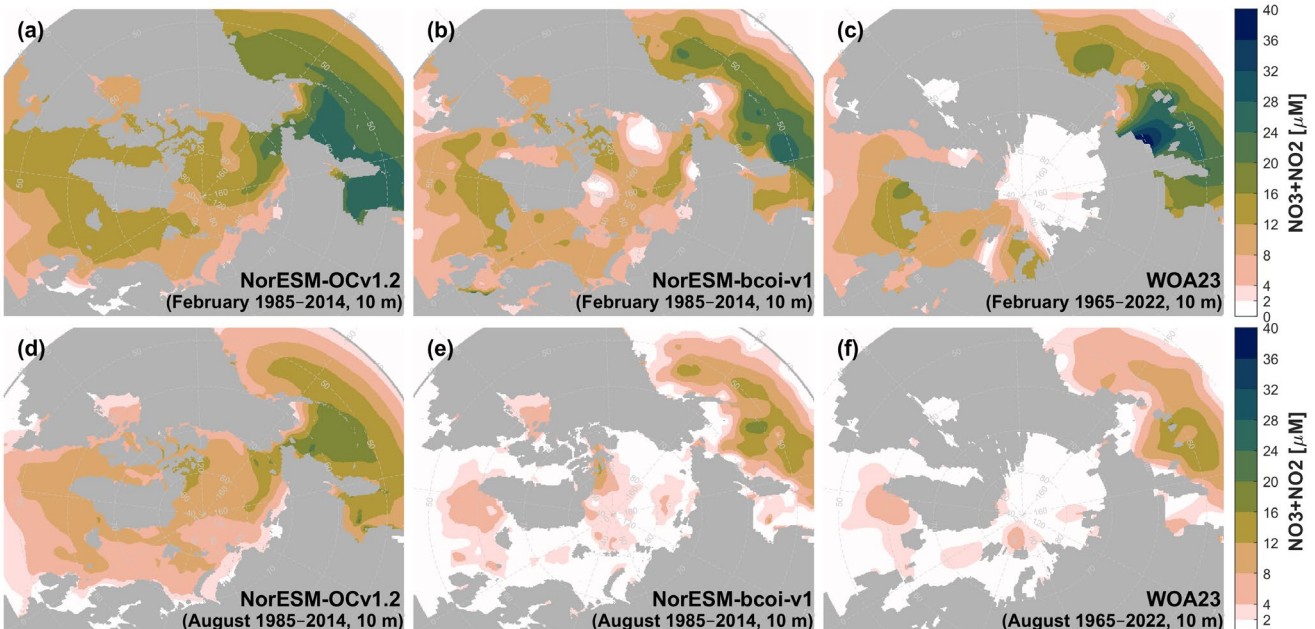

**Figure 8. Climatology comparison for nitrate-plus-nitrite at 10 m depth, for February (upper) and August (lower), showing estimates from the uncorrected hindcast NorESM-OCv1.2 (left), the final reanalysis product NorESM-bcoi-v1 (middle), and the World Ocean**
**Atlas 2023 (right). All concentrations are in in-situ volume-specific units (µmol L⁻¹).**

The reanalysis shows significant departures from the hindcast, especially in August, reflecting the impact of the bias correction step, and to a lesser extent the anomaly kriging (Fig. 8a,b,d,e). Overall, the reanalysis agrees quite well with the WOA23 product, except for two major discrepancies in the February estimates (Fig. 8b,c) (ignoring Hudson Bay, which is beyond the





focus our reanalysis). First, the WOA23 data show a strong hotspot (>36 µM) in the Northeast Bering Sea, off the Siberian
coast, which is absent in the reanalysis product. This feature deserves scrutiny; we found no nearby statistical mean data, or
WOD18 observations during 1965-2022 and within (59–63° N, 170–180° E) with values >30 µmol kg$^{-1}$ at depths 0–10 m (on
observed or standard depth levels). Also, the high WOA monthly values cause a hotspot in the annual data at 0 m depth
(Garcia et al., 2024, their Fig. 3.2) that is not consistent with data from the GLODAP gridded climatology (Lauvset et al.,
2016, their Fig. 5a) or the GND13 climatology based on Certified Reference Materials (Aoyama, 2020). In theory such high
surface values could arise from deep/convective mixing (e.g. down to 500 m), but during February we expect this coastal area
to be ice-covered or near to the sea ice edge (e.g. Langehaug et al., 2013), where meltwater stratification would tend to inhibit
this process.

Second, there is an area of depleted wintertime surface nitrate (<2 µM) extending over most of the Arctic Ocean in the WOA23
data, but this is mostly limited to the Beaufort Sea in the reanalysis (Fig. 8b,c). The low values in the Beaufort Sea are
supported by trusted observations and are attributed to nutrient 'stripping' over long residence times and downwelling driven
by the Beaufort High (Codispoti et al., 2013), as well as denitrification on Arctic shelves (Chang and Devol, 2009; Juranek,
2015). In lieu of wintertime observations, Codispoti et al. (2013) suggest that these conditions may be extrapolated northwards
to the adjacent Amerasian Basin; however, a broader extrapolation seems questionable, especially for the East Siberian shelves
where we expect some influence of northward fluxes through the Bering strait, as observed in the model hindcast (Fig. 8a).
The reanalysis here tends to revert towards the hindcast due to lack of observations and the use of the model hindcast as a prior
(see Sect. 2.8); the result is spatially smoother and arguably more conservative than the WOA23 estimate (Fig. 8b,c), and also
more consistent with results from a pan-Arctic downscaling model (Slagstad et al., 2015, their Fig. 3).

At 200 m depth, the reanalysis and WOA23 are consistent except for depleted February values in the Beaufort Sea in the
WOA23 product (Fig. A7). This latter also deserves scrutiny, since the observations in NORBGCv1 and in Codispoti et al.
(2013) only show depletion down to 50 m, and the WOA23 profiles in this region also show a sudden 'jump' moving to depths
>200 m. At 1000 m depth, the NorESM-bcoi-v1 data are entirely consistent with WOA23 (Fig. A8).

Phosphate climatologies at 10 m depth (Fig. 9) show a milder correction between hindcast and reanalysis, and a closer
consistency between reanalysis and WOA23, except for a hotspot off the Siberian coast in the WOA23 data (Fig. 9c)
corresponding to the nitrate hotspot (Fig. 8c) discussed above. At 200 and 1000 m, the reanalysis shows no important
discrepancies with the WOA23 data (Figs. A9, A10).

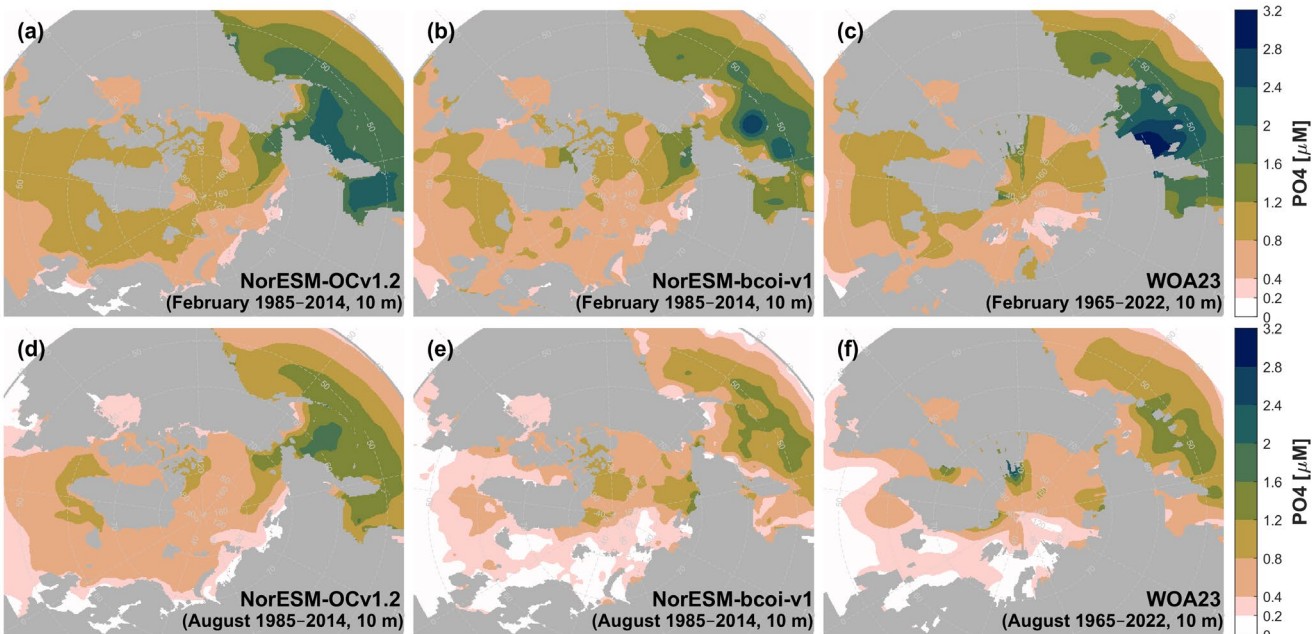

**Figure 9. Climatology comparison for phosphate at 10 m depth, for February (upper) and August (lower), showing estimates from the uncorrected hindcast NorESM-OCv1.2 (left), the final reanalysis product NorESM-bcoi-v1 (middle), and the World Ocean Atlas 2023 (right). All concentrations are in in-situ volume-specific units (μmol L⁻¹).**

For silicate, there are relatively large biases in the model hindcast at 10 m, and these result in large reductions moving from hindcast to reanalysis (Fig. 10a,b,d,e). Aside from the Siberian coast nutrient hotspot discussed above, the main discrepancy between reanalysis and WOA23 is for wintertime high Arctic silicate concentrations, which reach somewhat higher values in the reanalysis (Fig. 10b,c). As for nitrate and phosphate, there is a lack of wintertime observations beyond the Beaufort Sea and north of Greenland, and so the reanalysis tends to revert back towards the hindcast values in these areas. In this case,

however, the high bias in the original hindcast may be driving high bias in the reanalysis, notably the 'tongue' of values 20–40 μM extending from the East Siberian Shelf, although the implied molar ratios to nitrate and phosphate (Fig. 8b, 9b) are still within observed ranges. At 200 and 1000 m depth the reanalysis shows no important discrepancies with the WOA23 data (Figs. A11, A12).

   For dissolved oxygen, we found no significant discrepancies between reanalysis and WOA23 climatologies, while the

reanalysis offers slightly better spatial coverage (Figs. 11, A13, A14).


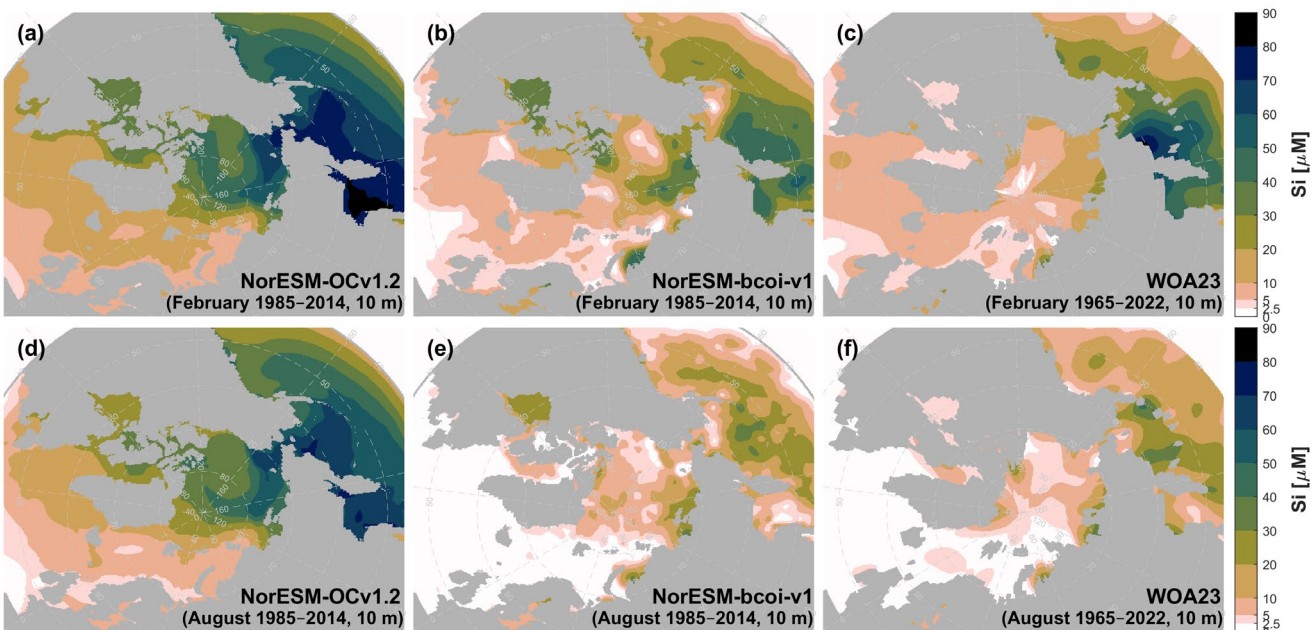

**Figure 10. Climatology comparison for silicate at 10 m depth, for February (upper) and August (lower), showing estimates from the uncorrected hindcast NorESM-OCv1.2 (left), the final reanalysis product NorESM-bcoi-v1 (middle), and the World Ocean Atlas 2023 (right).  All concentrations are in in-situ volume-specific units (μmol L⁻¹).**

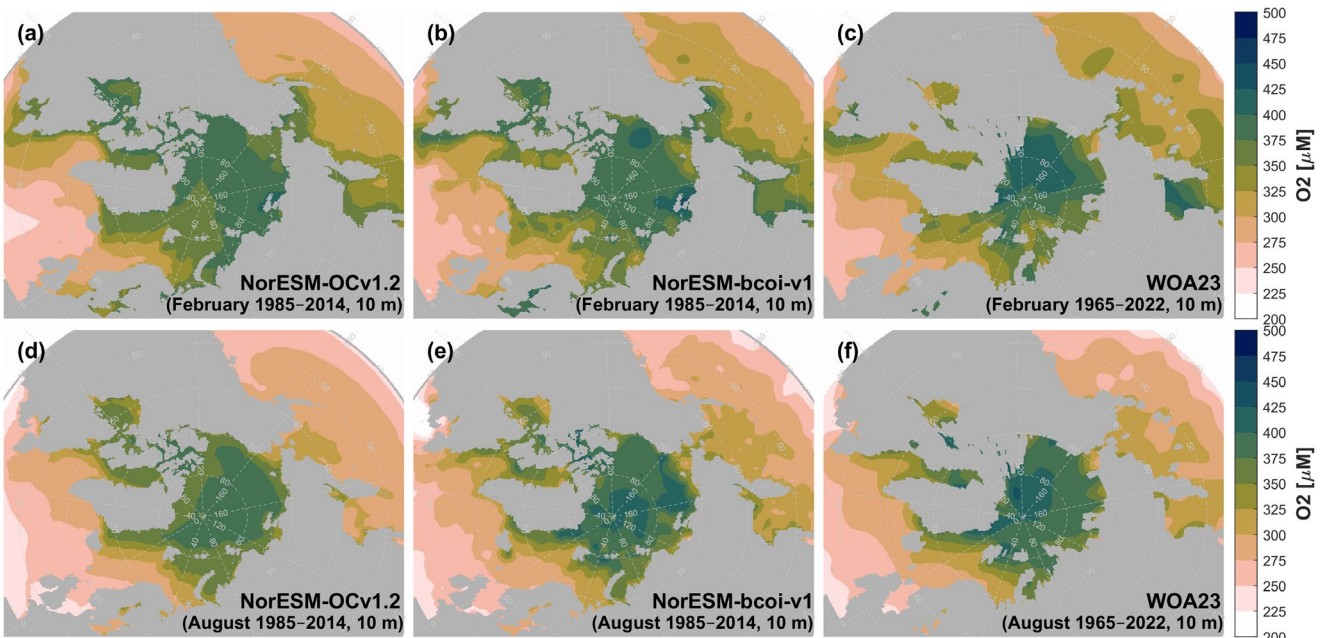


**Figure 11. Climatology comparison for dissolved oxygen at 10 m depth, for February (upper) and August (lower), showing estimates from the uncorrected hindcast NorESM-OCv1.2 (left), the final reanalysis product NorESM-bcoi-v1 (middle), and the World Ocean Atlas 2023 (right).  All concentrations are in in-situ volume-specific units (μmol L⁻¹).**



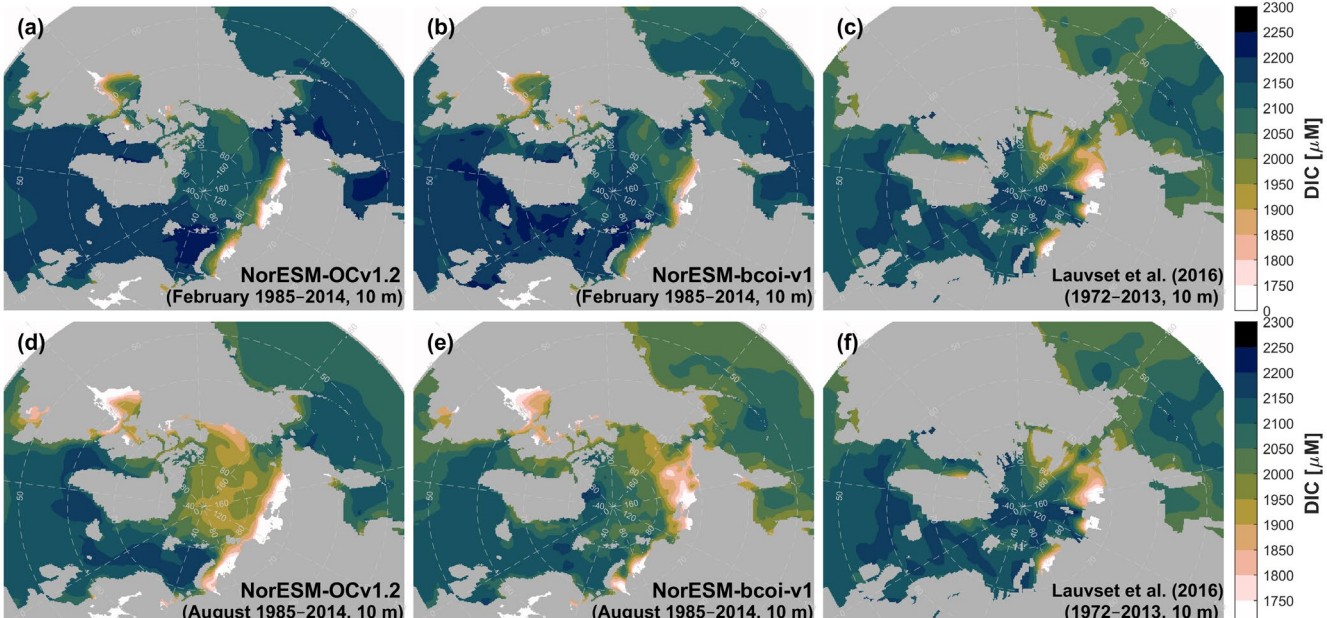

**Figure 12. Climatology comparison for dissolved inorganic carbon at 10 m depth, for February (upper) and August (lower), showing estimates from the uncorrected hindcast NorESM-OCv1.2 (left), the final reanalysis product NorESM-bcoi-v1 (middle), and the GLODAP gridded product (Lauvset et al., 2016) (right). All concentrations are in in-situ volume-specific units (μmol L$^{-1}$).**

For dissolved inorganic carbon and total alkalinity at 10 m depth (Figs. 12, 13), the annual climatologies of Lauvset et al. (2016) (repeated in panels c and f) agree better with the August climatologies from the reanalysis, which show generally lower values than in February due to effects of winter mixing, summertime ice melt, and photosynthetic DIC uptake. This is expected due to the strong seasonal sampling bias, especially in Arctic regions, towards summertime months, with the consequence that the L16 data are likely low-biased estimates of annual average climatologies in Arctic regions (Figs. A15, A16). Note also that the L16 product has missing data in the shallow layers of the Beaufort Sea (Fig. 12c,f). We find no important discrepancies between reanalysis and L16 products at 200 and 1000 m depth (Figs. A17-A20).





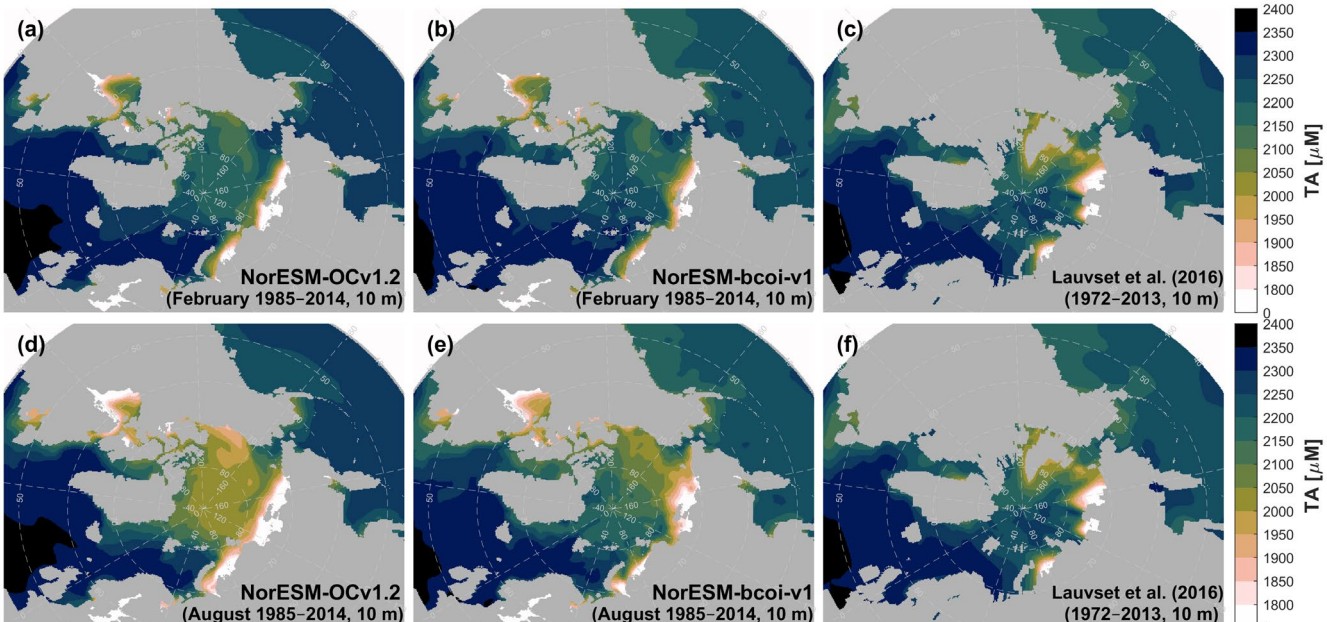

**Figure 13. Climatology comparison for total alkalinity at 10 m depth, for February (upper) and August (lower), showing estimates from the uncorrected hindcast NorESM-OCv1.2 (left), the final reanalysis product NorESM-bcoi-v1 (middle), and the GLODAP gridded product (Lauvset et al., 2016) (right). All concentrations are in in-situ volume-specific units (µmol L$^{-1}$).**

In summary, our analyses suggest that the NorESM-bcoi-v1 products have climatological values that are mostly consistent with the standard WOA23 and L16 products in the >40° N region, and that the few deviations appear to be plausible on the basis of presently available observations. The reanalysis products also show improvements in coverage and detail relative to the 1°-resolution WO23/L16 products.

## 3.4 Comparisons with the GOBAIv2.2 dissolved oxygen product

The GOBAIv2.2 product (Sharp et al., 2023) achieves better consistency with the O$_2$ test subset observations than the WOA23 climatology, but usually not as close as the NorESM-bcoi-v1(tr) product calibrated with only training subset observations (Table 4). The exception here is for the test subset data restricted to only GLODAP observations, where the GOBAIv2.2 product has 15% lower RMSE. However, in this case, the test subset was entirely included in the GOBAIv2.2 calibration dataset (comprising Argo floats and GLODAPv2 bottle samples, Sharp et al., 2023), so the RMSE value is likely low-biased as an estimate of interpolative RMSE. The final product NorESM-bcoi-v1, fitted to all NORBGCv1 observations, achieves better consistency with observations for all test data subsets (26–47% lower RMSE than GOBAIv2.2).

Time series plots for the Irminger Sea station (Fig. 14) suggest that the improved consistency is driven by over-smoothing of the seasonal and interannual variability by the GOBAIv2.2 product (here we compare only with GLODAP data, which were definitely included in the GOBAI calibration, see Sharp et al., 2023). Similar results are observed at the Iceland Sea station



(Fig. A21). At the Northwest Pacific station (Fig. 15), the GOBAI product performs rather better, but NorESM-bcoi-v1 still obtains lower RMSE and is more consistent with observed seasonality and trends (e.g. the recovery of 600 m $O_2$ concentrations since 2014, Fig. 15b).

Maps of 17-year linear temporal trends from annual averages show good agreement at 10 m depth (Fig. 16). Here the lower spatial resolution and coverage of GOBAI in the Northern seas is apparent. Note that since our approach involves temporally-constant bias corrections, plus transient kriging corrections, it is hard for the long term trends to be strongly altered, and the reanalysis trends mostly reflect the uncorrected hindcast trends (Fig. 16a,b). The fact that the GOBAI trends correspond well with the NorESM hindcast trends (Fig. 16a,c), which are produced mechanistically without nudging towards observations,
supports the ability of NorESM to simulate long-term biogeochemical trends. At 200 m there is also good agreement (Fig. A22), though here the reanalysis shows more dipole-like structures in the Pacific, suggestive of trends largely driven by shifts in ocean currents.

**Table 4. Consistency of climatological and model products with observational data for dissolved oxygen, as measured by Root Mean Square Error (RMSE, units µmol L⁻¹, in-situ volume specific) between interpolated product values and quality-controlled in situ**
**observational data for various test data subsets of the NORBGCv1 compilation (described in the first column). All products are matched with the same observational subsets (see number of matchups in the second column). The third column shows the fraction of test subset data belonging to GLODAP or WOD-PFL datasets, and therefore likely included in the GOBAIv2.2 calibration. Columns 4–7 show RMSE values for: the World Ocean Atlas 2023 climatology, the GOBAIv2.2 4D gridded product (Sharp et al., 2023), the NorESM-bcoi-v1(tr) reanalysis product based only on the training data subset (excluding all test subset data), and the**
**final reanalysis product NorESM-bcoi-v1 based on all NORBGCv1 observations.**

| Dissolved Oxygen test data subset | no. matchups | % GLODAP or WOD-PFL | RMSE [µM] | | | |
|---|---|---|---|---|---|---|
| | | | WOA23 | GOBAIv2.2 | NorESM-bcoi-v1(tr) | NorESM-bcoi-v1 |
| Northern Seas (>40° N, ≥40 km from coast, years 2004–2020) | 53793 | 78 | 19.9 | 18.1 | 13.1 | 9.8 |
| Northern Seas GLODAP ONLY | 7904 | 100 | 16.5 | 15.3 | 18.1 | 11.3 |
| Arctic Seas (>65° N, ≥40 km from coast, years 2004–2020) | 2647 | 98 | 10.8 | 7.8 | 6.0 | 5.1 |
| North Atlantic (40–65° N, 80° W–10° E, ≥40 km from coast, years 2004–2020) | 15140 | 90 | 14.4 | 13.0 | 9.6 | 7.9 |
| North Pacific (40–65° N, 130° E–120° W, ≥40 km from coast, years 2004–2020) | 36006 | 71 | 22.3 | 20.3 | 14.6 | 10.8 |

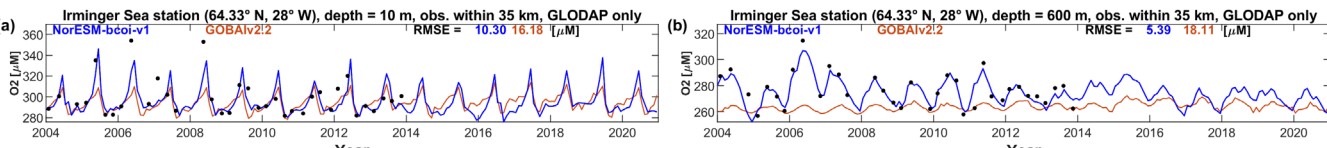

**Figure 14. Irminger Sea station time series comparison, showing monthly estimates of dissolved oxygen concentration (units µmol L⁻¹, in-situ volume specific) from the NorESM-bcoi-v1 product (blue lines) and the GOBAIv2.2 product (Sharp et al., 2023, orange**
**lines), and all GLODAP-sourced observations from the NORBGCv1 dataset within 35 km of the nominal station location (64.33° N, 28° W) (black dots). Columns show data at sampled depths of 10 m (left) and 600 m (right), applying linear vertical interpolation to produce profiles where necessary. Each subplot shows Root Mean Square Error (RMSE) values relative to observational values for the NorESM-bcoi-v1 product (blue) and the GOBAIv2.2 product (orange).**



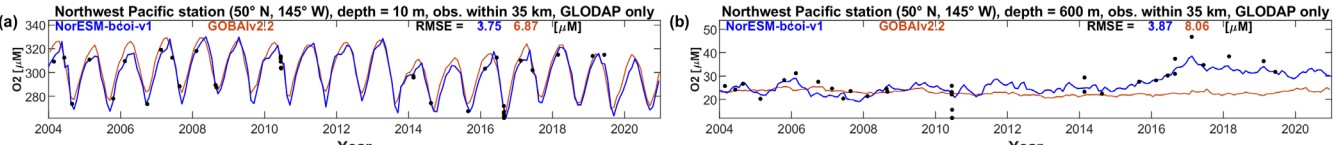

**Figure 15. As in Figure 14, but for the Northwest Pacific station.**

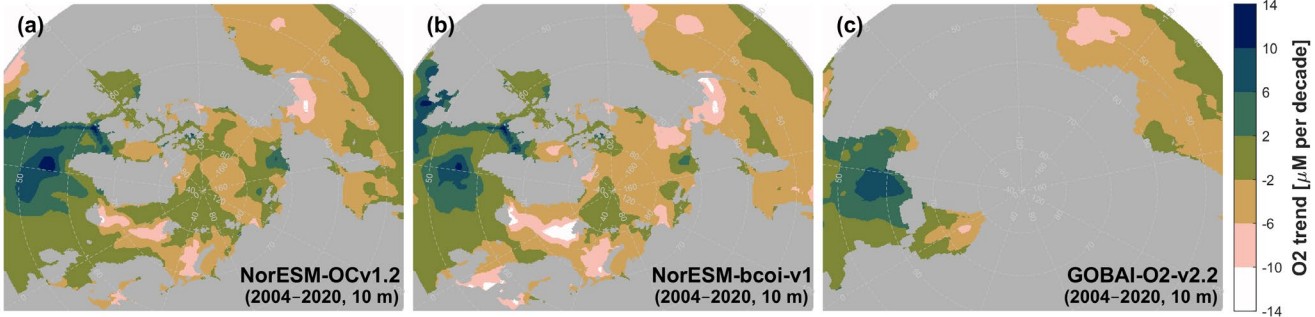

**Figure 16. Dissolved oxygen long term linear trends (17 years, 2004–2020) from annual averages at 10 m depth, showing estimates from the uncorrected hindcast NorESM-OCv1.2 (left), the final reanalysis product NorESM-bcoi-v1 (middle), and the GOBAI-O2-v2.2 gridded product (Sharp et al., 2023) (right). All trends are in in-situ volume-specific units (μmol L⁻¹ per decade).**

## 3.5 Comparisons with the OceanSODA-ETHZ surface dissolved inorganic carbon and total alkalinity products

Compared with the GLODAP gridded climatology, the OceanSODA-ETHZ products (Gregor and Gruber, 2021) show better consistency with observations for surface DIC (shallower than 5 m depth), and generally lower consistency for surface TA (Table 5). Here we use Median Absolute Error (MEDAE) as a robust skill metric, because the matchup sample sizes are relatively low and the data at low salinities (and DIC, TA) are prone to large deviations, such that metrics such as RMSE become strongly dependent on one or two matchups. The reanalysis based only on training subsets, NorESM-bcoi-v1(tr), obtains equal (within 5%) or up to 40% lower MEDAE than OceanSODA-ETHZ, while the final product NorESM-bcoi-v1 has lower MEDAE for all test subsets.

Time series plots for the Irminger Sea and Northwest Pacific stations (Fig. 17) at the shallowest sampled depths (10 and 5 m respectively) show better consistency with observations for the NorESM-bcoi-v1 products (35–51% lower RMSE). At the Iceland Sea station, however, the OceanSODA-ETHZ product shows better consistency (9–25% lower RMSE, see Fig. 18), most likely due to the instability here introduced by the SODA reanalysis salinity data that are used to convert back from normalized variables (see Sect. 2.10 and Fig. A4).

Maps of 30-year linear temporal trends from surface DIC annual averages show good overall agreement between the NorESM-bcoi-v1 and OceanSODA-ETHZ products (Fig. 19b,c). The increasing North Atlantic trends, driven by invasion of anthropogenic $CO_2$, are slightly stronger in NorESM-bcoi-v1, and perhaps more consistent with observed trends (Fig. 17a). In the Arctic region, which is mostly masked out of OceanSODA-ETHZ because wintertime below-ice values are missing, the NorESM trends are dominated by freshening/salinization (Fig. 19a,b). Here we have significant differences between hindcast



and reanalysis because of the use of normalization and (salinity, temperature) reanalysis data from SODA3.4.2 (e.g. the strong
decreasing trend of 30–50 µM per decade in the Beaufort Sea is driven by a freshening trend of 1–1.3 psu per decade in

SODA3.4.2, see Table 1).  Trends in TA show similar patterns, but without the mild increasing trends due to invasion of

anthropogenic $CO_2$ (Fig. A23).

**Table 5.  Consistency of climatological and model products with observational data for surface dissolved inorganic carbon (DIC)
and surface total alkalinity (TA), as measured by Median Absolute Error (MEDAE, units µmol L$^{-1}$, in-situ volume specific) between
interpolated product values and quality-controlled in situ observational data for various test data subsets of the NORBGCv1**

**compilation (described in the first column, with number of product-observation matchups shown in the second column).  Columns
3-6 show results for: the  GLODAPv2-gridded climatology (Lauvset et al., 2016), the OceanSODA-ETHZ monthly product (Gregor
and Gruber, 2021), the NorESM-bcoi-v1(tr) reanalysis product based only on training data subsets (excluding all test subset data),
and the final reanalysis product NorESM-bcoi-v1 based on all available observations.**

| Surface DIC or TA test data subset | no. matchups | MEDAE [µM] | | | |
|---|---|---|---|---|---|
| | | Lauvset et al. 2016 | OceanSODA-ETHZ | NorESM-bcoi-v1(tr) | NorESM-bcoi-v1 |
| DIC Northern Seas (>40°N, ≥40 km from coast, ≤5 m depth, years 1982–2020) | 193 | 16.7 | 12.3 | 11.3 | 9.0 |
| DIC Northern Seas, GLODAP/CARINA ONLY | 183 | 16.4 | 12.3 | 11.0 | 8.7 |
| DIC Arctic Seas (>65°N, ≥40 km from coast, ≤5 m depth, years 1982–2020) | 36 | 42.0 | 26.2 | 20.9 | 16.7 |
| DIC North Atlantic (40–65°N, 80°W–10°E, ≤5 m depth, ≥40 km from coast, years 1982–2020) | 64 | 12.4 | 9.2 | 9.4 | 8.1 |
| DIC North Pacific (40–65°N, 130°E–120°W, ≤5 m depth, ≥40 km from coast, years 1982–2020) | 93 | 16.7 | 11.4 | 10.4 | 7.6 |
| TA Northern Seas (>40°N, ≥40 km from coast, ≤5 m depth, years 1982–2020) | 197 | 8.1 | 10.2 | 7.4 | 5.6 |
| TA Northern Seas, GLODAP/CARINA ONLY | 187 | 7.7 | 9.6 | 7.1 | 5.6 |
| TA Arctic Seas (>65°N, ≥40 km from coast, ≤5 m depth, years 1982–2020) | 38 | 26.5 | 18.1 | 18.8 | 16.1 |
| TA North Atlantic (40–65°N, 80°W–10°E, ≤5 m depth, ≥40 km from coast, years 1982–2020) | 64 | 7.1 | 9.9 | 8.7 | 6.7 |
| TA North Pacific (40–65°N, 130°E–120°W, ≤5 m depth, ≥40 km from coast, years 1982–2020) | 95 | 6.8 | 8.9 | 5.6 | 3.9 |

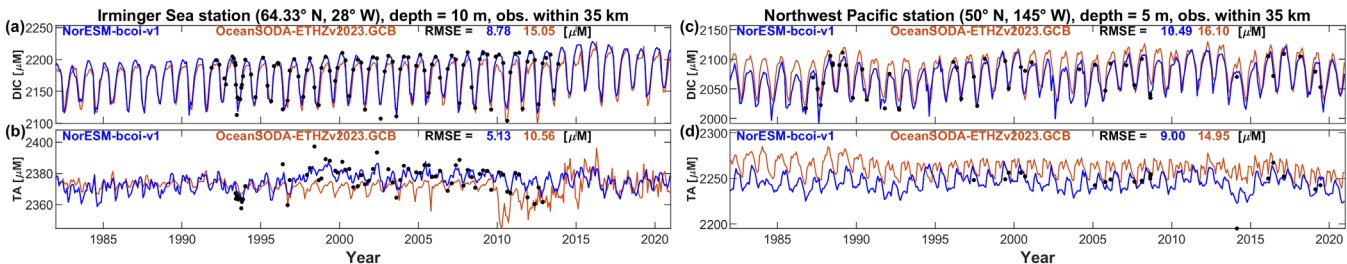


**Figure 17.  Time series comparisons for the Irminger Sea station (left) and Northwest Pacific station (right), at the shallowest sampled
depths (10 and 5 m respectively, concentrations in units µmol L$^{-1}$, in-situ volume specific).  Plots show monthly estimates of dissolved
inorganic carbon (DIC, upper) and total alkalinity (TA, lower) from the NorESM-bcoi-v1 product (blue lines) and the OceanSODA-
ETHZ product (Gregor and Gruber, 2021, orange lines), and all in situ observations from the NORBGCv1 compilation within 35**

**km of the nominal station locations (black dots).  Each subplot shows Root Mean Square Error (RMSE) with respect to observational
values for the NorESM-bcoi-v1 product (blue) and the OceanSODA-ETHZ product (orange).**





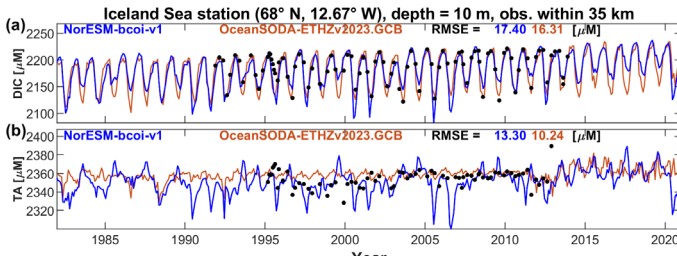

**Figure 18.** As in Figure 17, but for the Iceland Sea station.

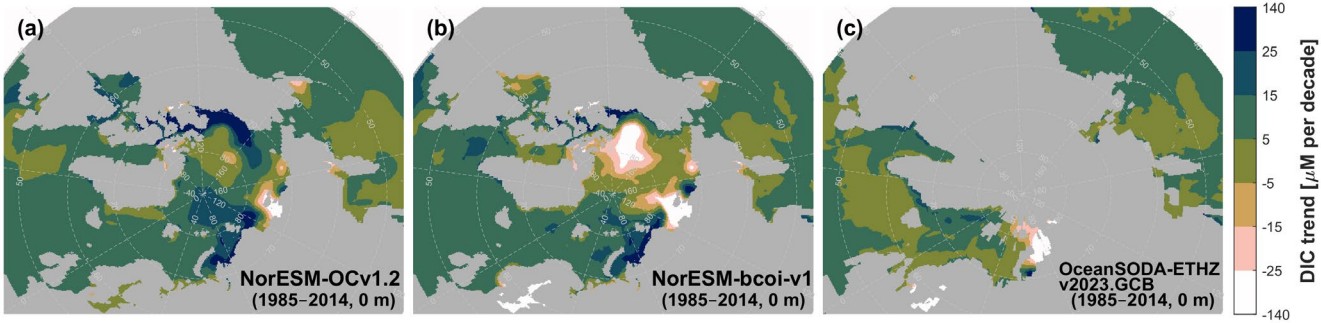

**Figure 19.** Dissolved inorganic carbon long term linear trends (30 years, 1985–2014) from annual averages at 0 m depth, showing estimates from the uncorrected hindcast NorESM-OCv1.2 (left), the final reanalysis product NorESM-bcoi-v1 (middle), and the OceanSODA-ETHZv2023.GCB gridded product (Gregor and Gruber, 2021) (right). All trends are in in-situ volume-specific units (µmol L$^{-1}$ per decade).

### 3.6 Comparisons with CMEMS/Copernicus products

Compared with WOA23 and NorESM-bcoi-v1(tr), the TOPAZ-ECOSMO (Wakamatsu et al., 2022a,b) and GOBH (Perruche, 2019) products are notably less consistent with observations, showing matchup RMSE values roughly twice higher in the >65° N region covered by all products (Table 6). Similar results are obtained using only the gold-standard GLODAPv2 observations (Table C2). The TOPAZ and GOBH products are, however, significantly more consistent with observations than the uncorrected NorESM hindcast data, except for phosphate which is the best-simulated nutrient within NorESM-OCv1.2.

As a further test, we reran the matchup comparisons using the in situ observational datasets from IMR for the Barents Sea (Gundersen et al., 2022) (see Table 7, Fig. A24). None of these quality-controlled data were included in the NORBGCv1 compilations or in the calibration of any of the NorESM-bcoi-v1 reanalysis products. The GOBH products perform remarkably well here, considering they are based on a free-running hindcast, and achieve similar RMSE to the WOA23 products. Still, the NorESM-bcoi-v1 products achieve the lowest RMSE values for all variables.



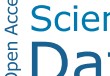

**Table 6. Consistency of model and climatological products with observational data, as measured by Root Mean Square Error (RMSE) between interpolated product values and quality-controlled in situ observational data from Arctic test subsets of the NORBGCv1 compilations. Columns show results for nitrate-plus-nitrite (NO₃+NO₂), phosphate (PO₄), silicate (Si), and dissolved oxygen (O₂), all in in-situ volume specific units (µmol/L). Test subset RMSE values are shown for the World Ocean Atlas 2023 climatologies (WOA23), the CMEMS TOPAZ-ECOSMO biogeochemical reanalysis (Wakamatsu et al., 2022a,b), the CMEMS Global Ocean Biogeochemical Hindcast products (GOBH, Perruche, 2019), the original NorESM model hindcast (NorESM-OCv1.2, Schwinger et al., 2016), and the reanalysis products based only on training subset data (NorESM-bcoi-v1(tr)). Results are shown for matchups within the Arctic region (>65° N, ≥40 km from coast) within the 2007–2019 period covered by all products. The first row shows the number of product-observation matchups for each variable, applicable to all products (the same observation subsets are used for all products).**

|  | NO₃+NO₂ [µM] | PO₄ [µM] | Si [µM] | O₂ [µM] |
|---|---|---|---|---|
| no. matchups | 2911 | 2330 | 2231 | 9395 |
| WOA23 | 1.51 | 0.157 | 3.51 | 12.8 |
| TOPAZ-ECOSMO | 3.52 | 0.384 | 7.45 | 29.8 |
| GOBH | 3.31 | 0.295 | 7.53 | 24.3 |
| NorESM-OCv1.2 | 5.13 | 0.308 | 19.89 | 40.5 |
| NorESM-bcoi-v1(tr) | 1.18 | 0.144 | 3.34 | 11.1 |

**Table 7. Consistency of model and climatological products with observational data, as measured by Root Mean Square Error (RMSE) between interpolated product values and quality-controlled in situ observational data from the IMR Barents Sea datasets (Gundersen et al., 2022). Columns show results for nitrate-plus-nitrite (NO₃+NO₂), phosphate (PO₄), silicate (Si), and dissolved oxygen (O₂), all in in-situ volume specific units (µmol/L). Test subset RMSE values are shown for the World Ocean Atlas 2023 climatologies (WOA23), the CMEMS TOPAZ-ECOSMO biogeochemical reanalysis (Wakamatsu et al., 2022a,b), the CMEMS Global Ocean Biogeochemical Hindcast products (GOBH, Perruche, 2019), the original NorESM model hindcast (NorESM-OCv1.2, Schwinger et al., 2016), and the final reanalysis products (NorESM-bcoi-v1) calibrated to the full NORBGCv1 compilations, which did not include any of the IMR data. Results are shown for matchups within the Barents Sea (70–80° N, 10–60° E, ≥40 km from coast), and for nutrients within the 2007–2019 period covered by all products. For dissolved oxygen the observational coverage (<2007) did not overlap with the TOPAZ-ECOSMO coverage (>2006) so here we consider the period 1993–2006 covered by all other products. The first row shows the number of product-observation matchups for each variable, applicable to all products (the same observation subsets are used for all products).**

|  | NO₃+NO₂ [µM] | PO₄ [µM] | Si [µM] | O₂ [µM] |
|---|---|---|---|---|
| no. matchups | 22943 | 22593 | 23005 | 689 |
| WOA23 | 2.77 | 0.128 | 1.24 | 19.6 |
| TOPAZ-ECOSMO | 3.64 | 0.196 | 2.06 | – |
| GOBH | 2.40 | 0.143 | 1.23 | 19.5 |
| NorESM-OCv1.2 | 4.53 | 0.266 | 6.72 | 18.8 |
| NorESM-bcoi-v1 | 1.80 | 0.111 | 1.04 | 18.0 |





### 3.7 Potential uses, limitations, and improvements of NorESM-bcoi-v1


The original purpose of NorESM-bcoi-v1 was to provide unbiased boundary and initial conditions for regional ocean biogeochemical models at latitudes >40° N. However, there may be other applications, such as to provide environmental input data for ecological and habitat models, to provide reference datasets for bias correction of Earth System Model (ESM) projections, or to compute climatologies for the validation of ESMs.

One limitation of the NorESM-bcoi-v1 products is the spatiotemporal resolution, which following the resolution of NorESM-OCv1.2 is 27–82 km in the horizontal (with higher resolution further north), 5–1000 m in the vertical (higher near the surface), and monthly in time. This may not be enough to resolve important coastal gradients and (sub-)mesoscale variability. That said, we do resolve some coastal gradients in well-observed areas such as the North Sea (Figs. 8–11b,f), while the (DIC, TA) products inherit coastal freshening gradients from the SODA3.4.2 reanalysis (Figs. 12–13b,f). For providing boundary

conditions, we recommend that users consider using higher-resolution physical reanalysis data for their study region, if available, to derive (DIC, TA) conditions at the same resolution using the relationships in Table 1 and interpolating the normalized products (DICn, TAn). Users should always check whether known coastal gradients in their study region are resolved by the boundary conditions, and where possible avoid placing regional model boundaries in areas of strong coastal gradients, or high uncertainty due to limited observational coverage (such as in seasonally or permanently ice-covered areas).

To some degree, lack of detail/resolution in boundary conditions will be 'forgotten' by the model moving towards the interior of the domain, but if there is a strong in-flowing coastal current, then lack of coastal resolution in boundary conditions could be an issue. Sub-monthly and (sub)-mesoscale oceanic variability in (nutrients, $O_2$, DIC, TA) could be locally important, but is generally small compared to basin-scale spatial and seasonal variations (we capture >95% of the observational variance with the present reanalysis products, see correlations ≥0.98 in Fig. A3) and again should be generated internally by the regional

model moving away from the boundaries.

Another obvious limitation is in the spatial coverage (>40° N). This choice was made to limit computational expense, but also the human effort needed to quality-control the underlying observational datasets. For the Mediterranean and Black Seas, product values were masked out because we did not have time to QC the observations, and we expected that existing region-specific products (e.g. from CMEMS) would anyway be superior. The latter also applied to the Baltic Sea, but we have left it

unmasked in our products to allow continuity with data in the Kattegat, and since data density and primary QC level was generally high. Exceptions here include (DIC, TA), for which we had no observational coverage because sources were mostly restricted to GLODAPv2, which excluded Baltic carbonate chemistry data. Also, for Hudson Bay our observational compilations and model products are likely suboptimal and we have excluded it as a region for test data, but left the products unmasked in the interests of continuity.

The products are limited to ($NO_3$+$NO_2$, $PO_4$, Si, $O_2$, DIC, TA) as a set of biogeochemical variables that are likely most important as boundary conditions for regional ocean models, and that also have sufficient observational coverage for bias



correction purposes. Ammonium ($NH_4$) could be important in coastal regions, but in open ocean regions lacks observational coverage, such that a simple low-valued constant value (e.g. 0.1 µM) may suffice on boundary conditions away from coastal sources. Dissolved organic matter (DOM) can also be important in coastal areas, but observational coverage is typically

lacking, and global model datasets either lack DOM variables or have large associated uncertainties. We have found that local statistical models, based on regional observations and e.g. (temperature, salinity) as predictor variables, can provide useful estimates for DOM boundary conditions. In open ocean regions, low-valued constants may suffice if only labile DOM components are included in the regional model. Potentially also particulate variables could be important as boundary conditions, but our experience so far has not indicated any important impacts of such conditions moving to the regional model

interior – suggesting again that low-valued constants may suffice.

The NorESM-bcoi-v1 products for (DIC, TA) are only trained on bottle sample observations for (DIC, TA, $pCO_2$, pH). They do not exploit underway observational data from sensors, in particular for surface $fCO_2/pCO_2$ which is the most numerous observational data type for carbonate chemistry in the ocean (e.g. Gregor and Gruber, 2021). In this sense, our estimates cannot be optimal, and for the surface ocean may therefore be less accurate than estimates that do exploit underway data, at

least at moderate latitudes where ship-of-opportunity traffic is frequent. Nevertheless, the estimated interpolative accuracy of our (DIC, TA) products was similar to that of the corresponding OceanSODA-ETHZ products within the >40 °N regions and subregions (see Table 5, fourth and fifth columns). This may reflect the fact that, as with many products based on surface $pCO_2$–TA submodels, the OceanSODA-ETHZ products did not include bottle sample DIC in training even for the final products, which is also suboptimal. Also, there is some loss of accuracy when the primary ($pCO_2$, TA) estimates are used to

calculate DIC via CO2SYS. We would still recommend a $pCO_2$–TA-based product such as OceanSODA-ETHZ if the user requires surface $pCO_2$, and probably also surface pH since this is most accurately calculated from the ($pCO_2$, TA) pair (Raimondi et al., 2019). However, our products appear to be competitive with $pCO_2$–TA-based products for estimating surface (DIC, TA), while also providing much less frequently-estimated subsurface values. An optimal approach would of course merge information from all data types, and ways to achieve this should be considered in future work.

Another limitation concerns biases in the process-based model hindcast (NorESM-OCv1.2) that are not fully removed in our bias correction step, due to limitations in observational coverage, smoothness constraints, and the use of the model output as a 'prior' to stabilize estimates in poorly-sampled regions (see Sect. 2.8). This may have caused some significant residual biases in the reanalysis products, especially for silicate, due to strong biases in the original hindcast data, and for (DIC, TA), due to poorer observational coverage. Users should apply extra caution with these products in regions with poor or seasonally-

restricted sampling such as the high Arctic, the East Siberian Shelf, and the North Pacific shelves. Future versions of the reanalysis could benefit from reduced bias in the NorESM-OCv1.2 hindcast, or could alternatively employ a different hindcast model; for example the CMEMS GOBH and TOPAZ-ECOSMO products showed usually better skill than NorESM-OCv1.2 (see Tables 6, 7), but are available over shorter time spans (since 1993 and 2007 respectively) that would therefore limit the description of long term trends. For the purpose of ESM validation or skill comparison, the use of NorESM-OC is also



problematic as it may bias the validation data in favour of NorESM models; products based on multiple model hindcasts would provide fairer and likely more robust validation datasets. Similarly if using such reanalyses to bias-correct ESM ensembles, it may be important to vary the underlying hindcast model in order to capture the full uncertainty in projections.

Our reanalysis methodology had also some limitations. We estimated bias by kernel smoothing as a non-parametric method suitable for big data, but there was nevertheless a need to specify several hyperparameter values, and these were chosen from

limited trials rather than systematic optimization. Also, the weightings of model hindcast values as 'prior' estimates were assigned in an ad hoc manner without full consideration of spatiotemporal variation (e.g. from smoothing matchup error magnitude). Our anomaly treatment involved only simple, univariate kriging, which may have missed important information from multivariate cross-correlations, or spatio-temporal structures induced by circulation (e.g. Roach and Bindoff, 2023), although our analyses suggest that improvements in the bias correction step may be more important (Table 3). Nevertheless,

two general concepts may prove useful moving forward: i) to utilize information from mechanistic, process-based biogeochemical models to constrain estimates, particularly in areas/times with poor observational coverage, and to better inform long term (climatic) trends; ii) to have a methodological distinction between temporary anomalies and long term biases, such that data-driven corrections are not only transient as in many traditional reanalysis approaches.

## 4. Code and data availability

The NorESM-bcoi-v1 reanalysis data, the NORBGCv1 observational data compilations, and the code used to create them, are all available from the NorESM-bcoi-v1 repository at https://doi.org/10.5281/zenodo.14525787 (Wallhead et al., 2024).

## 5. Conclusions

We developed 4D reanalysis products for nutrients, dissolved oxygen, and carbonate chemistry variables at northern latitudes (>40° N) by first bias-correcting a process-based model hindcast (NorESM-OCv1.2) then adding anomalies based on a simple

kriging approach. These products showed improved interpolative accuracy relative to standard climatologies (WOA23 and GLODAPv2 gridded) while also providing smooth estimates of seasonal variability and long term trends. Climatologies computed from the reanalysis products were mostly consistent with the standard climatologies; significant deviations were restricted to poorly-sampled regions (e.g. the high Arctic, East Siberian and northern Pacific shelves) and were generally considered plausible, although we cannot rule out significant model-derived biases in such regions (especially during winter,

and for silicate and carbonate chemistry variables). The NorESM-bcoi-v1 products also showed equal or better interpolative accuracy in comparison with existing purely-data-driven/AI-based products, and with biogeochemical reanalysis/hindcast products from the CMEMS/Copernicus database. They also showed some significant improvements in spatial coverage and detail resolution at latitudes >40° N compared to existing alternatives.

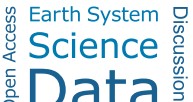

The NorESM-bcoi-v1 products were developed primarily to provide boundary/initial conditions for regional ocean biogeochemical models, but they may also prove useful for other purposes, e.g. as inputs for other models, or to provide climatologies for model validation and skill assessment. The main limitations here are that coastal regions may not be sufficiently resolved, and the use of NorESM-OC output as basis data may bias skill assessments in favour of NorESM models, particularly in poorly sampled regions/seasons where the basis data still exert significant influence. We also provide the quality-controlled compilations of in situ observations at >40° N (NORBGCv1) that were used to develop the reanalysis

products – these also include indicator variables that will allow users to replicate our training and test subsets, and thereby directly compare the efficacy of other approaches with ours.






Earth System
Open Access Science
Discussions Data

# Appendix A: Supplementary Figures

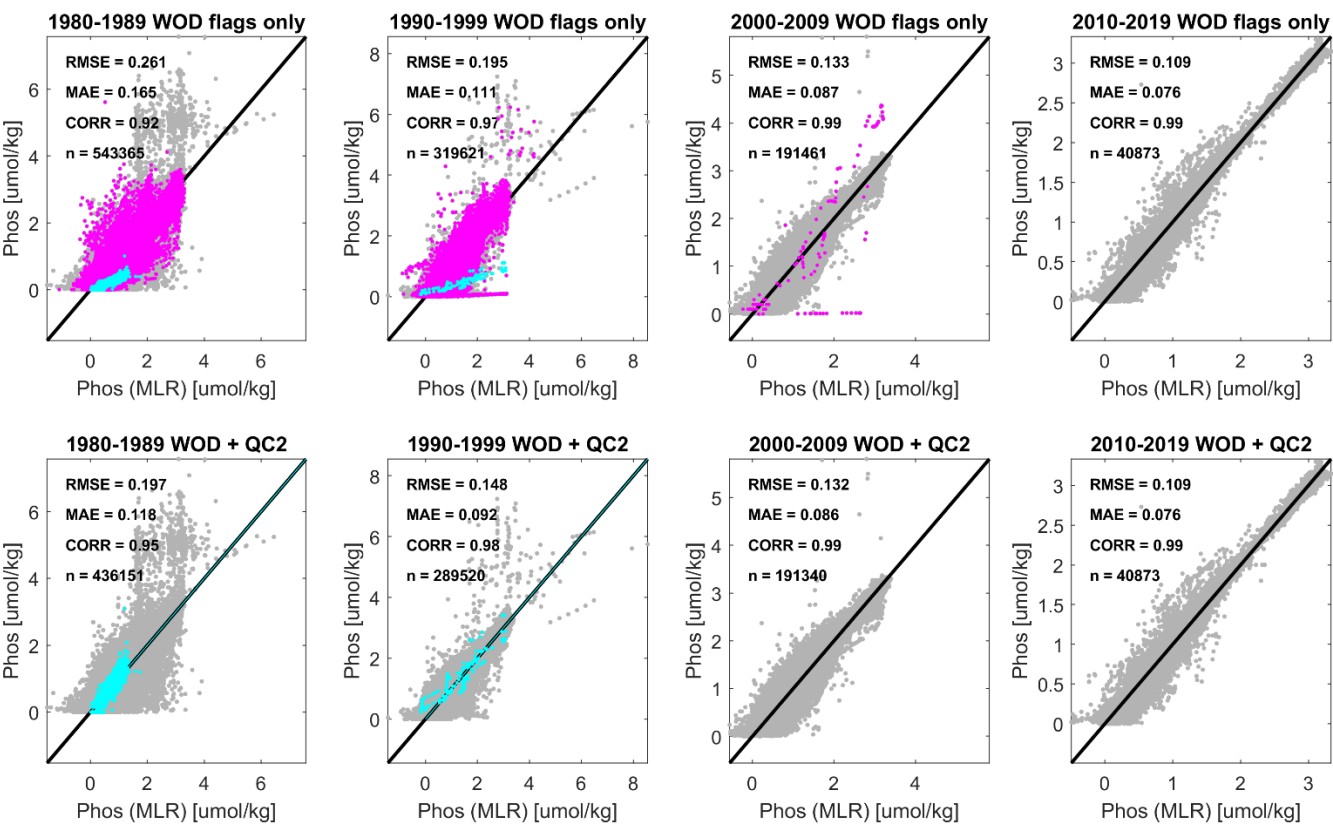

**Figure A1.** Secondary quality control (QC) comparison of phosphate concentrations from the World Ocean Database (2018) bottle
sample dataset (OSD), showing predicted concentrations from regionalized Multiple Linear Regression (MLR) models (functions
of salinity, temperature, and dissolved oxygen, x-axes) with corresponding observed values (y axes), both in mass-specific units (µmol
kg⁻¹), for different decades (left to right subplots), and before/after applying the secondary QC exclusions and corrections
(upper/lower subplots). Magenta color denotes suspect data based on secondary QC, and cyan denotes data that are corrected for
units confusion by the secondary QC. All data shown have passed the primary QC from WOD18 (flag value = 0, 'accepted' values).
Black lines are 1:1, and skill metrics measuring agreement level between observed and modelled values include: Root Mean Square
Errors (RMSE), Mean Absolute Errors (MAE), Pearson correlations (CORR), and the number of model-observation pairs (n).






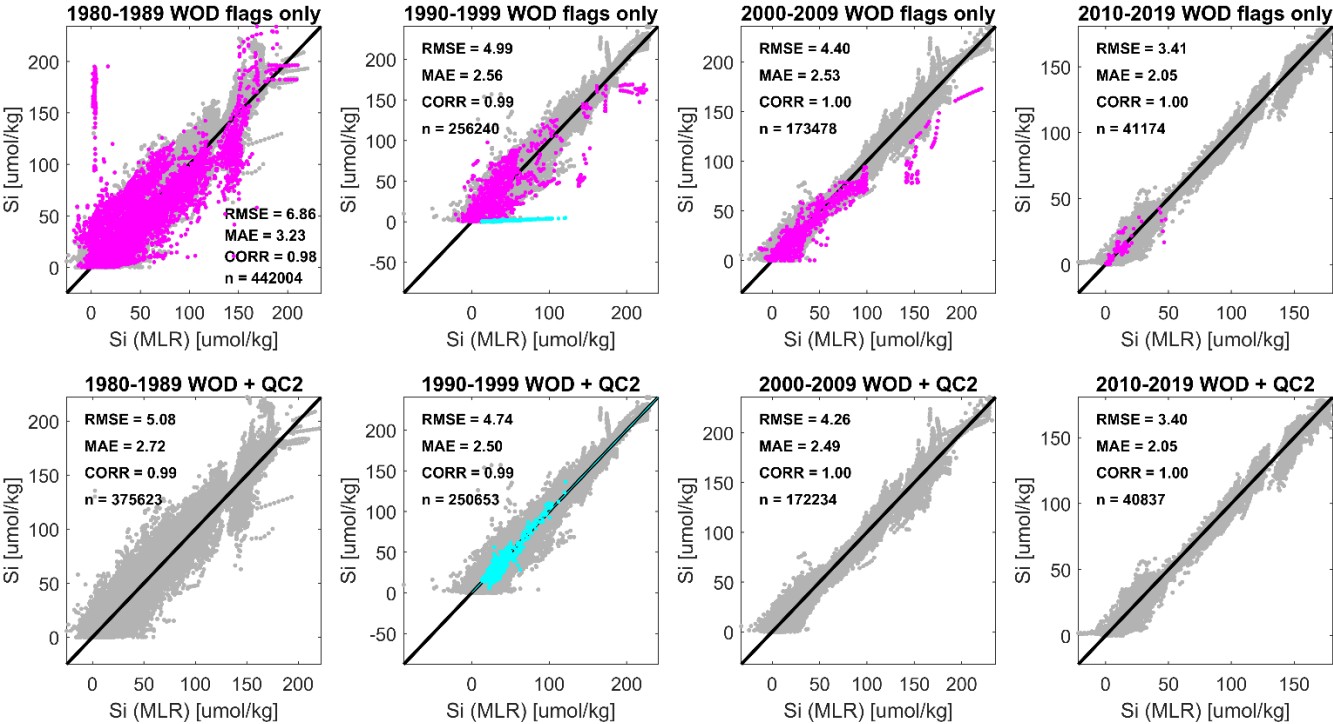

**Figure A2. Secondary quality control (QC) comparison of silicate concentrations from the World Ocean Database (2018) bottle sample dataset (OSD), showing predicted concentrations from regionalized Multiple Linear Regression (MLR) models (functions**
**of salinity, temperature, and dissolved oxygen, x-axes) with corresponding observed values (y axes), both in mass-specific units (µmol kg⁻¹), for different decades (left to right subplots), and before/after applying the secondary QC exclusions and corrections (upper/lower subplots). Magenta color denotes suspect data based on secondary QC, and cyan denotes data that are corrected for units confusion by the secondary QC. All data shown have passed the primary QC from WOD18 (flag value = 0, 'accepted' values). Black lines are 1:1, and skill metrics measuring agreement level between observed and modelled values include: Root Mean Square**
**Errors (RMSE), Mean Absolute Errors (MAE), Pearson correlations (CORR), and the number of model-observation pairs (n).**



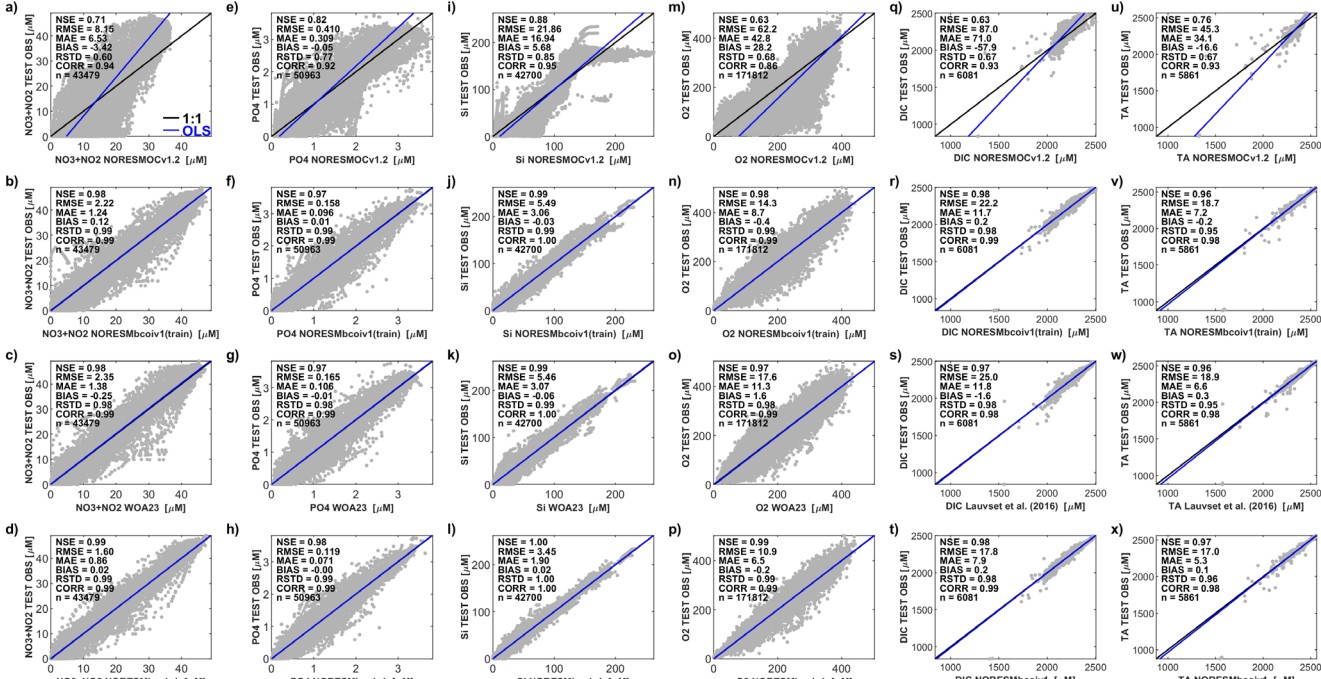

**Figure A3. Matchup scatterplots for the full Northern Seas test data subset (>40° N, 1980–2020. Columns from left to right show results for:** nitrate-plus-nitrite (NO3+NO2), phosphate (PO4), silicate (Si), dissolved oxygen (O2), dissolved inorganic carbon, and total alkalinity, all in in-situ volume-specific units (μmol L$^{-1}$). Rows from top to bottom show results for the uncorrected model hindcast (NorESM-OCv1.2), the reanalysis based only on training subset data (NorESM-bcoi-v1(tr)), the standard climatological product (either the World Ocean Atlas 2023, WOA23, or the GLODAP gridded product from Lauvset et al., 2016), and the final reanalysis fitted to training+test data (NorESM-bcoi-v1). Inset skill metrics show the Nash-Sutcliffe efficiency (NSE), the Root Mean Square Error (RMSE), the Mean Absolute Error (MAE), the mean of model-minus-observation residuals (BIAS), the ratio of modelled to observed standard deviation (RSTD), the Pearson correlation between observations and interpolated model values (CORR), and the total number of matchup pairs (n).

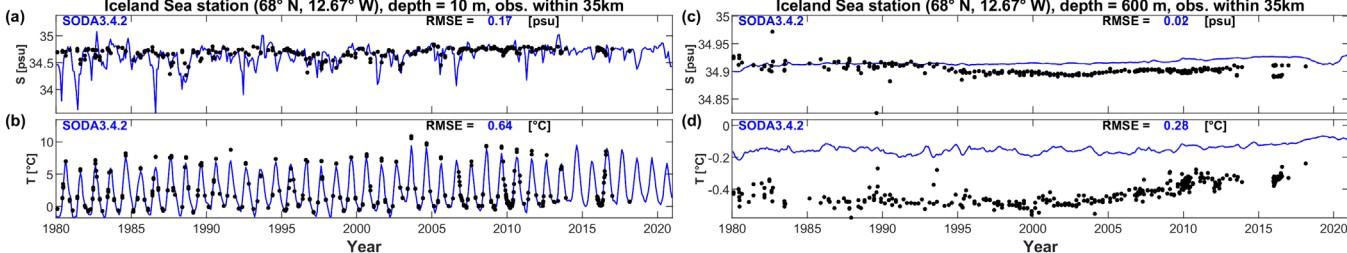

**Figure A4. Iceland Sea station time series comparison, showing monthly estimates from the SODA3.4.2 reanalysis product (blue lines), and all in situ observations within 35 km of the nominal station location (68° N, 12.67° W) (black dots). Subplots show results for salinity (S, upper) and temperature (T, lower) at 10 m depth (left) and 600 m depth (right), applying linear vertical interpolation to product and observational profiles where necessary. Each subplot shows Root Mean Square Error (RMSE) values relative to the observational values shown.**

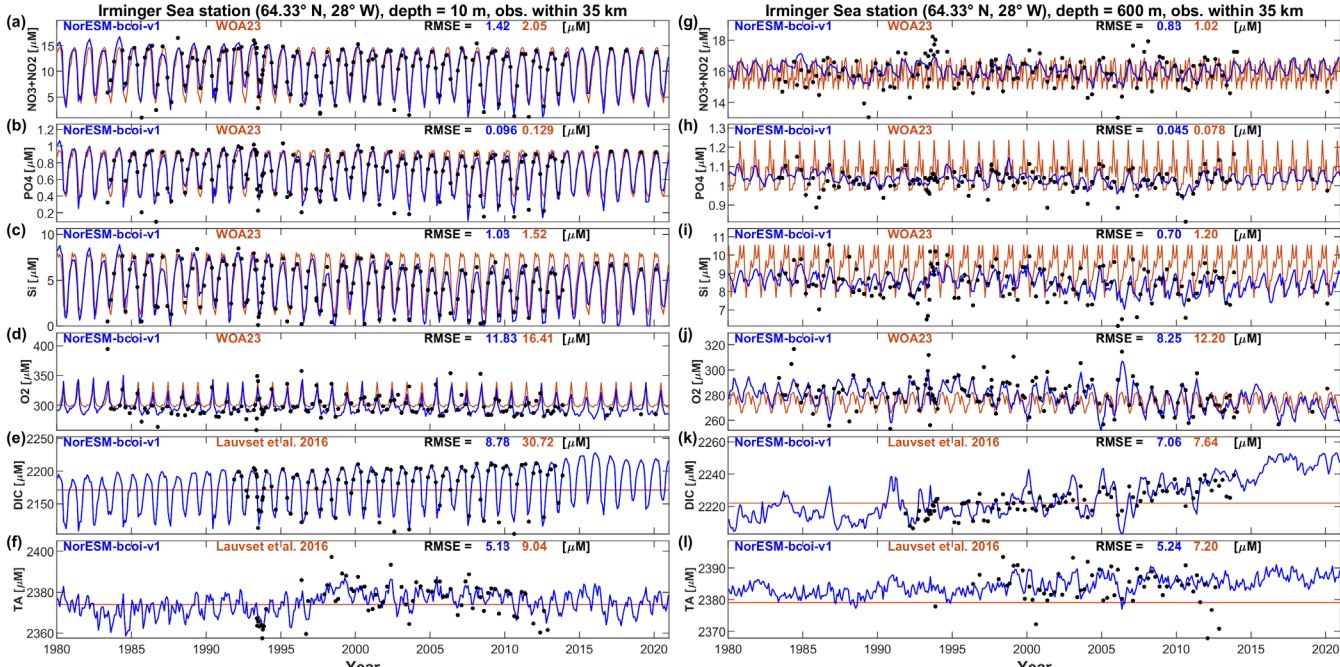

**Figure A5.** Irminger Sea station time series comparison, showing monthly estimates from the NorESM-bcoi-v1 product (blue lines). the World Ocean Atlas 2023 or GLODAPv2-gridded (Lauvset et al., 2016) climatologies (orange lines), and all in situ observations within 35 km of the nominal station location (64.33° N, 28° W) (black dots). Rows from top to bottom show data for nitrate-plus-nitrite (NO3+NO2), phosphate (PO4), silicate (Si), dissolved oxygen (O2), dissolved inorganic carbon (DIC), and total alkalinity (TA), all in in-situ volume specific units (µmol L⁻¹). Columns show data at 10 m depth (left) and 600 m depth (right), applying linear vertical interpolation to product and observational profiles where necessary. Each subplot shows Root Mean Square Error (RMSE) values relative to observational values for the NorESM-bcoi-v1 product (blue) and the WOA23 or Lauvset et al. 2016 climatological product (orange).

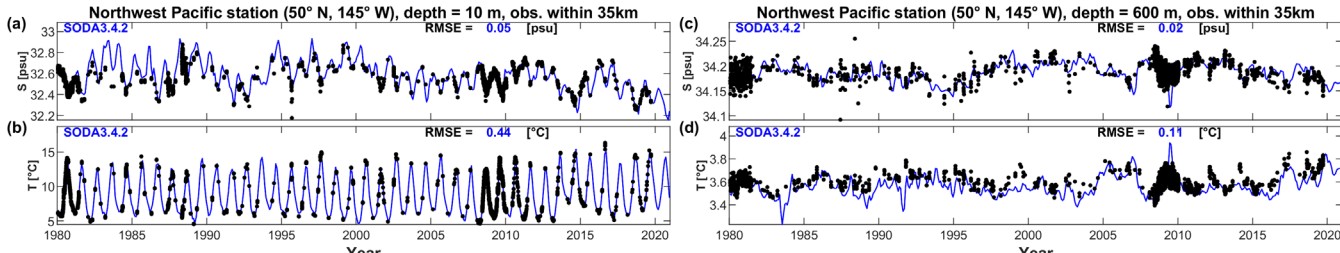

**Figure A6.** Northwest Pacific station time series comparison, showing monthly estimates from the SODA3.4.2 reanalysis product (blue lines), and all in situ observations within 35 km of the nominal station location (50° N, 145° W) (black dots). Subplots show results for salinity (S, upper) and temperature (T, lower) at 10 m depth (left) and 600 m depth (right), applying linear vertical interpolation to product and observational profiles where necessary. Each subplot shows Root Mean Square Error (RMSE) values relative to the observational values shown.



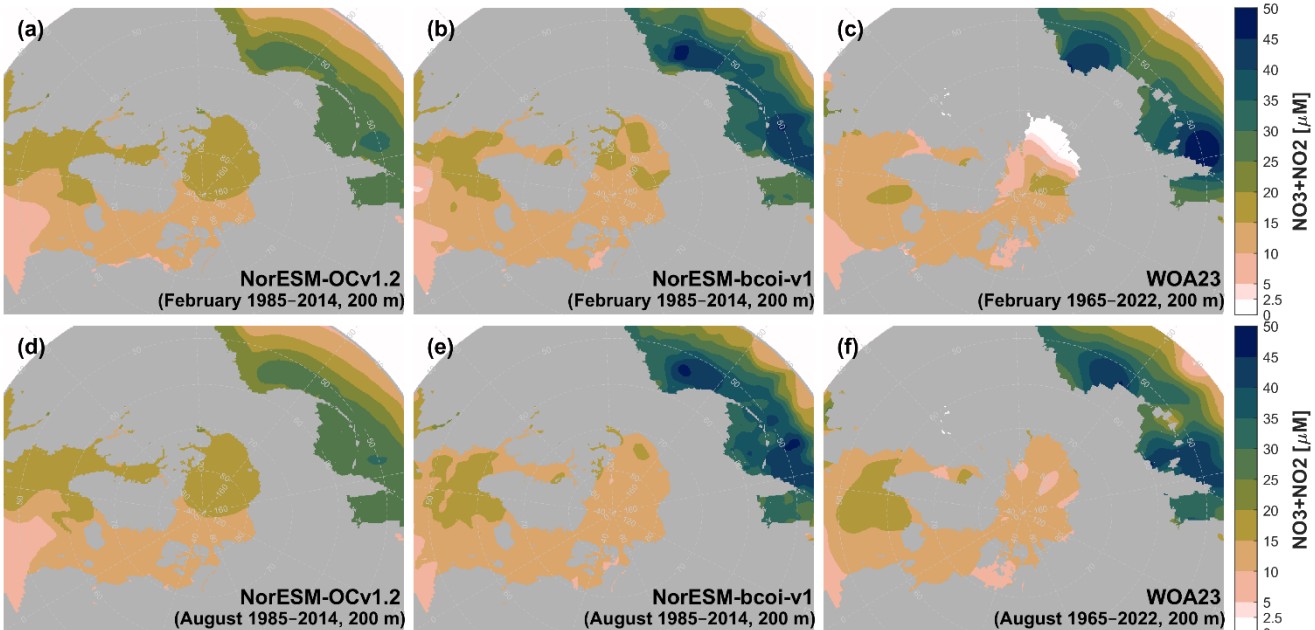

**Figure A7.** Climatology comparison for nitrate-plus-nitrite at 200 m depth, for February (upper) and August (lower), showing estimates from NorESM-OCv1.2 (left), NorESM-bcoi-v1 (middle), and the World Ocean Atlas 2023 (right). All concentrations are in in-situ volume-specific units (μmol L$^{-1}$).

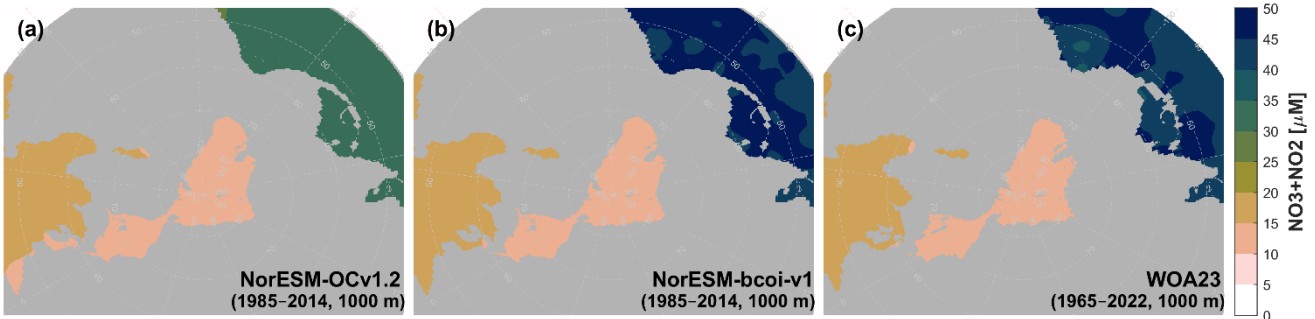

**Figure A8.** Annual climatology comparison for nitrate-plus-nitrite at 1000 m depth, showing estimates from NorESM-OCv1.2 (left), NorESM-bcoi-v1 (middle), and the World Ocean Atlas 2023 (right). All concentrations are in in-situ volume-specific units (μmol L$^{-1}$).





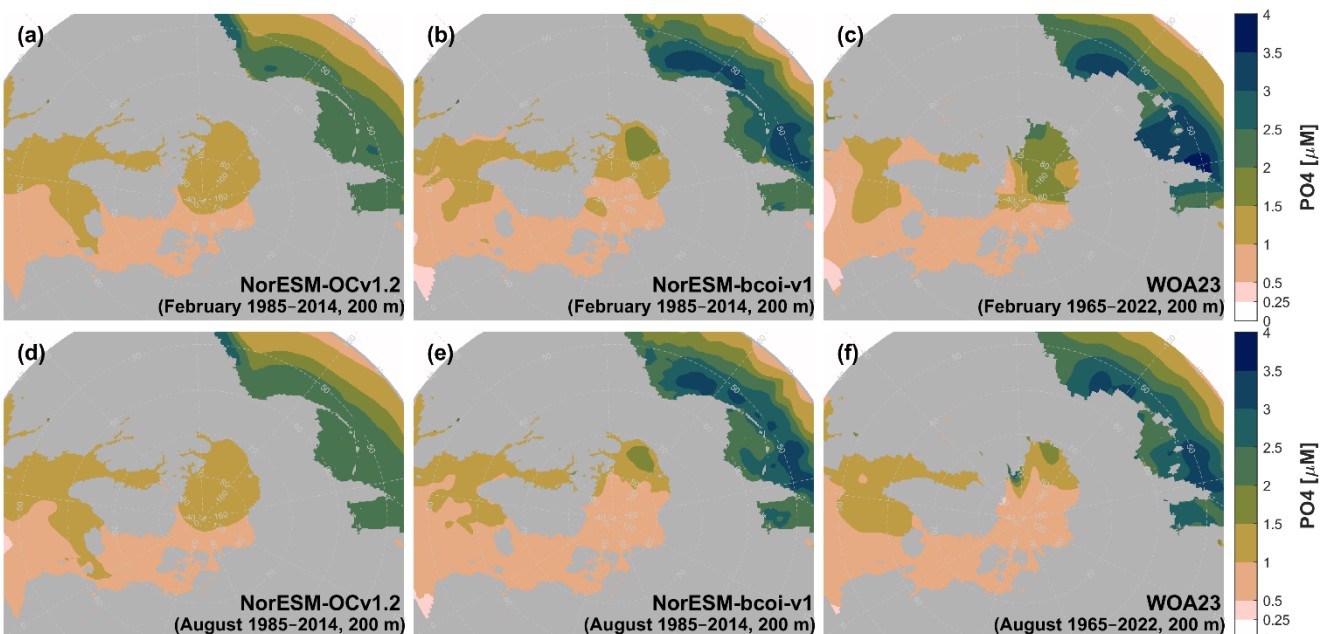

**Figure A9. Climatology comparison for phosphate at 200 m depth, for February (upper) and August (lower), showing estimates from NorESM-OCv1.2 (left), NorESM-bcoi-v1 (middle), and the World Ocean Atlas 2023 (right). All concentrations are in in-situ volume-specific units (µmol L⁻¹).**

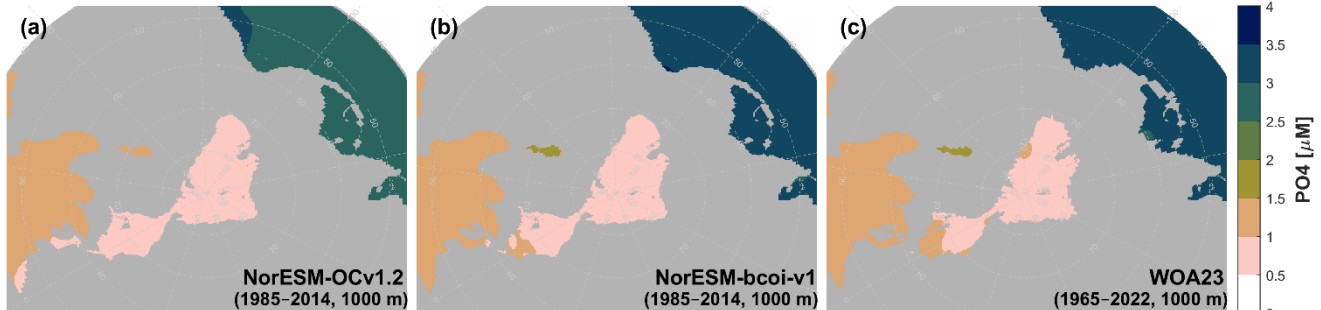

**Figure A10. Annual climatology comparison for phosphate at 1000 m depth, showing estimates from NorESM-OCv1.2 (left), NorESM-bcoi-v1 (middle), and the World Ocean Atlas 2023 (right). All concentrations are in in-situ volume-specific units (µmol L⁻¹).**



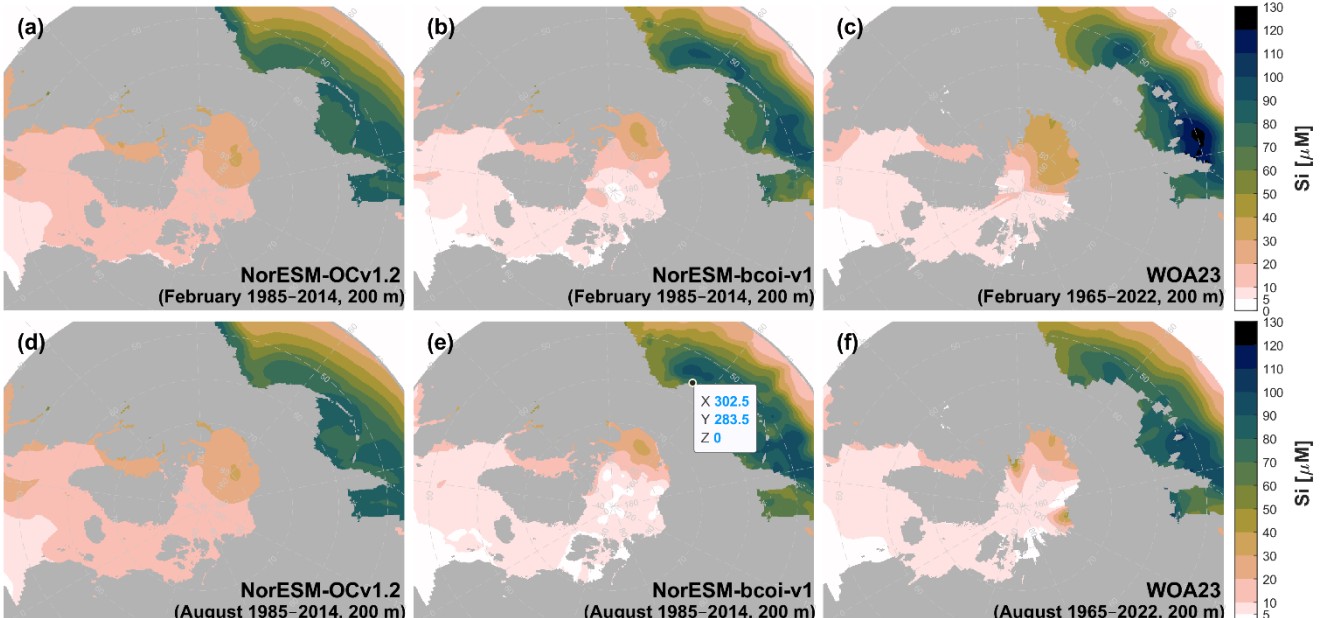

**Figure A11. Climatology comparison for silicate at 200 m depth, for February (upper) and August (lower), showing estimates from NorESM-OCv1.2 (left), NorESM-bcoi-v1 (middle), and the World Ocean Atlas 2023 (right). All concentrations are in in-situ volume-specific units (μmol L⁻¹).**

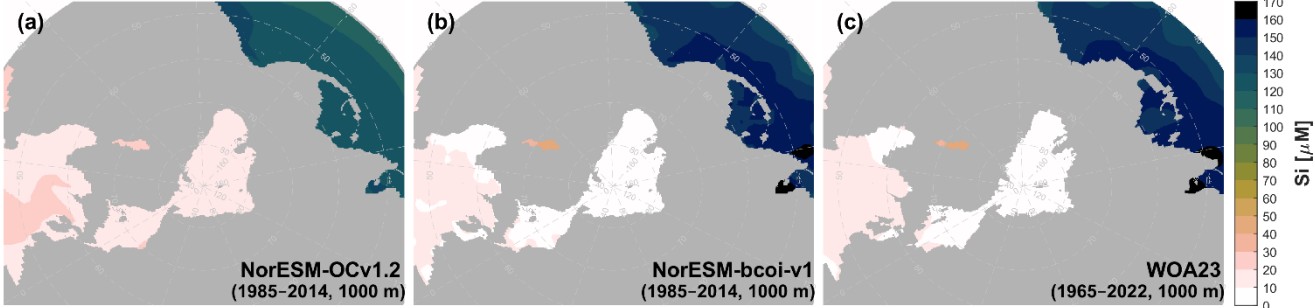

**Figure A12. Annual climatology comparison for silicate at 1000 m depth, showing estimates from NorESM-OCv1.2 (left), NorESM-bcoi-v1 (middle), and the World Ocean Atlas 2023 (right). All concentrations are in in-situ volume-specific units (μmol L⁻¹).**





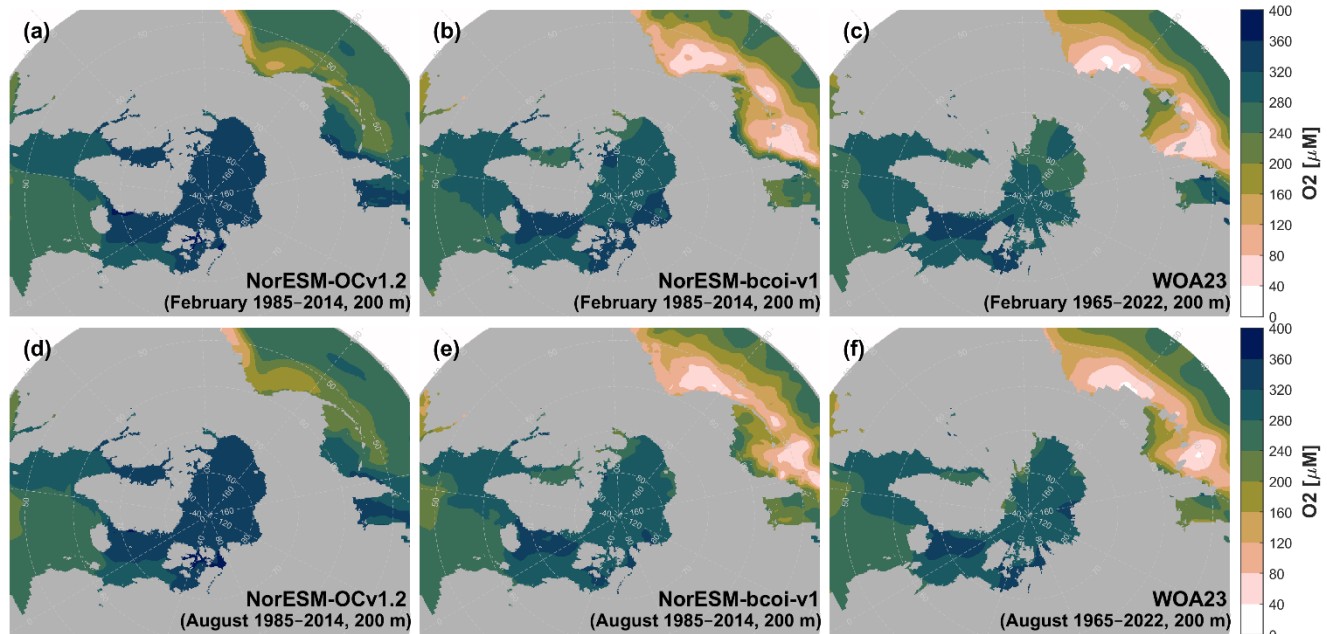

**Figure A13. Climatology comparison for dissolved oxygen at 200 m depth, for February (upper) and August (lower), showing estimates from NorESM-OCv1.2 (left), NorESM-bcoi-v1 (middle), and the World Ocean Atlas 2023 (right). All concentrations are in in-situ volume-specific units (μmol L⁻¹).**

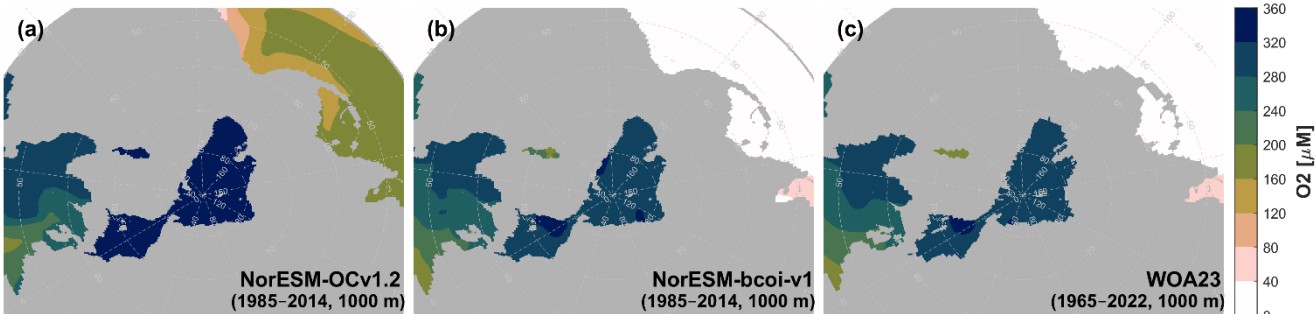

**Figure A14. Annual climatology comparison for dissolved oxygen at 1000 m depth, showing estimates from NorESM-OCv1.2 (left), NorESM-bcoi-v1 (middle), and the World Ocean Atlas 2023 (right). All concentrations are in in-situ volume-specific units (μmol L⁻¹).**



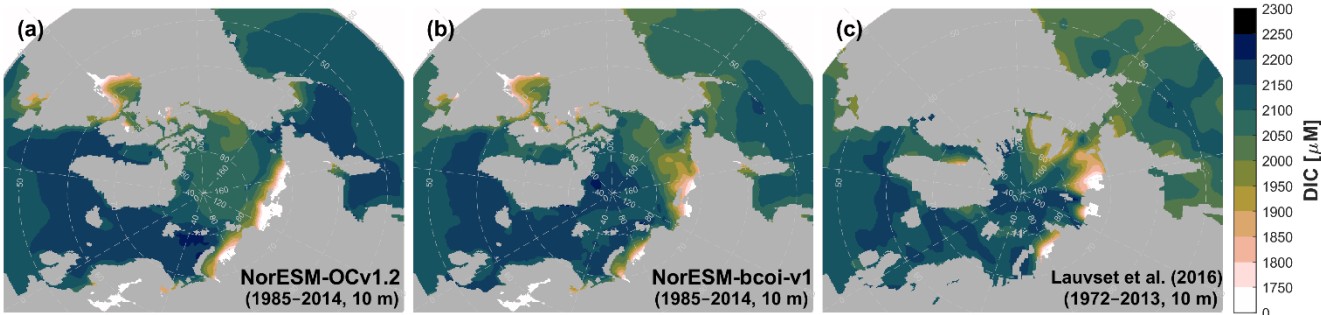

**Figure A15. Annual climatology comparison for dissolved inorganic carbon at 10 m depth, showing estimates from NorESM-OCv1.2 (left), NorESM-bcoi-v1 (middle), and the GLODAP gridded product (Lauvset et al., 2016) (right). All concentrations are in in-situ volume-specific units (μmol L⁻¹).**

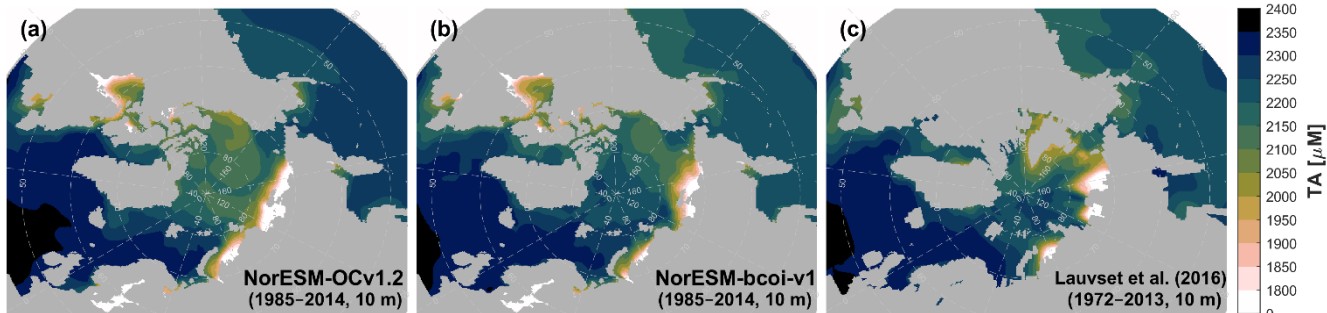

**Figure A16. Annual climatology comparison for total alkalinity at 10 m depth, showing estimates from NorESM-OCv1.2 (left), NorESM-bcoi-v1 (middle), and the GLODAP gridded product (Lauvset et al., 2016) (right). All concentrations are in in-situ volume-specific units (μmol L⁻¹).**



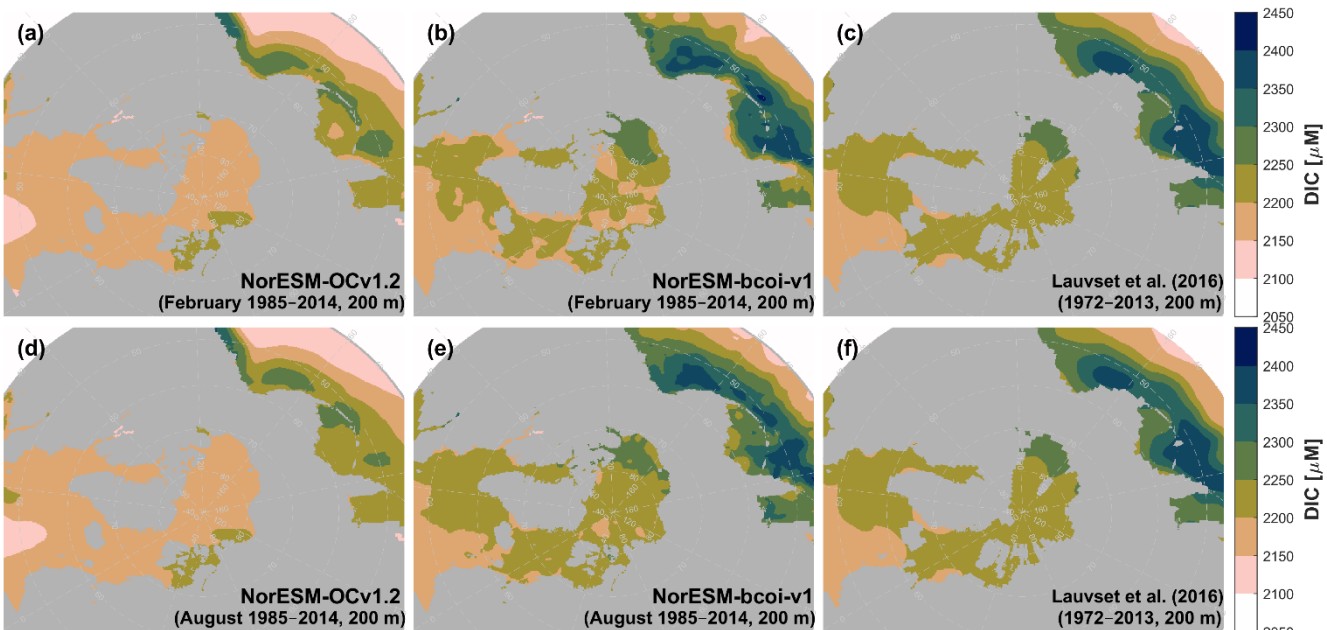

**Figure A17.** Climatology comparison for dissolved inorganic carbon at 200 m depth, for February (upper) and August (lower), showing estimates from NorESM-OCv1.2 (left), NorESM-bcoi-v1 (middle), and the GLODAP gridded product (Lauvset et al., 2016) (right). All concentrations are in in-situ volume-specific units (µmol L⁻¹).

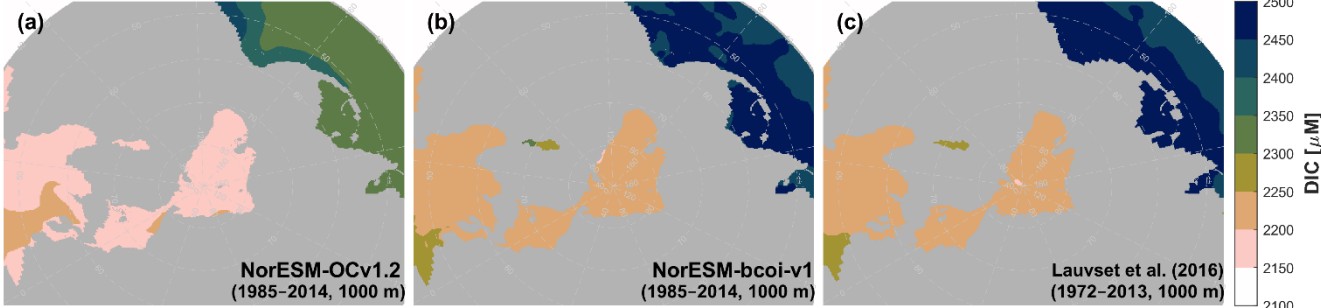

**Figure A18.** Annual climatology comparison for dissolved inorganic carbon at 1000 m depth, showing estimates from NorESM-OCv1.2 (left), NorESM-bcoi-v1 (middle), and the GLODAP gridded product (Lauvset et al., 2016) (right). All concentrations are in in-situ volume-specific units (µmol L⁻¹).

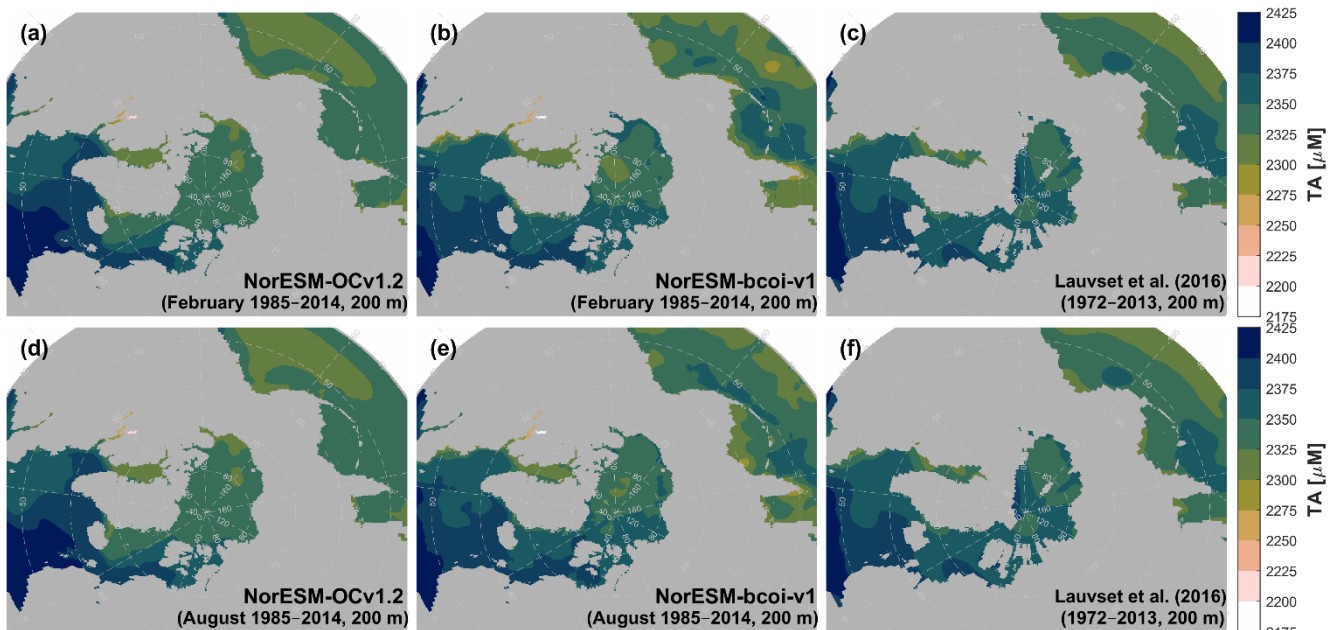

**Figure A19. Climatology comparison for dissolved inorganic carbon at 200 m depth, for February (upper) and August (lower), showing estimates from NorESM-OCv1.2 (left), NorESM-bcoi-v1 (middle), and the GLODAP gridded product (Lauvset et al., 2016) (right). All concentrations are in in-situ volume-specific units (μmol L⁻¹).**

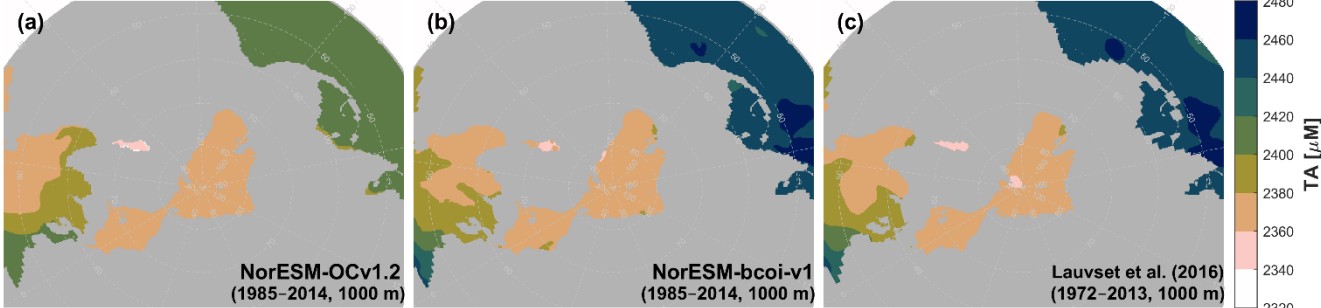

**Figure A20. Annual climatology comparison for dissolved inorganic carbon at 1000 m depth, showing estimates from NorESM-OCv1.2 (left), NorESM-bcoi-v1 (middle), and the GLODAP gridded product (Lauvset et al., 2016) (right). All concentrations are in in-situ volume-specific units (μmol L⁻¹).**

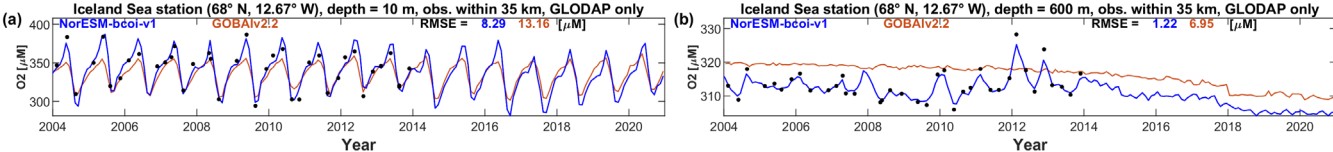

**Figure A21. Iceland Sea station time series comparison, showing monthly estimates of dissolved oxygen concentration (units μmol L⁻¹, in-situ volume specific) from the NorESM-bcoi-v1 product (blue lines) and the GOBAIv2.2 product (Sharp et al., 2023, orange lines), and all GLODAP-sourced observations from the NORBGCv1 dataset within 35 km of the nominal station location (68° N, 12.67° W) (black dots). Columns show data at sampled depths of 10 m (left) and 600 m (right), applying linear vertical interpolation to product profiles where necessary. Each subplot shows Root Mean Square Error (RMSE) values relative to observational values for the NorESM-bcoi-v1 product (blue) and the GOBAIv2.2 product (orange).**





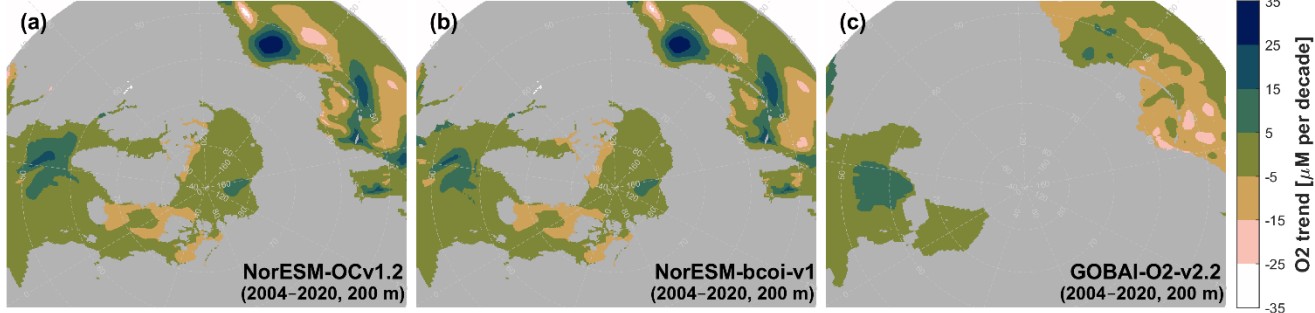

**Figure A22. Dissolved oxygen long term linear trends (17 years, 2004–2020) from annual averages at 200 m depth, showing estimates from the uncorrected hindcast NorESM-OCv1.2 (left), the final reanalysis product NorESM-bcoi-v1 (middle), and the GOBAI-O2-v2.2 gridded product (Sharp et al., 2023) (right). All trends are in in-situ volume-specific units (µmol L⁻¹ per decade).**

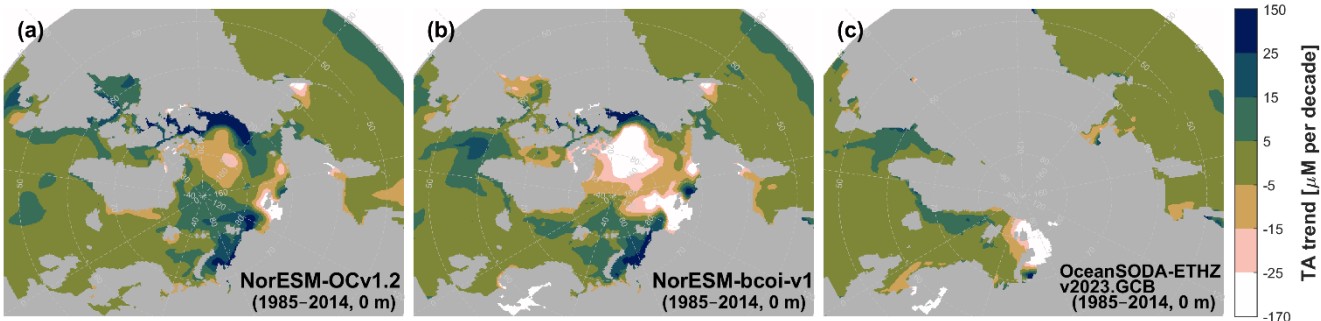

**Figure A23. Total alkalinity long term linear trends (30 years, 1985–2014) from annual averages at 0 m depth, showing estimates from the uncorrected hindcast NorESM-OCv1.2 (left), the final reanalysis product NorESM-bcoi-v1 (middle), and the OceanSODA-ETHZv2023.GCB gridded product (Gregor and Gruber, 2021) (right). All trends are in in-situ volume-specific units (µmol L⁻¹ per decade).**

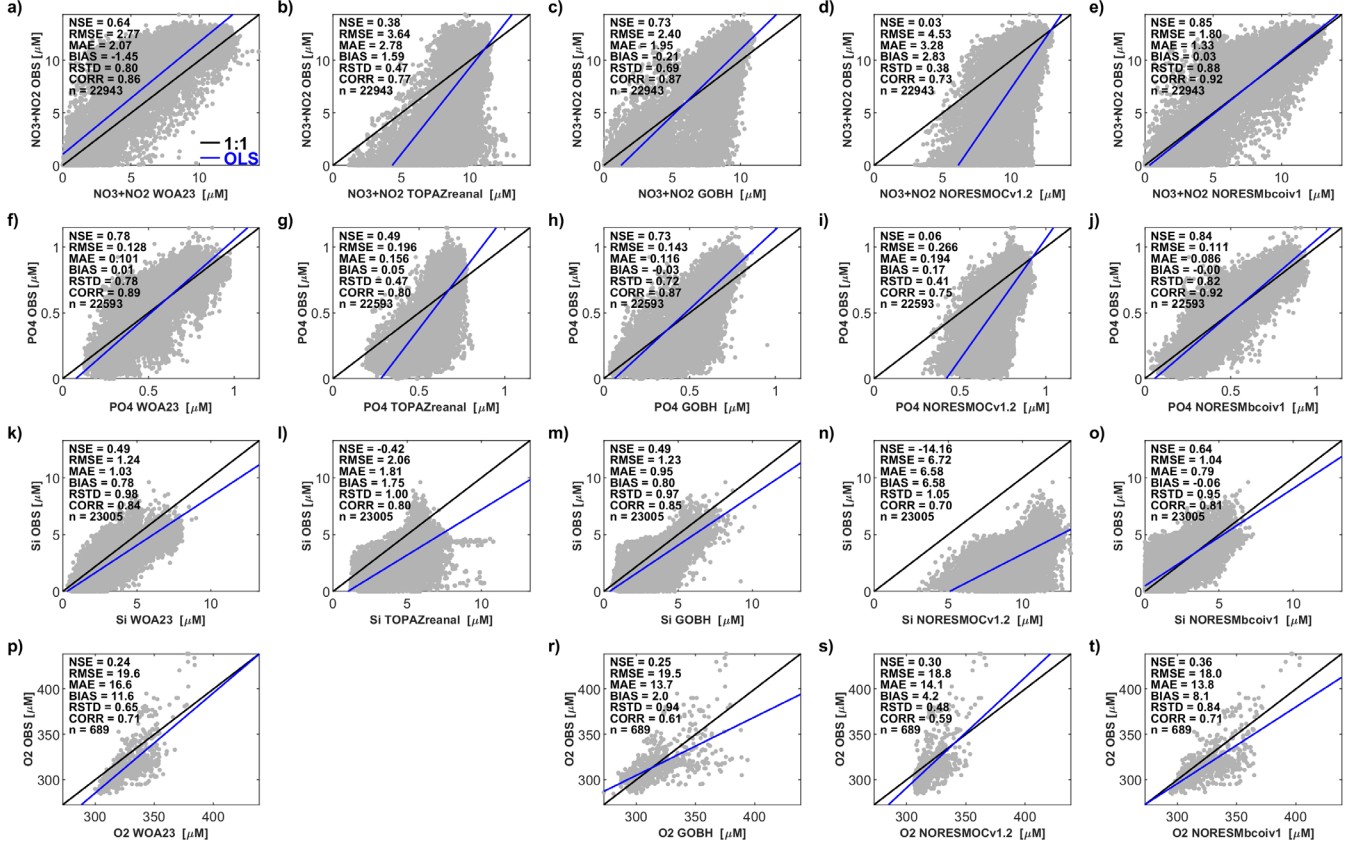

**Figure A24. Matchup scatterplots for the IMR Barents Sea observational datasets (70–80° N, 10–60° E, ≥40 km from coast). Rows from top to bottom show results for: nitrate-plus-nitrite (NO3+NO2), phosphate (PO4), silicate (Si), and dissolved oxygen (O2), all in in-situ volume-specific units (µmol L⁻¹). Columns from left to right show results for: the World Ocean Atlas 2023 (WOA23), the CMEMS TOPAZ-ECOSMO biogeochemical reanalysis (Wakamatsu et al., 2022a,b), the CMEMS Global Ocean Biogeochemical Hindcast (GOBH, Perruche, 2019), the original NorESM model hindcast (NorESM-OCv1.2, Schwinger et al., 2016), and the final reanalysis products (NorESM-bcoi-v1) calibrated to the full NORBGCv1 compilations, which did not include any of the IMR Barents Sea data. Inset skill metrics show the Nash-Sutcliffe efficiency (NSE), the Root Mean Square Error (RMSE), the Mean Absolute Error (MAE), the mean of model-minus-observation residuals (BIAS), the ratio of modelled to observed standard deviation (RSTD), the Pearson correlation between observations and interpolated model values (CORR), and the total number of matchup pairs (n). Note that dissolved oxygen data were not provided by the TOPAZ-ECOSMO products, hence subplot q) is missing.**

## Appendix B1: Test subset selection procedure

First, we grouped the observational locations (as latitude-longitude pairs) into a set of distinct sampled locations that were at least 5 km separated from each other. This 'clustering' step helped to avoid eventual selection of test data locations that were too close to training data locations. Next, for each distinct sampled location, we calculated the minimum horizontal distance to another distinct sampled location. For DIC, we calculated the minimum distance to another sampled locations for either DIC or TA. We also calculated the minimum distances from each oceanic target grid point to any distinct sampled location. For this purpose we used a uniform target grid defined by a polar stereographic projection with grid spacing of 20 km, excluding



non-target regions of the Mediterranean and Black Seas and Hudson Bay, as well as land points defined using ETOPO1
bathymetry.

We next selected the test sample locations as follows. First, the target grid was divided into 1000 × 1000 km cells. Each cell
was assigned a quota of test data locations based on the target total number of test locations, set as 5 % of the total number of
distinct sampled locations, and the fraction of the total number of target wet grid points represented in that cell (excluding non-
target regions). These quotas helped to promote uniformity of test location distribution within the target grid. The quotas
were low-capped at the total number of sampled locations within each cell, and then adjusted upward to reach the target total
number of test locations. Next, for each cell, we extracted the distribution of minimum distances between target grid points
within the cell and distinct sampled locations, and computed $n$ evenly-spaced quantiles of minimum distance $d_i$, with $n$ the
quota of test locations for that cell. Then, for each of the $d_i$, we found the distinct sampled location within that cell with the
most similar minimum distance to any other distinct sampled location, and added this location to the test subset. Once added,
the mininum distance for the selected location was set to a missing value, to prevent repeated selection. After assigning distinct
test subset locations for all cells, the corresponding test subset of the full (unclustered) NORESM-matched dataset was
identified, and all other data within this dataset were assigned to the training subset.

We thus constructed training-test subsets that statistically mimic the level of spatial interpolation required to estimate values
at randomly chosen points within the final model data. To account for the use of $O_2$ reanalysis data in creating the (DIC, TA)
data (see Sect. 2.10), the distinct test subset locations for (TA, DICn, TAn) were fixed as those coinciding with the distinct test
subset locations selected for DIC (within 5 km tolerance); for $O_2$ we first added all locations within 5 km of the DIC test subset
locations as 'fixed' test subset locations, then applied the above random assignment procedure to make up the remainder of
the prescribed 5 % of distinct sampled locations. In this way, we gauranteed that any locations in the test subsets for (DIC,
TA, DICn, TAn) would be also in the test subset for $O_2$. Then when we back-transformed the (DICn, TAn) products based
only on training subsets, we used the $O_2$ product based only on the $O_2$ training subset.





**Appendix C: Supplementary Tables**

**Table C1. Similarity of the distributions of minimum distances between test subset observations and training subset observations, and the distributions of minimum distances between reanalysis product grid points and any observations used to calibrate the final product (NorESM-bcoi-v1). Here we assess the statistical similarity by comparing 25th, 50th, and 75th quantiles of the respective distributions for nitrate-plus-nitrite (NO3+NO2), phosphate (PO4), silicate (Si), normalized dissolved inorganic carbon (DICn), and normalized total alkalinity (TAn).**

| | 25th percentile minimum distance [km] | | 50th percentile minimum distance [km] | | 75th percentile minimum distance [km] | |
|---|---|---|---|---|---|---|
| | Test obs. to any training obs. | Grid points to any obs. | Test obs. to any training obs. | Grid points to any obs. | Test obs. to any training obs. | Grid points to any obs. |
| NO3+NO2 | 18.5 | 18.8 | 37.3 | 38.8 | 70.8 | 82.7 |
| PO4 | 16.8 | 16.7 | 32.7 | 33.2 | 61.5 | 70.2 |
| Si | 16.5 | 16.4 | 32.2 | 32.2 | 59.0 | 67.5 |
| DO | 10.0 | 9.6 | 18.2 | 19.1 | 38.7 | 41.4 |
| DICn | 32.4 | 35.0 | 57.0 | 73.8 | 94.7 | 155.1 |
| TAn | 34.1 | 36.7 | 57.4 | 77.3 | 96.4 | 163.4 |


**Table C2. Consistency of model and climatological products with observational data, as measured by Root Mean Square Error (RMSE) between interpolated product values and quality-controlled in situ observational data from test subsets of the NORBGCv1 compilations, restricted to only GLODAPv2 observations. Columns show results for nitrate-plus-nitrite (NO3+NO2), phosphate**
**(PO4), silicate (Si), and dissolved oxygen (O2), all in in-situ volume specific units (µmol/L). Test subset RMSE values are shown for the World Ocean Atlas 2023 climatologies (WOA23), the CMEMS TOPAZ-ECOSMO biogeochemical reanalysis (Wakamatsu et al., 2022a,b), the CMEMS Global Ocean Biogeochemical Hindcast products (GOBH, Perruche, 2019), the original NorESM model hindcast (NorESM-OCv1.2, Schwinger et al., 2016), and the reanalysis products based only on training subset data (NorESM-bcoi-v1(tr)). Results are shown for matchups within the Arctic region (>65° N, ≥40 km from coast) within the 2007–2019 period covered**
**by all products. The first row shows the number of product-observation matchups for each variable, applicable to all products (the same observation subsets are used for all products).**

| | NO3+NO2 [µM] | PO4 [µM] | Si [µM] | O2 [µM] |
|---|---|---|---|---|
| no. matchups | 2099 | 2063 | 2046 | 5079 |
| WOA23 | 1.33 | 0.161 | 3.57 | 11.7 |
| TOPAZ-ECOSMO | 3.53 | 0.396 | 7.61 | 29.5 |
| GOBH | 3.51 | 0.302 | 7.71 | 25.1 |
| NorESM-OCv1.2 | 5.51 | 0.318 | 20.28 | 38.5 |
| NorESM-bcoi-v1(tr) | 1.26 | 0.145 | 3.42 | 11.5 |




**Acknowledgments**

We gratefully acknowledge financial support from: the Joint Programming Initiative Healthy and Productive Seas and Oceans

(JPI Oceans) under the project CE2COAST (RCN no. 321890), the European Union Horizons 2020 and Horizon Europe programmes under the project Invest4Nature (Grant no. 101061083) and OceanICU (Grant no. 101083922), the Research Council of Norway under the project Migratory Crossroads (RCN no. 344079), and the Cumulative impact of multiple stressors in High North ecosystems (CLEAN) project funded through the Fram Centre Research Cooperation Program. We also acknowledge Norwegian Metacenter for Computational Science and Storage Infrastructure (Notur/Norstore) projects nn9490k,

nn8103k, nn2980k, ns2980k, and ns9630k. Inspiration for methodological aspects of this work drew in part from discussions between the lead author and Andrew Solow at Woods Hole Oceanographic Institution, under the WHOI postdoctoral scholar program. We also wish to express our sincere thanks for all the hard work of those involved in observational data collection and stewardship, and to all those who have contributed their project data to the Norwegian and international databases utilized in this study.

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
