# Peer review of "NorESM-bcoi-v1: A bias-corrected reanalysis of ocean biogeochemistry at >40° N, 1980–2020, based on a global ocean model hindcast"

_Earth System Science Data, 2025_

## Referee Comment (RC1)

[referee-annotated manuscript omitted]

---

## Author Response (AR1)

This paper describes what looks like an extremely useful data product for usage in regional modelling as boundary and initial conditions, to evaluate model hindcasts, or to provide forcing for biological models. The data product is available in a Zenodo address provided by the authors.

I strongly recommend the publication of this paper.

We thank the reviewer for taking the time to review our manuscript and for this positive response!  Reponses to reviewer questions and suggestions are inserted below in blue.

I made some comments directly on the manuscript. There is a question I have to the authors about the usage of a product based on a training dataset (NorESM-bcoi-v1(tr)) and the complete product based on the whole dataset (NorESM-bcoi-v1). I wonder why showing the results of both products, when the latter is the one of most interest to potential users, given its lower bias? I understand that when evaluating a model, we want to use a dataset that was not used for calibrating/learning the model, but this is not a case of model validation – here the goal is to get a dataset with the minimum possible bias. Therefore, it is not entirely clear to me the usefulness of adding both products in the paper.

The short answer here is that skill analysis of the products based on training subsets (NorESM-bcoi-v1(tr)) is needed in order to estimate the 'out-of-sample' skill of the final products (NorESM-bcoi-v1), i.e. the skill at general locations and times beyond the calibration data of the final products. These analyses also aim to provide a more fair assessment of the skill advantage of the final products in comparison with alternative products (WOD23 etc.).

We show the results of NorESM-bcoi-v1(tr) only in regard to skill assessment over test datasets excluded from the training datasets (in Tables 3–6).  These training products are not shown in any of the figures and are not final products made available to users (see Figure 1).

We show the skill of NorESM-bcoi-v1(tr) over test datasets because this should better approximate the skill of the final product NorESM-bcoi-v1 *at any, likely unsampled, time and location* within the regions/periods defined in the Tables.  In other words, it is an

attempt to approximate the 'out-of-sample' skill, or the skill at new sampled locations not included in the final observational datasets (NORBGCv1) that were used to calibrate our final products. We have attempted to design the (disjoint) training and test subsets such that the level of spatial interpolation required between training and test subsets reflects (as well as possible) the level of spatial interpolation that will be required of the final product when it is used to estimate values at arbitrary locations and times, after calibration to the full NORBGCv1 dataset.

If users were only interested in product skill at locations and times that are sampled in the NORBGCv1 datasets, then the in-sample skill scores based on matchups with NorESM-bcoi-v1 would indeed be more relevant. However, our basic assumption is that users are most interested in the out-of-sample skill of the products at general times and locations (within the regions/periods defined in the Tables), since this should reflect the accuracy of the products when used e.g. to set boundary conditions for regional models. For this latter purpose, the skill scores for NorESM-bcoi-v1 in Tables 3–6 will be biased in an optimistic sense. The skill scores for NorESM-bcoi-v1(tr) should be less biased, because of the way we have designed the training and test subsets.

The skill results for NorESM-bcoi-v1(tr) also enable a more 'fair' comparison with other products. If we only showed skill results for NorESM-bcoi-v1, these would have an unfair advantage over alternative products (WOA23 etc.) because the NorESM-bcoi-v1 products have been fitted to the full NORBGCv1 datasets that include the data subsets used to assess skill in Tables 3–6. The use of NorESM-bcoi-v1(tr) makes the comparisons generally fairer, although in some cases the comparisons could be slightly unfair *against* our products, because some proportions the test datasets would have been used to calibrate the alternative products, while none of the test data were used to calibrate NorESM-bcoi-v1(tr) products. In Table 4 these proportions are quantified in the column "% GLODAP or WOD-PFL" which estimates how much of the test data were included in the calibration of the GOBAIv2.2 alternative product. This may make the estimated skill advantage of our final products relative to the alternative products slightly conservative when based on the NorESM-bcoi-v1(tr) results.

By contrast, for comparisons with the IMR Barents Sea datasets in Table 7, use of NorESM-bcoi-v1(tr) is not required because none of these 'validation' data were included in NORBGCv1. We can therefore assess the out-of-sample skill of NorESM-bcoi-v1 directly using matchups with the NorESM-bcoi-v1 products, hence there is no row in Table 7 for NorESM-bcoi-v1(tr).

In preparing the manuscript we considered whether it would be less confusing to show the skill scores of only one set of products (either NorESM-bcoi-v1(tr) or NorESM-bcoi-v1) in each Table. However, comparing the two does provide some useful additional quality control, confirming that: 1) the final products give consistently better skill scores relative to the test data (as they must!), and 2) that the improvement in skill moving from

training to final products is not *drastic*, lying in the range 0-50% RMSE reduction over all subsets in Tables 3.  If this latter improvement were drastic (e.g. 50-100% RMSE reduction), this would be a warning sign, suggesting that the model calibration is not robust, or that the model 'over-accommodates' the observational data, possible leading to unsmooth spatio-temporal variability in the vicinity of fitted data (e.g. 'jumps' or 'tent-pole' patterns).

In the revised manuscript we have made some modifications to improve clarity.  We had an internal debate among coauthors whether use of the term 'interpolative skill' should be replaced by 'out-of-sample skill' throughout the manuscript.  The former was used in the original manuscript on the basis that it might be more accessible (and less 'statistical') for marine science readers. However, the latter is a more precise term, it yields an excellent web-based definition e.g. from GoogleAI, and with the burgeoning of AI methods in marine science since we starting working on this manuscript we feel that the term 'out-of-sample' is already more common, and is likely to become yet more so in coming years.  We have therefore replaced 'interpolative' with 'out-of-sample' in the revised manuscript.

In Section 2.11 where the training/test subset design is explained in detail, we have modified an introductory sentence, briefly defining the term 'out-of-sample skill':

"To estimate the out-of-sample skill of the final NorESM-bcoi-v1 products, i.e. the skill at general locations and times beyond the calibration data, we divided the observational data into 'training' subsets, used these to develop provisional model products, NorESM-bcoi-v1(tr), and measured the skill of these products against disjoint 'test' subsets (see Figure 1)."

Furthermore, and in consideration of the second reviewer's comment that some readers may wish to skip the whole of Section 2, we have made some modifications to the first section of the Results. The first two paragraphs now read:

"Table 3 shows the Root-Mean-Square-Error (RMSE) for various products over test data subsets of quality-controlled in situ observations from NORBGCv1, showing also results further restricted to the GLODAP/CARINA source as a gold standard, and to different spatial subsets.  Fig. A3 shows matchup scatterplots and other skill metrics for the Northern Seas subsets (first section of Table 3).  We first compare the RMSE of the original model hindcast (NorESM-OCv1.2) with that of the bias-corrected hindcast based only on the training subsets (NorESM-OCv1.2bc(tr)) and of the full reanalysis product based only on the training subsets (NorESM-bcoi-v1(tr)).  Here the RMSE values estimate the out-of-sample accuracy of the final products; the test subset locations include none of the training subset locations, and are selected to require a level of interpolation that is consistent with application of the final products at general grid

locations (see section 2.11 and Figs. 4, 5).  Table 3 shows that the reanalysis achieves a large improvement in out-of-sample accuracy relative to the original hindcast, for all variables and all spatial domains, and most of the improvement is obtained by the bias correction step (see NorESM-OCv1.2bc(tr)), with a small further improvement obtained by anomaly kriging (see NorESM-bcoi-v1(tr)).  RMSE is generally reduced by a factor 2–6 relative to the original hindcast.

Comparing with the WOA23 climatological products for nutrients and dissolved oxygen, NorESM-bcoi-v1(tr) achieves similar RMSE, usually slightly lower (14 out of 20 comparisons in Table 3).  Considering the full test subset (first section of Table 3), the RMSE of NorESM-bcoi-v1(tr) is 4–6% lower for nitrate and phosphate, ~1% higher for silicate, and 19% lower for dissolved oxygen.  Since the test subset data were excluded from the NorESM-bcoi-v1(tr) development, but were likely mostly included in the WOA23 product development, these comparisons favour WOA23 and may give conservative estimates of the improvement in out-of-sample accuracy achieved by NorESM-bcoi-v1. The final products based on training+test datasets (NorESM-bcoi-v1) achieve lower RMSE than WOA23 in all cases (19–42% lower), showing that they are more consistent with the observational data at the sampled times and locations.  Also, note that the reductions in RMSE moving from NorESM-bcoi-v1(tr) to NorESM-bcoi-v1 are only moderate, in the range 0–50%; more drastic reductions (e.g. 50–100%) could have indicated lack of robustness in the final calibrations, or excessive accommodation of observational data leading to un-smooth spatial or temporal variability."

The authors present Root Mean Square Errors to compare the performance of the developed products with other products and they show that NorESM-bcoi-v1 performs better for the selected biogeochemical variables. They also present time series to show that this new product captures seasonal variability as well as long-term trends. Finally, they make comparisons with climatological data. So, I think they make a relatively in-depth assessment of their results. I would suggest the usage of Taylor diagrams to compare different products based on their bias, correlation with observations and variability. I am not implying that this "must" be done, but I am just suggesting them to consider doing it.

We thank the reviewer for these positive remarks and an interesting suggestion.  After some consideration, we have decided we would not like to add Taylor (or Target) diagrams to the manuscript.  The main issue here is the length and number of figures in the existing manuscript; we are concerned that adding further figures could detract from the overall accessibility and perhaps draw attention from the RMSE numbers provided in the Tables which, we feel, may provide a more concrete accuracy benchmark against which future products can be compared (bearing in mind that the

effort we have put into enabling that future product development can exactly reproduce the skill assessments shown in the manuscript).

My comments here and directly on the manuscript imply merely a minor revision.

RC2 comments

I have read and reviewed the manuscript by Wallhead et al. The paper presents a very well-documented dataproduct derived from a combination of data and model hindcasts. The figures are of great quality, and the uncertainties, advantages, and disadvantages of the approach are very well explained. I have just a few comments that could improve the accessibility of the manuscript.

We thank the reviewer for committing the time to review our manuscript and for this positive feedback.  Reponses to reviewer questions and suggestions are inserted below in blue.

First of all, this is a highly specialized paper, with lots of jargon. This makes it hard for a non-expert to follow the text. While this is not a problem in itself, and you do include a section discussing uses and limitations in accessible language (section 3.7), it would be useful to refer to this section immediately after the introduction – which would improve accessibility for non experts who might be turned away because of the dense description of the methods. This could go alongside some more info about the structure of the netcdf files in the README of the database itself to make reading them easier for the users.

These are fair points and we thank the reviewer for these helpful suggestions.  In the revised manuscript, we have added the following paragraph to the end of the Introduction:

"The paper is structured as follows.  Section 2 provides detailed Methods; this is a technical section that can be skipped without losing information about the performance of the products or how they should be used, although Figure 1 may still be consulted for a general overview.  Section 3 presents the Results, focusing on the performance of the new products in comparison to existing alternative products. Section 4 provides a Discussion of potential uses and limitations of the new products. This has the most accessible language and we strongly recommend that it be read by all potential users of the new products.  Section 5 provides Conclusions in a concise format with limited technical language."

We have also revised and expanded the relevant section of the README in the database itself.  It now reads:

"Reanalysis data in files <variable name>_NorESM-bcoi-v1_40N_1980-2020.nc are 4D gridded biogeochemical datasets covering latitudes >40 degrees N and years 1980-2020 inclusive.  Spatial resolution is on the original NorESM-OCv1.2 grid (27–82 km in

the horizontal, with higher resolution further north, 5–1000 m in the vertical, higher near the surface).  Temporal resolution is monthly.

NorESM-bcoi-v1 reanalysis data are provided in NetCDF format for: no3no2 = nitrate plus nitrite (NO3+NO2 in manuscript), po4 = phosphate, si = silicate, o2 = dissolved oxygen, dissic = dissolved inorganic carbon (DIC in manuscript), talk = total alkalinity (TA in manuscript), dissicn = normalized dissolved inorganic carbon (DICn in manuscript), talkn = normalized total alkalinity (TAn in manuscript).  Each of these variables is provided in a separate NetCDF file.  All units are mol/m3 in the corresponding in situ conditions of (temperature, salinity, pressure).

Each NetCDF file contains, in addition to one of the above reanalysis variables, the following variables:

depth: mid-layer depths of the reanalysis data (m). This is a 1D variable describing uniform depth levels ranging from 1.25 m to 7312.5 m, with irregular spacing, same as the original NorESM hindcast data.

time: nominal timestamps of the reanalysis data (days since 1970-01-01 00:00:00). This is a 1D variable describing the nominal timestamps of the monthly reanalysis data, positioned on the 15th day of each month and covering 1980-01-15 through 2020-12-15.

depth_bnds: depths of upper and lower layer surfaces (m). This is 2D variable describing the depths of upper/lower depth layer surfaces, where the second dimension 'bounds' specifies upper/lower, and the values range over 0-8000 m, same as the original NorESM hindcast data.

lat: latitude of the horizontal grid locations (degrees north). This is a 2D variable describing the latitude of the each horizontal grid location defined by the dimension variables (x, y).  Values range over 40.4 to 89.8 degN and comprise a subset of the original NorESM hindcast values.  NOTE: The NorESM grid is irregular over (latitude, longitude), hence the latitudes and longitudes are described by 2D rather than 1D variables.

lon: longitude of the horizontal grid locations (degrees east). This is a 2D variable describing the longitude of the each horizontal grid location defined by the dimension variables (x, y).  Values range over -180.0 to 180 degE.

The reanalysis variables themselves are dimensioned over (time, depth, y, x).  This does mean that care is needed to extract subsets defined by latitude-longitude ranges.  One must first establish subselection indices over the dimension variables (x, y) using the 2D variables (lat, lon).  The irregular grid spacing over latitude-longitude should also be noted when interpolating the reanalysis output to new locations."

L163: a bit confusion you end here by saying coastal data were not excluded, but then your example only has data >40km offshore – which does exclude coastal data. And you do mention this again at L354. So in the end you do remove coastal data. Why not do so from the beginning?

Coastal data are excluded from calibration of the NorESM-bcoi-v1 products by inclusion of the model land mask in the interpolation mesh, thus forbidding coastward extrapolation, see Section 2.7:

"No extrapolation was allowed over time or horizontally towards model coastlines (this was achieved by including land mask cells with missing values in the interpolation mesh)."

Coastal data were further excluded from product skill assessment and validation by imposing a >40 km criterion, see Section 2.11:

"Given the basin-scale focus of this study, and to avoid potential influence of near-coastal extreme values (e.g. of nutrient concentrations), we further restricted real coastline, using the intermediate resolution data from GSHHGv2.3.7 (Wessel and Smith, 1996)."

*In addition*, any comparisons with alternative products are restricted to the subset for which *all* products in the multi-product comparison are forbidden to make coastward extrapolation. Given the 1-degree resolution of most of the alternative products, this latter requirement makes the >40 km criterion mostly redundant, but we include it to be sure that all skill assessment data are at least 40 km from the standard coastline dataset. This is noted also in Section 2.11:

"Also, for comparing skill between different products, we further restrict to the common subset of test data that can be reached by all products without extrapolation in time, downwards, or 'coastwards' towards land masses, such that exactly the same set of observations is used in each product observation comparison."

However, at the analysis stage described in L163, the coastal data are not excluded. We have clarified this in the revised text:

"Allowances were made for possible coastal effects (e.g. river inputs) by plotting data locations with respect to land masses (note: coastal data were thus not excluded at this stage)."

We do not exclude the coastal data at this early stage because:

1) We want to allow use of data ≤40 km from coastlines in calibration of the NorESM-bcoi-v1 products as long as they pass quality control and are not requiring coastward extrapolation from the NorESM grid points.

2) We want to include the coastal data in the NORBGCv1 observational datasets, primarily so that these data may be used in future skill analyses considering weaker coastal data exclusion criteria (e.g. >20 km from coastlines).

L224: filename in the data archive is NORBGCv1_NUT.nc I presume (please double check throughout the manuscript)

Thanks for spotting that.  In the revised manuscript we have searched and corrected all erroneous instances of 'NORBGC_'.